

**A systems-based approach to parameterise seismic hazard in regions with little historical or**
**instrumental seismicity: The South Malawi Active Fault Database**
Jack N. Williams[a*], Hassan Mdala[b], Åke Fagereng[a], Luke N.J. Wedmore[c], Juliet Biggs[c], Zuze
Dulanya[d], Patrick Chindandali[e], Felix Mphepo[b]
**Affiliations**
[a]*School of Earth and Ocean Sciences, Cardiff University, Cardiff, UK*
[b]*Geological Survey Department, Mzuzu Regional Office, Mzuzu, Malawi*
[c]*School of Earth Sciences, University of Bristol, Bristol, UK*
[d]*Geography and Earth Sciences Department, University of Malawi, Zomba, Malawi*
[e]*Geological Survey Department, Zomba, Malawi*
*Corresponding author: Jack Williams (williamsj132@cardiff.ac.uk)
**Abstract**
Seismic hazard is frequently characterised using instrumental seismic records. However, in regions
where the instrumental record is short relative to earthquake repeat times, extrapolating it to estimate
seismic hazard can misrepresent the probable location, magnitude, and frequency of future large
earthquakes. Although paleoseismology can address this challenge, this approach requires certain
geomorphic settings and carries large inherent uncertainties. Here, we outline how fault slip rates
and recurrence intervals can be estimated through an approach that combines fault geometry,
earthquake-scaling relationships, geodetically derived regional strain rates, and geological
constraints of regional strain distribution. We then apply this approach to the southern Malawi Rift
where, although no on-fault slip rate measurements exist, there are theoretical and observational



constraints on how strain is distributed between border and intrabasinal faults. This has led to the
development of the South Malawi Active Fault Database (SMAFD), the first database of its kind in
the East African Rift System (EARS) and designed so that the outputs can be easily incorporated
into Probabilistic Seismic Hazard Analysis. We estimate earthquake magnitudes of Mw 5.4-7.2 for
individual fault sections in the SMAFD, and Mw 6.0-7.8 for whole fault ruptures. These potentially
high magnitudes for continental normal faults reflect southern Malawi's 11-140 km long faults and
thick (30-35 km) seismogenic crust. However, low slip rates (intermediate estimates 0.05-0.8
mm/yr) imply long recurrence intervals between events: $10_2$-$10_5$ years for border faults and $10_3$-$10_6$
years for intrabasinal faults. Sensitivity analysis indicates that the large range of these estimates can
be reduced most significantly from an improved understanding of the rate and partitioning of rift-
extension in southern Malawi, earthquake scaling relationships, and earthquake rupture scenarios.
Hence these are critical areas for future research. The SMAFD provides a framework for using
geological and geodetic information to characterize seismic hazard in low strain rate settings with
few on-fault slip rate measurements, and could be adapted for use elsewhere in the EARS or
globally.
**1.  Introduction**
Earthquake ruptures tend to occur on pre-existing faults (Brace and Byerlee, 1966; Jackson, 2001;
Scholz, 2002; Sibson, 1989). Thus, the identification and systematic mapping of active faults, which
are then compiled with other fault attributes (e.g. slip rate and slip sense) into a geospatial active
fault database, provides an important tool in assessing regional seismic hazard (Christophersen et al.,
2015; Hart and Bryant, 1997; Langridge et al., 2016; Shyu et al., 2016; Styron et al., 2020; Taylor
and Yin, 2009). Not only can these databases inform on the surface rupture risk (Hart and Bryant,
1997; Villamor et al., 2012), they can also be converted into earthquake sources for Probabilistic
Seismic Hazard Assessment (PSHA) to forecast future levels of ground shaking (e.g. Beauval et al.,





2018; Hodge et al., 2015; Stirling et al., 2012). Furthermore, the data contained in active fault
databases are inherently useful in understanding regional geological evolution (Agostini et al.,
2011b; Basili et al., 2008; Taylor and Yin, 2009).
Despite their benefits, active fault databases have yet to be developed for many seismically active
regions (Christophersen et al., 2015). This partly reflects the difficulty in estimating fault slip rates
and earthquake recurrence intervals, as instrumental seismic records typically cover only a fraction
of a fault's seismic cycle (Stein et al., 2012), whilst obtaining these attributes from dating offset
surfaces and/or paleoseismology requires certain geomorphic settings and can involve large
uncertainties (Cowie et al., 2012; McCalpin, 2009; Nicol et al., 2016b). Alternatively, decadal time-
scale fault slip rates can be estimated using geodetic estimates and block models where the crust is
divided by mapped faults (e.g. Field et al., 2014; Wallace et al., 2012; Zeng and Shen, 2014).
However, not all plate boundaries are covered by sufficiently dense geodetic networks to perform
this analysis, and/or sometimes geodetic data cannot resolve how strain is distributed (Calais et al.,
2016; Stein et al., 2012).
In this study, by combining geodetic and geological information, we present a new systems-based
approach to estimate slip rates and earthquake recurrence intervals within narrow (<100 km width;
Buck, 1991) amagmatic continental rifts. However, this method could be adapted for any region
with low strain rates, well developed active fault maps, and an understanding of strain partitioning.
We then apply this method to southern Malawi, which has culminated in the development of the
South Malawi Active Fault Database (SMAFD).
The SMAFD represents the first active fault database to be developed within the East African Rift
System (EARS), where population growth and seismically vulnerable building stock is driving an





increased exposure to seismic hazard (World Bank, 2019; Goda et al., 2016; Hodge et al., 2015;
Kloukinas et al., 2019; Ngoma et al., 2019; Novelli et al., 2019). Notably, previous PSHA in the
EARS has typically been conducted using the ~65 year long instrumental record of seismicity alone
(Ayele, 2017; Goitom et al., 2017; Midzi et al., 1999; Poggi et al., 2017). However, low EARS strain
rates (regional extension rates ~1-6 mm/yr; Stamps et al., 2018) imply that this record is incomplete
and may underestimate seismicity rates compared to those inferred from geodesy and
paleoseismology (Ebinger et al., 2019; Hodge et al., 2015; Stein et al., 2012; Vallage and Bollinger,
2019). Hence, by providing more complete earthquake sources for PSHA, active fault databases in
the EARS may play an important role in characterising its ever-increasing seismic risk (Goda et al.,
2016; Hodge et al., 2015).
Southern Malawi lies at the southern incipient end of the EARS where the rift floor is nearly entirely
onshore, follows regional Proterozoic fabrics, and is not buried by magmatism or significant
amounts of sediments (Wedmore et al., 2020a, 2020b; Williams et al., 2019). The SMAFD therefore
also provides constraints on faulting during the early stages of continental rifting and the influence
of pre-existing mechanical anisotropies in the crust on fault evolution.
This study first describes the seismotectonic setting of southern Malawi (Sect. 2), and the approach
used for mapping its active faults (Sect. 3). In Sect. 4 we then describe the method used to estimate
fault slip rates, earthquake magnitudes and recurrence intervals, using southern Malawi as an
example. Results from the SMAFD are documented in Sect. 5 along with an evaluation of fault slip
rate estimates and a sensitivity analysis. Finally, in Sect. 6, we discuss the implication of the
SMAFD in terms of fault growth in continental rifts, southern Malawi's seismic hazard, and the
strategies needed to reduce uncertainties when assessing this hazard.





## 2. Southern Malawi seismotectonics

*2.1 Tectonic history of southern Malawi*

Southern Malawi lies at a complex intersection of orogenic belts that formed during the Pan African

Orogeny (~800-450 Ma) and possibly earlier Irumide age deformation (~1,020-950 Ma) as the

African continent gradually amalgamated during the Proterozoic, and which imparted amphibolite-

granulite facies metamorphic fabrics (mineral segregations and alignments) within the rift's

basement rocks (Figs. 1 and 2; Andreoli, 1984; Fritz et al., 2013; Fullgraf et al., 2017; Kröner et al.,

2001; Manda et al., 2019). Within the Phanerozoic (540 Ma to present day), Permian-Triassic

sediments were deposited in the Lower Shire Graben under NW-SE Karoo extension (Fig. 1b;

Castaing, 1991; Habgood et al., 1973; Wedmore et al., 2020b). NE-SW striking dykes then formed

during the Jurassic, followed by minor accumulations of Cretaceous sediments under NE-SW

extension (Castaing, 1991). Evidence for Upper Jurassic to Cretaceous magmatism is also observed

across southern Malawi with the emplacement of the Chilwa Alkaline Province (Bloomfield, 1965;

Dulanya, 2017; Eby et al., 1995; Manda et al., 2019).

*2.2 Southern Malawi tectonic setting*

Southern Malawi lies towards the southern incipient end of the EARS Western Branch, where it

represents the divergent boundary between the Rovuma and Nubia plates (Fig. 1; Saria et al., 2013;

Stamps et al., 2008, 2018). The rift itself consists principally of three linked 50-150 km long grabens

and half grabens that follow regional Proterozoic fabrics and channel the Shire River, Lake

Malawi's only outlet, towards its confluence with the Zambezi River (Figs. 1 and 2; Chapola and

Kaphwiyo, 1992; Dulanya, 2017; Ebinger, 1989; Ebinger et al., 1987; Ivory et al., 2016; Wedmore

et al., 2020a; Williams et al., 2019). Like elsewhere in the Western Branch, each of these grabens is

defined by one or more border faults, whose footwall escarpments dominate the topographic

expression of the rift (Ebinger, 1989).






Border fault footwall escarpments in southern Malawi tend to be 300-1000 m high (Figs. 2 and 3;
Laõ-Dávila et al., 2015; Wedmore et al., 2020a, 2020b), and so are lower than in northern Malawi
(1000-2000 m; Accardo et al., 2018; Flannery and Rosendahl, 1990; Laõ-Dávila et al., 2015).
Furthermore, in central and northern Malawi, the rift is occupied by Lake Malawi and a <5 km thick
synrift sedimentary sequence (Accardo et al., 2018). Conversely, the floors of the grabens in
southern Malawi are subaerially exposed except for the 450 km2 flooded by Lake Malombe.
Boreholes and electrical resistivity surveys suggest that alluvial and colluvial sediments that cover
the basement in the Zomba and Makanjira grabens are <100 m thick (Fig. 2b; Bloomfield and
Garson, 1965; King and Dawson, 1976; Mynatt et al., 2017; Walshaw, 1965; Walter, 1972).
Cumulatively, these observations indicate that the rift in southern Malawi has accommodated less
extensional strain than further north. Therefore, although the age of EARS rifting in southern
Malawi is poorly constrained (Dulanya, 2017; Wedmore et al., 2020a), it is unlikely to be older than
the mid-Pliocene (~4.5 Ma) onset of sediment accumulation in Lake Malawi (Delvaux, 1995;
McCartney and Scholz, 2016), and almost certainly not older than the Oligocene (23-25 Ma) age of
the northern end of the Malawi Rift (Mesko, 2020; Mortimer et al., 2016; Roberts et al., 2012).
However, it is unclear if the rift in southern Malawi is actually younger than in northern Malawi,
and/or it is the same age but has been extending at a slower rate due to its proximity to the Nubia-
Rovuma Euler pole (Fig. 1a).

The floors of the Zomba and Makanjira graben sit at an altitude only ~10 m higher than Lake
Malawi. Hence, the sediments deposited in these grabens likely formed during base level changes in
lake level, when it was up to 150 m higher than present (Ivory et al., 2016; Lyons et al., 2015;
McCartney and Scholz, 2016), and would have flooded this section of the rift (Wedmore et al.,
2020a). Between the Zomba and Lower Shire grabens, the rift floor elevation drops from ~450 to





~100 m; however, there is no evidence that this is controlled by active faults (Dulanya, 2017;
Wedmore et al., 2020a). The Rungwe Volcanic Province, the closest EARS volcanism to southern
Malawi, is ~700 km to the north (Fig. 1a), and hot springs in southern Malawi do not indicate a
magmatic origin (Dulanya et al., 2010). Nevertheless, minor intrusions into the lower crust cannot be
excluded (Wang et al., 2019).

Prior to this study, the only systematic active fault mapping in southern Malawi was conducted by
Castaing (1991) and Chapola and Kaphwiyo (1992), whose maps were subsequently incorporated by
Macgregor (2015) into EARS scale maps, and later into the Global Earthquake Model (GEM)
Global Active Faults map (Fig. 2b; https://blogs.openquake.org/hazard/global-active-fault-viewer/,
date last accessed 4 June 2020). However, the faults are mapped at a coarse scale, and there is no
additional information such as slip rates or evidence for recent fault activity that are both vital
components of an active fault database.

*2.3 Southern Malawi seismicity*
There are no known historical accounts of surface rupturing earthquakes in southern Malawi,
although a continuous written record only extends to c. 1870 (Pike, 1965; Stahl, 2010). However, in
northern Malawi, the previously unrecognised St Mary fault exhibited surface rupture following the
2009 Karonga earthquake sequence (Fig. 1b; Hamiel et al., 2012; Kolawole et al., 2018b; Macheyeki
et al., 2015). This sequence primarily consisted of four shallow (focal depths <8 km) Mw 5.5-5.9
events over a 13 day period (Biggs et al., 2010; Gaherty et al., 2019). Another relevant event for
southern Malawi is the 1910 Ms 7.4 Rukwa Earthquake in southern Tanzania (Ambraseys, 1991).
The fault source for this event is not certain, though the Kanda fault is a likely candidate (Vittori et
al., 1997), and its steep and laterally continuous scarp closely resembles that of some faults in
southern Malawi (Hodge et al., 2018a, 2019, 2020; Wedmore et al., 2020a, 2020b).




The largest instrumentally recorded earthquake in southern Malawi is a M 6.7 event in 1954 (De
Bremaeker, 1956; Delvaux and Barth, 2010). The International Seismological Centre (ISC) record
for Malawi is complete for events with magnitude (Mw) > 4.5 to 1965 (Figs. 1b and 2a; Hodge et al.,
2015), with the largest event in this record being the 1989 Mw 6.3 Salima Earthquake (Jackson and
Blenkinsop, 1993). Notably, seismicity in Malawi is commonly observed to depths far greater (30-
35 km; Craig et al., 2011; Delvaux and Barth, 2010; Jackson and Blenkinsop, 1993) than would be
expected for continental crust of typical composition and geothermal gradient (10-15 km). Thick
cold anhydrous lower crust (Craig et al., 2011; Jackson and Blenkinsop, 1997; Nyblade and
Langston, 1995), localised weak viscous zones embedded within strong lower crust (Fagereng,
2013), and/or volumes of mafic material in the lower crust (Shudofsky et al., 1987) that are velocity
weakening at temperatures <700 °C (Hellebrekers et al., 2019) have been proposed as explanations
for this unusually deep seismicity.

*2.4 Estimates of stress and strain in southern Malawi*
Earthquake focal mechanism stress inversions for the entire Malawi Rift indicate a normal fault
stress state (i.e. vertical maximum principal compressive stress) with an ENE-WSW to E-W
trending minimum principal compressive stress ($\sigma_3$; Fig. 1b Delvaux and Barth, 2010; Ebinger et al.,
2019; Williams et al., 2019). This $\sigma_3$ orientation is comparable to the $\sigma_3$ direction inferred from
regional joint orientations (Williams et al., 2019), and the geodetically-derived extension direction
between the Nubia and Rovuma plates (Fig. 1b; Saria et al., 2014; Stamps et al., 2018). ENE-WSW
to E-W extension indicates that NE-SW striking faults in Malawi should accommodate oblique slip.
However, slickensides and earthquake focal mechanisms indicate approximately dip-slip motion
regardless of fault strike in southern Malawi (Fig. 1b; Delvaux and Barth, 2010; Hodge et al., 2015;
Wedmore et al., 2020a). This apparent inconsistency between faults that are simultaneously





accommodating near pure dip-slip and strike oblique to the regional extension direction can be
explained if the lower crust in southern Malawi contains lateral rheological heterogeneities such as
an anastomosing shear zone (Fagereng, 2013; Hodge et al., 2018a; Philippon et al., 2015; Wedmore
et al., 2020a; Williams et al., 2019).

*2.5 Seismic hazard assessment in southern Malawi*
Using instrumental catalogues, PSHA have been conducted for southern Malawi as part of EARS-
wide studies (Midzi et al., 1999; Poggi et al., 2017). These indicate that there is a 10% probability of
exceeding 0.15 g peak ground acceleration in southern Malawi in the next 50 years (Poggi et al.,
2017). However, the rift extension rate calculated from the seismic moment release rate in Malawi
(0.3 mm/yr; Hodge et al., 2015) is less than the geodetically estimated extension rate (~0.5-2 mm/yr;
Saria et al., 2013, 2014; Stamps et al., 2018). This implies that stress is accumulating in the crust that
has not been released in earthquake ruptures during the relatively short instrumental time (Ebinger et
al., 2019; Hodge et al., 2015). Thus, the geodetic and geomorphological information incorporated
into SMAFD may be a better guide to the magnitude and locations of future seismicity in southern
Malawi.

**3.  Mapping active faults in the SMAFD**
An active fault database consists of an active fault map, where for each fault, attributes are added
that detail geomorphic and geological information about the fault, and estimates of the parameters
required to incorporate them as earthquake sources in PSHA (Christophersen et al., 2015).
Typically, an active fault database is stored in a Geographic Information System (GIS) environment,
in which the fault attributes are assigned to a linear feature that represents the fault's geomorphic
trace (e.g. Langridge et al., 2016; Machette et al., 2004; Styron et al., 2020). In this section, we





describe the methodology for mapping active faults in southern Malawi and assigning some basic
geomorphological attributes. Note here that to keep the fault mapping complete for this EARS
section, some faults in the SMAFD also extend into Mozambique (Fig. 2). Estimates of associated
earthquake source parameters (including slip rate, earthquake magnitudes and recurrence intervals)
are described in Sect. 4.

*3.1 Identifying active and inactive faults in southern Malawi*
There are many inherent limitations in mapping active faults. Even in countries with well-developed
databases such as Italy and New Zealand, their success in accurately predicting the locations of
future surface rupturing earthquakes is, at best, mixed (Basili et al., 2008; Nicol et al., 2016a). An
active fault might not be recognised because evidence of previous surface rupture is subsequently
buried, eroded (Wallace, 1980), or the fault itself is blind (e.g. Quigley et al., 2012), which in turn
depends on earthquake magnitudes, thickness of the seismogenic crust, and the local geology.
Furthermore, although active and inactive faults are typically differentiated by the age of the most
recent earthquake, the precise maximum age that is used to define 'active' varies between different
active fault databases depending on the regional strain rate (i.e. plate boundary vs. stable craton) and
the prevalence of youthful sediments (Clark et al., 2012; Jomard et al., 2017; Langridge et al., 2016;
Machette et al., 2004). Indeed, in some cases it may not be possible to reliably determine if an
exposed fault has been recently 'active' or not (Cox et al., 2012; Nicol et al., 2016a).

Each of these issues has relevance to mapping active faults in southern Malawi. Firstly, they may be
buried by sediments during regular (10-100 ka) ~100 m scale fluctuations in the level of Lake
Malawi (Ivory et al., 2016; Lyons et al., 2015). Alternatively, the relatively thick (30-35 km)
seismogenic crust in southern Malawi means that even moderate-large earthquakes (Mw>6) do not
necessarily result in surface rupture, as illustrated by the Mw 6.3 Salima earthquake (Gupta, 1992;





Jackson and Blenkinsop, 1993). Finally, there is little chronostratigraphic control for this section of
the EARS (Dulanya, 2017; Wedmore et al., 2020a) to help differentiate between inactive and active
faults.

For the SMAFD, we therefore define active faults based on evidence of activity within the current
tectonic regime. Such an approach has been advocated elsewhere in the EARS (Delvaux et al., 2017)
and in other areas with low levels of seismicity, few paleoseismic studies, and/or where there are
faults that are favourably oriented for failure in the current stress regime, but which have no
definitive evidence of recent activity (Nicol et al., 2016a; De Pascale et al., 2017; Villamor et al.,
2018). In practice, this means that faults will be included in the SMAFD if they can be demonstrated
to have been active during East African rifting. This evidence can vary from the accumulation of
post Miocene hanging wall sediments to the presence of a steep fault scarp, offset alluvial fans,
and/or knickpoints in rivers that have migrated only a short vertical distance (<100 m) upstream
(Hodge et al., 2019, 2020; Jackson and Blenkinsop, 1997; Wedmore et al., 2020a). We note that the
absence of post-Miocene sediments in the hanging-wall of a normal fault does not necessarily imply
that it is inactive, if for example, faults are closely spaced across strike so that sediments are eroded
during subsequent footwall uplift of an interior normal fault (e.g. Chirobwe-Ncheu fault, Fig. 3c; see
also Mortimer et al., 2016; Muirhead et al., 2016). In these cases, if there is other evidence of recent
activity (e.g. scarp, triangular facets), these faults are still included. Previous active fault mapping in
southern Malawi was based on the extent of scarps alone (Hodge et al., 2019; Wedmore et al.,
2020a). Therefore, the relaxed definition of an 'active' fault in the SMAFD means that it includes
more faults than these maps, and that the lengths of some faults have been increased.

For the sake of completeness, major faults that control modern day topography, but that do not fit
the criteria of being active (e.g. Karoo faults), were mapped separately (Fig. 2b). However, this map



is not necessarily complete for all inactive faults in southern Malawi, and we also cannot definitively
exclude the possibility that some of these faults are still active although they display no evidence for
it. The relatively broad definition of an active fault may also mean that some inactive faults are
included in the SMAFD. However, in applying the opposite approach (i.e. requiring an absolute age
for the most recent activity on a fault) there is a greater risk that faults mistakenly interpreted to be
inactive subsequently rupture in a future earthquake (Litchfield et al., 2018; Nicol et al., 2016a).

*3.2 Datasets for mapping faults in southern Malawi*
*3.2.1 Legacy geological maps*
Between the 1950s and 1970s, the geology of southern Malawi was systematically mapped at
1:100,000 scale and these maps, and their associated reports, were consulted in detail when defining
and naming faults. These studies noted evidence of recent displacement on the Thyolo (Habgood et
al., 1973), Bilila-Mtakataka, Tsikulumowa (Walshaw, 1965), and Mankanjira faults (King and
Dawson, 1976). However, no attempt was made to systematically distinguish between active and
inactive faults. Furthermore, there is ambiguity in these studies with equivalent structures in the
Zomba Graben being variably described as 'terrace features' (Bloomfield, 1965), active fault scarps
(Dixey, 1926) and Late Jurassic-Early Cretaceous faults (Dixey, 1938).

*3.2.2 Geophysical datasets*
Regional-scale aeromagnetic data were acquired across Malawi in 2013 by the Geological Survey
Department of Malawi (Fig. 2c; Kolawole et al., 2018a; Laõ-Dávila et al., 2015). These surveys
were used to refine fault mapping in cases where features interpreted as faults in the aeromagnetic
survey extended beyond their surface expression. A revised fault map for the Lower Shire Graben
based on gravity surveys (Chisenga et al., 2018) was also consulted when compiling the SMAFD.





### 3.2.3 Digital Elevation Models

*3.2.3 Digital Elevation Models*
The topography of southern Malawi is primarily controlled by EARS faulting (Dulanya, 2017; Laõ-
Dávila et al., 2015; Wedmore et al., 2020a) except in the case of the Kirk Range (Fig. 2b), and
readily identifiable igneous intrusions and Karoo faults (Figs. 3c and 4b). To exploit this interaction
between topography and active faulting, TanDEM-X digital elevation models (DEMs) with a 12.5 m
horizontal resolution and an absolute vertical mean error of ± 0.2 m (Wessel et al., 2018) were
acquired for southern Malawi (Fig. 2b). This small error means that the TanDEM-X data performs
better at identifying the metre-scale scarps common in southern Malawi (Hodge et al., 2019;
Wedmore et al., 2020a) than the more widely-used but lower resolution Shuttle Radar Topography
Mission (SRTM) 30 m DEMs (Sandwell et al., 2011). Furthermore, TanDEM-X data can be used to
assess variations in along-strike scarp height (Hodge et al., 2018a, 2019; Wedmore et al., 2020a,
2020b) and assess the interactions between footwall uplift and fluvial incision (Fig. 4a; Wedmore et
al., 2020a). The Mwanza and Nsanje faults partly extended out of the region of TanDEM-X
coverage, and these sections were mapped using the SRTM 30 m resolution DEM (Fig. 2b).

*3.2.4 Fieldwork*
To corroborate evidence of recent faulting recognised in DEMs and geological reports, fieldwork
was conducted on several faults (Fig. 2b). This ranged from documenting features indicative of
recent displacement on the faults, such as scarps and triangular facets, to comprehensively sampling
the fault and surveying it with an Unmanned Aerial Vehicle (Fig. 3; see also: Hodge et al., 2018;
Wedmore et al., 2020a, 2020b; Williams et al., 2019).

*3.3 Strategy for mapping and describing active faults in the SMAFD*
Following the 'active' fault definition and synthesis of the datasets described above, faults in
southern Malawi are mapped following the approach outlined for the GEM neotectonics fault



database (Christophersen et al., 2015; Litchfield et al., 2013). This database uses a hierarchical
system to map faults, in which 'traces' are the basic unit, and one or more traces may be used to
define 'sections,' and one or more sections define 'faults' (Christophersen et al., 2015; Litchfield et
al., 2013). For faults in the SMAFD, which typically propagate to the surface, traces denote a linear,
relatively uniform active fault geomorphic expression. The end of a trace is defined by where the
geomorphic feature changes. For example, where a scarp may have been eroded to leave a gently
dipping escarpment.

'Sections' are portions of faults that have a distinct geometric, kinematic, or paleoseismic attribute
(Christophersen et al., 2015; Litchfield et al., 2013; Styron et al., 2020). Except in the case of linking
sections, they also represent distinct surface rupturing earthquake sources in PSHA and so should be
>5 km in length (Christophersen et al., 2015). Given the lack of paleoseismic information on active
faults in the SMAFD, sections are generally defined by geometrical boundaries such as bends or
step-overs (Fig. 2d; DuRoss et al., 2016; Jackson and White, 1989; Wesnousky, 2008; Zhang et al.,
1991). Along-strike minima in fault displacement (e.g. scarp or knickpoint height) may also be
indicative of segmentation (Willemse, 1997), but these do not always coincide with geometrical
complexities in southern Malawi (Fig. 4; Hodge et al., 2018a, 2019; Wedmore et al., 2020a, 2020b).
This may indicate that deeper structures, not visible in the surficial fault geometry, are also
influencing fault segmentation (Wedmore et al., 2020b). Therefore, where along-strike scrap height
measurements exist, these local minima are also used to define fault sections (Figs. 2d and 4).

'Faults' as defined by Christophersen et al. (2015) represent trace(s) and/or section(s) capable of
rupturing together in a single earthquake. Empirical observations and Coulomb stress modelling
suggests that normal fault earthquakes rarely rupture across steps whose width is >20% of the length





of the interacting sections (Biasi and Wesnousky, 2016; Hodge et al., 2018b), and we use this as a
criteria to assign whether two *en echelon* sections in the SMAFD are part of the same fault.

*3.4 Fault trace attributes*
The attributes added to each mapped fault in the SMAFD are modelled on the GEM neotectonics
fault database guidelines (Christophersen et al., 2015). These are listed and briefly described in
Table 1, along with the hierarchical level it is assigned (i.e. trace, section, or fault). The first set of
attributes is linked to information collected about each trace, and so relate to geomorphic
observations (Table 1). The attributes 'scale' and 'confidence' reflect that two distinct
considerations must be made when mapping a geomorphic feature as an active fault (Barrell, 2015;
Styron et al., 2020): (1) its prominence in the landscape, which is indicated by the scale at which a
fault is mapped, and (2) the confidence that the feature is an active fault, which indicated by a
qualitative score from 1 (high) to 4 (low, Table 1).

*3.5 Section and fault geometry attributes*
The second set of attributes describes fault geometry, and these are assigned to both individual
sections and whole faults (Table 1). Section length ($L_{sec}$) is defined as the straight-line distance
between its end points (Fig. 4b). This approach avoids the difficulty of measuring the length of
possibly fractal features, and accounts for the hypothesis that small-scale (<km scale) variations in
fault geometry in southern Malawi may represent only near-surface complexity (depths <5 km), and
that the faults are actually relatively planar at depth (Hodge et al., 2018a). However, it only provides
a minimum estimate of section length. For segmented faults, fault length ($L_{fault}$) is the sum of $L_{sec}$,
otherwise $L_{fault}$ is the distance between its tips (Fig. 4b).





In southern Malawi, fault dip is either unknown or uncertain, because fault planes are rarely
exposed, surface processes affect scarp angle (Hodge et al., 2020), and/or dip at depth is not
constrained. This difficulty in measuring fault dip is commonly encountered, and in these cases dip
is instead parametrised by using a range of reasonable values (Christophersen et al., 2015; Langridge
et al., 2016; Styron et al., 2020). We follow this approach by assigning minimum, intermediate, and
maximum dip values of 40°, 53°, and 65°, which encapsulates dip estimates from field data in
southern Malawi (Hodge et al., 2018a; Williams et al., 2019), and earthquake focal mechanisms
(Biggs et al., 2010; Ebinger et al., 2019), seismic reflection data (Mortimer et al., 2007; Wheeler and
Rosendahl, 1994), and aeromagnetic surveys (Kolawole et al., 2018a) elsewhere in Malawi.

In the GEM neotectonics database, fault width ($W$) is estimated by projecting by the difference in
lower and upper seismogenic depth into fault dip ($\delta$), with the assumption that faults are
equidimensional up to the point where $W$ is limited by the thickness of the seismogenic crust ($z$;
Christophersen et al., 2015):

$$W = \begin{cases} L_{fault}, & where\ L_{fault} \leq \dfrac{z}{\sin\delta}; \\ \dfrac{z}{\sin\delta}, & where\ L_{fault} > \dfrac{z}{\sin\delta} \end{cases}$$

(1)

In southern Malawi, both $z$ (30-35 km; Jackson and Blenkinsop, 1993; Craig et al., 2011), and $\delta$
(40°-65° as justified above) are poorly constrained, so a range of $W$ values must be considered.
Furthermore, ruptures not limited by $z$ are not necessarily equidimensional (Leonard, 2010;
Wesnousky, 2008). We therefore also consider an alternative approach where $W$ is estimated using
an empirical scaling relationship between fault length and $W$ (Leonard, 2010):



$$W = C_1 L_{fault}^{\beta}$$

(2)

where $L_{fault} > 5$km, and where $C_1$ and $\beta$ are empirically derived constants and equal 17.5 and 0.66
respectively for interplate dip-slip earthquakes (Leonard, 2010).

When these equations are applied to the mapped length of faults in southern Malawi (Figs. 2 and
5a), both estimate $W \sim 40$ km, for its longest faults ($L_{fault} > 50$ km, Fig. 5c). Hence, Eq. 2 is
consistent with observations of thick seismogneic crust in East Africa (Craig et al., 2011; Ebinger et
al., 2019; Jackson and Blenkinsop, 1993; Lavayssière et al., 2019; Nyblade and Langston, 1995).
However, for shorter faults ($L_{fault} = 5$-$50$ km), Eq. (2) estimates smaller values of $W$ relative to the
approach outlined in Eq. (1) (Fig. 5c). As noted above, this follows empirical observations that the
aspect ratio of dip-slip earthquakes will be $>1$ where $L_{fault} > 5$ km. In this context, Eq. (2) provides
more reasonable $W$ estimates for 5-50 km long faults in south Malawi than Eq. (1) and makes little
difference for longer faults; hence it is used to estimate $W$ in the SMAFD. Furthermore, along with
$W$, the Leonard (2010) regressions are used to estimate earthquake magnitudes and average
displacement in the SMAFD (Sect. 4.2), and so these parameters are all self-consistent.

**4. A systems-based approach to estimating earthquake source parameters: application to**

**the SMAFD**

In addition to the active fault map, the GEM neotectonics fault database requires estimates of fault
slip rates, and earthquake magnitude and recurrence intervals (Christophersen et al., 2015).
However, given the lack of chronostratigraphic control for faulted surfaces in southern Malawi, no
direct measurements of these attributes can be assigned to faults in the SMAFD. Indeed, as noted in
the introduction, obtaining these parameters is difficult and even regions with well-developed active



fault databases such as in California and New Zealand only have directly measured slip rates and
paleoseismic information for a small number of faults (Field et al., 2014; Langridge et al., 2016).

Instead, we suggest that fault slip rates can be estimated through a systems-level approach in which
geodetically derived plate motion rates are partitioned across faults in a manner consistent with their
geomorphology and regional tectonic regime. Although, such an approach has been used before over
small regions (Cox et al., 2012; Litchfield et al., 2014), it has not been applied to an entire plate
boundary. In addition, we also outline how the uncertainties and alternative hypotheses that are
inherent to this approach can, in common with seismic hazard practice elsewhere, be explored with a
logic tree approach (Fig. 6; Field et al., 2014; Vallage and Bollinger, 2019; Villamor et al., 2018).

The SMAFD is used as an example here of how this approach can be applied to a narrow amagmatic
continental rift, where the distribution of regional strain between border faults and intrabasinal faults
is well constrained by theoretical and observational studies (Agostini et al., 2011a; Corti, 2012;
Gupta et al., 1998; Morley, 1988; Muirhead et al., 2016, 2019; Nicol et al., 1997; Shillington et al.,
2020; Wedmore et al., 2020a; Wright et al., 2020). However, this framework could be adapted to
other tectonic regions with well mapped active faults, few on-fault slip rate measurements, and
where the partitioning of regional geodetic strain is, to an extent, predictable; for example fold and
thrust belts (Koyi et al., 2000; Poblet and Lisle, 2011) and strike-slip systems (Braun and Beaumont,

1995).


*4.1 Estimating fault slip rates*
For a narrow amagmatic continental rift, the first step is to divide the rift along its axis into each of
its graben/half grabens, and then within each graben/half graben, divide the mapped faults into



border and intrabasinal faults. Then, the slip rate for each fault or fault section *i* is estimated using
the equation:

$$
Slip\ rate\ (i) =
\begin{cases}
\dfrac{\alpha_{bf}\, v\, \cos(\theta(i) - \phi)}{n_{bf}\, \cos\delta}, & for\ border\ faults \\[2ex]
\dfrac{\alpha_{if}\, v\, \cos(\theta(i) - \phi)}{n_{if}\, \cos\delta}, & for\ intrabasinal\ faults
\end{cases}
$$

(3)

where $\theta(i)$ is the fault or fault section slip azimuth, $v$ and $\phi$ are the horizontal rift extension rate and
azimuth, $\alpha$ is a weighting applied to each fault depending on whether it is a border ($\alpha_{bf}$) or
intrabasinal ($\alpha_{if}$) fault, and it is divided by the number of mapped border faults ($n_{bf}$) or intrabasinal
faults ($n_{if}$) in each graben (Fig. 6). Though Eq. 3 is specific for rifts, it could be adapted in other
tectonic settings, for example to distribute regional strain between the basal detachment and thrust
ramps in a fold and thrust belt (Poblet and Lisle, 2011), between multiple subparallel faults in a
strike-slip system, or assess more complex strain partitioning between kinematically distinct fault
populations in transtensional or transpressional systems (Braun and Beaumont, 1995).

To estimate slip rates in the SMAFD we therefore first divide the rift into its principal grabens
(Makanjira, Zomba, and Lower Shire, Fig. 2a). In addition, we include the Nsanje fault, which is
located to the south of Malawi's principal EARS grabens (Fig. 2a) and where it bounds a poorly
defined section of the EARS with low footwall relief (~300 m) and no mapped intrabasinal faults.
There is, however, an eastern border fault to this section of the rift that has been mapped 25 km
along strike in Mozambique (Fig. A2; Macgregor, 2015), and we group these two faults together into
the same 'Nsanje' graben.





When considering how $v$ is distributed amongst border ($\alpha_{bf}$) and intrabasinal faults ($\alpha_{if}$) in an
amagmatic narrow rift, consideration should be given to factors such as total rift extension (Ebinger,
2005; Muirhead et al., 2016, 2019), rift obliquity (Agostini et al., 2011b), hanging-wall flexure
(Muirhead et al., 2016; Shillington et al., 2020), lower crustal rheology (Heimpel and Olson, 1996;
Wedmore et al., 2020a), and whether border faults have attained their maximum theoretical
displacement (Accardo et al., 2018; Olive et al., 2014; Scholz and Contreras, 1998). As an incipient
amagmatic rift, extensional strain in southern Malawi is expected to be localised (~80-90%) on its
border faults (Muirhead et al., 2019; Wright et al., 2020). Furthermore, the relatively small throws
on the border faults of southern Malawi (<1000 m) and thick seismogenic crust mean there that the
flexural extensional strain in the hanging wall (i.e. on the intrabasinal faults) is negligible (0.1-1.2%,
see Appendix A for full analysis; Billings and Kattenhorn, 2005; Muirhead et al., 2016). However,
detailed analysis of fault scarp heights across the Zomba Graben indicate that ~50% of extensional
strain is currently distributed onto its intrabasinal faults (Wedmore et al., 2020a). To recognise this
uncertainty in the SMAFD, lower, intermediate, and upper estimates of $\alpha_{bf}$ are set to 0.5, 0.7, and 0.9
respectively, with this uncertainty explored using a logic tree (Fig. 6). Since $\alpha_{if}$ is the 'remainder' of
the rift extension in each graben (i.e. $\alpha_{if} = 1- \alpha_{bf}$), it is set to 0.1, 0.3, and 0.5 for lower, intermediate,
and upper estimates (Fig. 6). For the Nsanje graben, where the rift consists of just two border faults,
each fault is assigned 50% of the regional geodetic extension rate.

In the SMAFD, $v$ is taken from the plate motion vector between the Rovuma and Nubia plates at the
centre of each individual graben (Table 2, Figs. 1b and A2) using the Euler poles reported in Saria et
al. (2013). We use the Euler pole (as defined by a location and rotation rate) and the uncertainties
associated with the Euler pole (defined by an error ellipse, Fig. B1) to calculate the plate motion and
the plate motion uncertainty between the Rovuma-Nubia plates for each graben (Table 2, Fig. 1b)
following the methods outlined in Robertson et al. (2016). With this approach, the lower bound of $v$





is negative (i.e. the plate motion is contractional, Table 2). However, the topography and seismicity
of southern Malawi clearly indicate it is not a contractional regime, nor is it a stable craton. A lower
bound of 0.2 mm/yr horizontal extension is therefore assigned in the SMAFD, which is considered
the minimum strain accrual that is measurable using geodesy (Calais et al., 2016).

Along with uncertainty in $v$, there is also considerable uncertainty in the rift extension azimuth ($\phi$) in
southern Malawi from geodesy (Table 2) due to the poorly constrained Euler pole (Saria et al.,
2013). Independent measurements of regional stress and strain in southern Malawi through focal
mechanism stress inversions, however, provide tighter constraints on $\phi$ (073°± 012°, Fig. 1b;
Delvaux and Barth, 2010; Ebinger et al., 2019; Williams et al., 2019), and so we instead incorporate
this additional prior knowledge into the SMAFD for all grabens.

As discussed in Sect. 2.3 earthquake focal mechanisms and fault slickensides in southern Malawi
indicate that faults accommodate normal dip-slip motion, regardless of strike; a phenomena that can
be explained by lateral heterogeneity in the lower crust (Corti et al., 2013; Philippon et al., 2015;
Wedmore et al., 2020a; Williams et al., 2019). Therefore, the slip azimuth ($\theta(i)$) is equivalent to the
dip direction of the fault or fault section (Fig. 6, Table 1). It is then necessary to project $\theta(i)$ into $\phi$ in
Eq. (3) as these parameters are not necessarily aligned. To account for the uncertainty in $\phi$, upper
and lower extension rates are obtained from varying $\phi$ by ±012° depending on the fault's dip
direction (e.g. upper slip rate estimates for NE and NW dipping fault are estimated with $\phi$ set to 061°
and 085° respectively, so that the difference between $\phi$ and $\theta$ tends towards 0° or 180°). In
converting extension rate to fault slip rate, $\delta$ is varied between 40-65° as discussed in Sect. 3.5 (Fig.
6). Finally, unlike in the GEM neotectonic fault database, only the dip-slip rate is reported in the
SMAFD as the assumption of normal faulting implies that this is equal to the net slip rate. An





example of these slip rate calculations for the central section of the Chingale Step fault is provided
in Fig. 7.

*4.2 Earthquake source attributes*
The next set of attributes in the GEM neotectonics fault database are related to a fault's earthquake
source attributes (i.e. earthquake magnitudes and recurrence intervals, *R*; Table 1). Although these
would ideally be assigned based on historical seismicity or paleoseismicity, where this information
is lacking, earthquake magnitudes can by estimated using empirically derived scaling relationships
between fault length and earthquake magnitude. Scaling relationships between fault length and
average single event displacement ($\overline{D}$) can then be combined with slip rate estimates to calculate *R*
through the relationship $R=\overline{D}$/slip rate (Cox et al., 2012; Stirling et al., 2012).

Potential errors exist in the datasets from which earthquake scaling relationships are derived,
because of: (1) the possible use of inaccurate historical datasets (Stirling et al., 2013), (2)
underestimates of rupture length caused by the low preservation potential of small displacement
rupture tips (Hemphill-Haley and Weldon, 1999), and (3) overestimates of $\overline{D}$ from the tendency for
paleoseismic investigations to target the largest scarps along a fault (DuRoss, 2008). Furthermore, in
the case of southern Malawi, relatively few events from regions with thick seismogenic crust are
included in these datasets, and earthquakes in such crust may follow difference scaling relationships
(Hodge et al., 2020; Rodgers and Little, 2006; Smekalin et al., 2010).

To select an appropriate set of earthquake scaling relationships for the SMAFD, we consider three
previously reported regressions, and apply them to its mapped faults: (1) between normal fault
length and moment magnitude (Mw; Wesnousky, 2008), (2) interplate dip-slip fault length and Mw
(Leonard, 2010), and (3) fault area and Mw (Wells and Coppersmith, 1994) where *A* is calculated





using $W$ derived from Eq. (1). Results are shown in Fig. 5d, which indicates that although generally
comparable, for Mw <7.5, the Wells and Coppersmith (1994) regression overestimates magnitudes
relative to Leonard (2010). This likely reflects the discrepancy in $W$ between applying Eq. (1) and
the Leonard (2010) regression (Eq. (2), Fig. 5c, Sect. 3.5). The Wesnousky (2008) regression
overestimates magnitudes for Mw <6.9 relative to Leonard (2010) equations and underestimates
them at larger magnitudes (Fig. 5d). This may reflect that the Wesnousky (2008) regression is
derived from only 6 events, and these events show a poor correlation between length and Mw
(Pearson's regression coefficient = 0.36). Given the above observations, the Leonard (2010)
regressions are applied to the SMAFD. Furthermore, these regressions are self-consistent when
estimating Mw and $\overline{D}$, which is not necessarily true for the other cases.

We recognise that segmented normal faults may rupture in both individual sections, as demonstrated
in Malawi by the Karonga earthquake sequence (Biggs et al., 2010; Fagereng, 2013), and whole
fault ruptures (DuRoss et al., 2016; Goda et al., 2018; Gómez-Vasconcelos et al., 2018; Hodge et al.,
2015; Iezzi et al., 2019; Valentini et al., 2020). Mw and $\overline{D}$ for each section (except linking sections)
or fault $i$ in the SMAFD are therefore estimated as:

$$M_W(i) = \begin{cases} \dfrac{\left(\dfrac{5}{2}\log L_{sec} + \dfrac{3}{2}\log C_1 + \log C_2\mu\right) - 9.09}{1.5}, & for\ individual\ section\ ruptures \\[4ex] \dfrac{\left(\dfrac{5}{2}\log L_{fault} + \dfrac{3}{2}\log C_1 + \log C_2\mu\right) - 9.09}{1.5}, & for\ whole\ fault\ ruptures \end{cases}$$

(4)

$$\log\overline{D}(i) = \begin{cases} \dfrac{5}{6}\log L_{sec} + \dfrac{1}{2}\log C_1 + \log C_2\mu, & for\ individual\ section\ ruptures \\[3ex] \dfrac{5}{6}\log L_{fault} + \dfrac{1}{2}\log C_1 + \log C_2\mu, & for\ whole\ fault\ ruptures \end{cases}$$

(5)



where $\mu$ is the crust's shear modulus (3.3x10$^{10}$ Pa), $C_1$ is the same empirically derived constant used
in Eq. (2), and $C_2$ is another constant derived by Leonard (2010). Both constants are varied between
the full range of values derived in a least square analysis (Leonard, 2010) to obtain, lower,
intermediate and upper estimates of Mw and $\overline{D}$ (Figs. 6 and 7). Following Eq. (5), lower,
intermediate, and upper estimates of each fault or section's recurrence intervals $R(i)$ can be
calculated through:

$$R(i) = \frac{\overline{D}(i)}{Slip\ rate(i)}$$

(6)

Where upper estimates of $R$ are calculated by dividing the upper estimate of $\overline{D}$ by the lowest
estimate of fault/section slip rate and vice versa (Fig. 6). An example of these earthquake source
calculations for the central section of the Chingale Step fault is provided in Fig. 7.

*4.3 Miscellaneous attributes*
For each fault, a data completeness score is given, where 1 is the highest and 4 is the lowest (Table
1). This score represents the data quality of the trace, fault geometry, and slip rate attributes
(Christophersen et al., 2015). In the SMAFD, the highest score is 2, given the uncertainty on fault
slip rates and dip. Following the GEM template, other information for each fault includes the date of
the most recent event, references for published fault mapping or derivation of fault attributes, the
date that the information was last updated, the compiler of the information, and free text details
recorded as 'Fault Notes' (Table 1).



**5.   Key features of the SMAFD**
*5.1 Fault geometry, slip rates and earthquake source attributes*
Below, we present the results of applying the framework described above to faults in southern
Malawi. By implementing a logic tree approach to assess uncertainty in the SMAFD, three values
(lower, intermediate, and upper) are derived for each calculated attribute (Table 1, Fig. 6), with the
range of values obtained by applying the lower and upper branches varying by up to three orders of
magnitude. However, by using a logic tree approach, it is implicit that these upper and lower values
have a low probability as they require a unique, and possibly unrealistic, combination of parameters.
We therefore primarily report values obtained from applying the intermediate branches in the logic
tree but discuss the uncertainties associated with our estimates in Sect. 5.3.

In total, the SMAFD contains 20 active faults, which comprise a total of 53 sections and 82 traces.
Section lengths ($L_{sec}$) ranges between 6-60 km, whilst fault lengths ($L_{fault}$) varies from 11 to 150 km
(Fig. 5a, Table 3). By applying Eq. (2), fault width ($W$) is typically <30 km but may exceed >40 km
for the longest faults in the SMAFD (Fig. 5b, Table 3). The highest slip rates are estimated to be on
the Thyolo and Zomba faults (intermediate estimates 0.6-0.8 mm/yr). On intrabasinal faults in the
SMAFD, intermediate slip rate estimates are 0.05-0.1 mm/yr (Fig. 8). Slip rates tend to be relatively
fast in the Makanjira Graben (Fig. 8c), as the extension rate is higher (Table 2), and its NNW-SSE
striking faults are more optimally oriented to the regional extension direction (Fig. 2). The difference
between upper and lower slip rate estimates in the SMAFD logic tree is two orders of magnitude;
~0.05-5 mm/yr for the border faults and ~0.005-0.5 mm/yr on the intrabasinal faults (Fig. 8).

For whole fault ruptures along border faults, intermediate estimate of earthquake recurrence
intervals ($R$) are between 2000-5000 years and 10,000-30,000 years for intrabasinal whole fault
ruptures (Fig. 9a-c). Considerably uncertainty exists with these values, with the upper and lower





estimates for $R$ varying from $10_2$-$10_5$ years and ~$10_3$-$10_6$ years for border and intrabasinal whole
fault ruptures respectively (Fig. 9a-c). Furthermore, if these faults rupture in individual sections, $R$
may be reduced by up to an order of magnitude (Fig. 9d-f). Intermediate estimates of earthquake
magnitudes range from Mw 5.4 to Mw 7.2 for individual section ruptures, and Mw 6.0 to Mw 7.8 for
faults that rupture their entire length (Table 3, Fig. 10b). Notably, we document 12 faults with the
potential for hosting earthquakes greater than the largest recorded event in southern Malawi (i.e.
Mw> 6.7, Fig. 10b, assuming intermediate branches for scaling laws in Fig. 6), the largest of which
would be a Mw 7.8 ± 0.5 complete rupture of the Bilila-Mtakataka or Mwanza faults.

*5.2 Robustness of fault slip rate estimates*
The key advantage of the SMAFD in comparison to other fault maps made for the EARS (Chapola
and Kaphwiyo, 1992; Delvaux et al., 2017; Macgregor, 2015) is that it provide slip rates estimates
for all individual faults and fault sections (Fig. 8). It is, however, possible that some proportion of
the geodetically derived rift extension may be accommodated by aseismic creep or on hitherto
unrecognised faults, in which case the SMAFD estimates are effectively upper bounds. With regards
to aseismic creep, the discrepancy between geodetic and seismic moment rates, and the low b-value
(~0.8) for seismicity in the Karonga region implies that faults in Malawi are strongly coupled
(Ebinger et al., 2019; Hodge et al., 2015). This is further supported by the velocity-weakening
behaviour of samples from the rift in deformation experiments at lower crustal pressure-temperature
conditions (Hellebrekers et al., 2019). We cannot definitively account for blind faults, and we
recommend that future PSHA in southern Malawi should still consider 'off-fault' distributed seismic
sources by using the instrumental record (e.g. Field et al., 2014; Hodge et al., 2015; Stirling et al.,

2012).






Conversely, the possible inclusion of inactive faults in the SMAFD would mean its slip rates
estimates are lower bounds. Without paleoseismic investigations and dating of faulted surfaces in
southern Malawi, it is difficult to test this point. Nevertheless, reactivation analysis that encompasses
the range of fault orientations in southern Malawi indicates that these faults are favourably oriented
in the current stress field (Williams et al., 2019). Therefore, even faults that have been inactive for a
considerable time (up to the entire age of the EARS) could still theoretically be reactivated.

An additional test for the slip rate estimates in the SMAFD is provided by comparisons to slip rates
for intrabasinal faults in the North Basin of Lake Malawi. Here, Shillington et al. (2020) estimated
slip rates of 0.15-0.7 mm/yr based on the 10-40 m vertical offset of a 75 ka horizon in seismic
reflection data, and assuming fault dips of between 50-65°. These rates are consistent with the
SMAFD only if the upper estimate branches for intrabasinal fault slip rates in the logic tree are used
(Fig. 8). Alternatively, high slip rates on intrabasinal faults in northern Malawi may reflect that this
section of the EARS is extending more quickly (1-3 mm/yr) as it is further from the Nubia-Rovuma
Euler pole (Fig. 1a; Saria et al., 2013; Stamps et al., 2018), and/or that intrabasinal faults in southern
Malawi accommodate significantly less hanging-wall flexure (0.1-1.2% vs. 2.5-7%, Appendix A;
Shillington et al., 2020). In this context, the 0.05-0.1 mm/yr intermediate slip rate estimates for
intrabasinal faults in the SMAFD may be consistent with these estimates in northern Malawi.

Given intermediate slip rate estimates of 0.6-0.8 mm/yr (Fig. 8) and fault dips of 53º, the throw
accumulated by the border faults in the Makanjira and Zomba grabens (~350-900 m, Table A1)
would have accumulated in ~0.5-1 Ma. This is younger than the estimated age for EARS rifting in
central and northern Malawi (4.5-25 Ma; Delvaux, 1995; McCartney and Scholz, 2016; Mesko,
2020; Mortimer et al., 2016; Roberts et al., 2012); however, it is unclear if this indicates that the
lower border fault slip rate estimates (~0.05 mm/yr) in the SMAFD should be favoured, the onset of





rifting occurred later in southern Malawi, or there are additional factors that have not been
considered in this comparison (e.g. temporal variations in rift extension rate, footwall erosion). In
either case, the range of border fault slip rate estimates in the SMAFD appears broadly consistent
with age constraints for EARS rifting in southern Malawi.

*5.3 Sensitivity analysis*
Upper and lower estimates of $R$ differ by up to three orders of magnitude in the SMAFD (Fig. 9). To
investigate these uncertainties, we performed a multi-parameter sensitivity analysis following the
methods presented in Box et al. (1978) and Rabinowitz and Steinberg (1991). Full details of this
analysis are given in Appendix B. However, in summary, 7 parameters that contribute to uncertainty
in $R$ for the central section of the Chingale Step fault are considered (Table 4). By exploring all
possible combinations in which these 7 parameters are set at their upper or lower estimates, 128 (i.e.
$2^7$) different values of $R$ can be calculated. However, by using a fractional factorial design (Box et
al., 1978), we instead considered 64 carefully selected parameter combinations at little cost to the
analysis (Table B1). From these combinations, the natural log of the average value of $R$ when a
parameter ($k$) is set at its upper ($\overline{lnR}(k+)$) and lower ($\overline{lnR}(k-)$) value is calculated and the difference
between these values defines the parameter effect ($A$; Rabinowitz and Steinberg, 1991):

$$A = \overline{lnR}(k+) - \overline{lnR}(k-)$$

(7)

This analysis indicates that $R$ is most sensitive to uncertainties in the partitioning of strain between
border and intrabasinal faults in the rift (i.e. $\alpha_{if}/n_{if}$), the rift extension rate ($v$), and the $C_2$ parameter
in Eq. (5), and least sensitive to uncertainties in the rift's extension azimuth, and the $C_1$ parameter in
Eq. (5) (Table 4). If, however, $v$ and its associated uncertainties were estimated using a different
Nubia-Rovuma Euler pole solution (Fig. B1, Table 2; Stamps et al., 2008), $R$ estimates are least





sensitive to $v$ and most sensitive to $C_2$ (Table B2). Finally, we note that there is no interaction effect
between two separate parameters that may influence their sensitivity on $R$ (Table B3). These results
are discussed further in Sect. 6.3

**6.  Discussion**
In the following section, we examine some key results of the SMAFD in terms of its contribution to
our understanding of fault growth in continental rifts, its implications for seismic hazard in southern
Malawi, and future strategies to reduce its uncertainties and apply this framework to other regions.

*6.1 Controls on fault growth in southern Malawi*
As discussed in Sect. 2.2, the height of footwall escarpments and thickness of hanging-wall
sediments indicates that the rift in southern Malawi has accommodated less extension than further
north in Malawi, and indeed elsewhere along the EARS (Ebinger, 1989; Muirhead et al., 2019).
Nevertheless, the lengths of faults in southern Malawi (~10-150 km, Fig. 5a) are similar to lengths in
more evolved sections of the EARS (Agostini et al., 2011b; Ebinger, 1989; Macgregor, 2015;
Muirhead et al., 2019; Shillington et al., 2020). This suggests that faults in southern Malawi may
have relatively low total displacement to length ratios. If true, this reflects the 'constant length' fault
growth model (Walsh et al., 2002), with fault tip propagation potentially facilitated by exploitation
of pre-existing Proterozoic fabrics that the faults follow (Fig. 2c), and which are favourably -but not
optimally- oriented to the regional stresses (Williams et al., 2019).

The length scale of faults in southern Malawi also reflects its abnormally thick (30-35 km)
seismogenic crust (Jackson and Blenkinsop, 1997). In particular, we document continuous 30-60 km
long fault sections (Fig. 2d and 5a), whereas in typical continental crust with a 10-15 km
seismogenic thickness, the length of continuous normal fault sections is typically <25 km (Jackson





and White, 1989). Finally, we note that though the lower crust in southern Malawi may be laterally
heterogenous with localised zones of viscously deforming material (Fagereng, 2013; Hellebrekers et
al., 2019; Wedmore et al., 2020a), there is geological evidence from exhumed metamorphic terranes
that in dry lower crust, earthquakes may both nucleate and propagate within a predominantly viscous
regime (Campbell et al., 2020; Menegon et al., 2017).

*6.2 Implications for seismic hazard in southern Malawi*
The existence of active faults within southern Malawi poses a significant risk to the 7.75 million
people living in this region (Malawi National Statistics Office, 2018), and adjacent to the rift in
northern Mozambique (Fig. 10a). Furthermore, with population growth at an annual rate of 2.7% in
southern Malawi (Malawi National Statistics Office, 2018) this risk will increase over the coming
decades. The rapidly growing city of Blantyre (population 800,000; Malawi National Statistics
Office, 2018), which is in the footwall of both the relatively fast slipping (intermediate estimates
~0.8 mm/yr) Zomba and Thyolo faults is at a particularly large risk (Fig. 10a). There is therefore an
urgent need to quantify the spatial and temporal distribution of this hazard through a PSHA that
incorporates the earthquake source data collected in the SMAFD.

Out of a global dataset of 61 historical surface rupturing continental normal fault earthquakes, only
six events had a rupture length >50 km, and only one event (the 1887 Mw 7.5 Sonora earthquake)
has a length >100 km (Valentini et al., 2020). Hence, the faults compiled within the SMAFD have
the potential to produce the largest continental normal fault earthquake globally. However, low
regional extension rates imply such events are likely to be very rare, with intermediate estimates of
recurrence interval of $10_3$-$10_4$ years (Figs. 9 and 10c).





*6.3 Reducing uncertainties*
*6.3.1. Improving fault slip rate estimates*
As noted in the introduction, one of the purposes of collating the SMAFD was to identify current
knowledge gaps in our understanding of active faulting and seismic hazard in southern Malawi.
Given the various aleatory (i.e. the uncertainty related to unpredictable nature of future event) and
epistemic (i.e. the uncertainty due to incomplete knowledge and data) uncertainties in parameters
used to derive earthquake recurrence intervals ($R$), lower and upper estimates differ by over three
orders of magnitude (Fig. 9). Although such a range of estimates in a low strain rate region with
limited paleoseismic information is common (e.g. Cox et al., 2012; Villamor et al., 2018) and can
still be incorporated into PSHA using synthetic seismicity catalogues (Hodge et al., 2015), reducing
uncertainties in these estimates in the SMAFD is an obvious priority.

Our sensitivity analysis (Sect. 5.3) indicates that the two biggest factors contributing to uncertainty
in $R$ in the SMAFD is related to our understanding of the distribution and rate of extension ($v$) in
southern Malawi (Table 4). In particular, we note there is considerable uncertainty in the position of
the Nubia-Rovuma Euler pole (Fig. B1; Saria et al., 2013), and we would not expect such large
differences between upper and lower fault slip rate estimates elsewhere. Although the uncertainties
associated with $v$ in the SMAFD could be reduced if an alternative solution for the Nubia-Rovuma
Euler pole was applied (Fig. B1, Tables 4 and B2; Stamps et al., 2008), this solution uses fewer
Global Positioning System (GPS) sites and a shorter position time series (Saria et al., 2013).
Furthermore, the Stamps et al. (2008) solution implies extensional rates in southern Malawi of 2.5-3
mm/yr (Table 2), which exceeds even the upper bound of those from Saria et al. (2013) model and
also more recent observations of individual GPS stations in southern Malawi (1-2 mm/yr; Saria et
al., 2014; Stamps et al., 2018). Therefore, in the short-term, the best refinements to $R$ estimates may





come from new regional geodetic data and further high resolution topographic analysis (e.g.
Wedmore et al., 2020a).

An alternative approach to constrain $R$ estimates would be to obtain on-fault slip rates and
paleoseismic information in southern Malawi. However, as noted previously, this information is
difficult to collect, and currently very few records exist across the entire EARS (Delvaux et al.,
2017; Zielke and Strecker, 2009). Paleoseismic investigations would be particularly challenging in
southern Malawi due to the potential for large (~10 m) single event displacements (Hodge et al.,
2020), and that these investigations carry significant aleatory variability in low strain rate regions
like southern Malawi if only a few earthquakes are sampled (Nicol et al., 2006, 2016b). This latter
point reflects the fact that earthquakes may be temporally clustered in low strain rate regions
(Pérouse and Wernicke, 2017; Taylor-Silva et al., 2019) due to elastic stress perturbations (Beanland
and Berryman, 1989; Cowie et al., 2012; Harris, 1998; Wedmore et al., 2017); the possibility of
these perturbations influencing seismicity in Malawi has already been demonstrated by the 2009
Karonga earthquake sequence (Biggs et al., 2010; Fagereng, 2013; Gaherty et al., 2019).

*6.3.2. Constraining earthquake magnitudes and fault rupture scenarios*
When considering how different rupture magnitude estimates in the SMAFD influence $R$, the main
source of uncertainty is the $C_2$ parameter from the Leonard (2010) regressions (Table 4). This factor
controls the amount of displacement for a given rupture area (Leonard, 2010). It is therefore likely
related to stress drops, and uncertainty in $C_2$ in southern Malawi will only be reduced by recording
more events in similar tectonic environments (i.e. normal fault earthquakes, ideally in regions with
low (~1-10 mm/yr) extension rates and thick (20-35 km) seismogenic crust).





Reduced uncertainty in $R$ estimates can also come from a more thorough investigation of the types
(i.e. lengths) and probabilities of different rupture scenarios in the SMAFD. Notably, only end
member scenarios are currently accounted for, as multi-segment ruptures that do not rupture the
entire fault are not currently considered in the SMAFD. By defining faults to consist of sections
capable of rupturing together in a single maximum magnitude earthquake (Christophersen et al.,
2015), the rupture of multiple 'faults' is also not included. However, given events such as the 2010
El Mayor-Cucapah (Fletcher et al., 2014) and 2016 Kaikōura earthquakes (Litchfield et al., 2018) in
which the rupture 'jumped' unusually large distances (>5 km), the possibility of multi-fault
earthquakes in southern Malawi should not be ruled out.

Accounting for the relative probabilities of single section, multi-section, or whole fault ruptures in
southern Malawi could be achieved by considering the static stress changes associated with different
rupture scenarios (Parsons et al., 2012) or by generating synthetic seismic catalogues in which the
relative frequency of different ruptures is fixed in a way so that the resulting magnitude-frequency
distribution matches that of the instrumental earthquake catalogue (Chartier et al., 2017).
Alternatively, synthetic seismicity could be generated using physics-based models (Marzocchi et al.,
2009; Marzocchi and Melini, 2014; Robinson et al., 2011), which will allow a better evaluation of
any earthquake clustering. Using this approach, certain branches in the logic tree used to calculate $R$
(Fig. 6) could be weighted to penalise unlikely rupture scenarios. Finally, it could be recognised that
the lower, intermediate, and upper estimates of $R$ obtained using the SMAFD logic tree (Fig. 6)
would more appropriately represented by a probability density function (e.g. Weibull, Brownian
Passage Time; Pace et al., 2016), with these distributions subsequently applied when selecting $R$ in a
synthetic seismicity catalogue.



*6.4 Development of new active fault databases in other tectonic settings*
The SMAFD provides a framework for developing active fault databases within other narrow
amagmatic continental rifts (e.g. Baikal Rift, Rhine Graben, Shanxi Graben). An obvious target is
the extension of the SMAFD to central and northern Malawi. Here, faults under Lake Malawi, have
been mapped using seismic reflection data (Flannery and Rosendahl, 1990; McCartney and Scholz,
2016; Scholz, 1989; Shillington et al., 2016, 2020), with Quaternary activity demonstrated on them
by their offset of a 75 Ka horizon (Shillington et al., 2020). In addition, by combining DEMs,
fieldwork, and aeromagnetic and electrical resistivity data, several onshore active faults have been
documented in the region struck by the Karonga earthquake sequence (Kolawole et al., 2018b,
2018a; Macheyeki et al., 2015).

The SMAFD framework could also be applied more widely to other types of continental rifts,
however, further adaptions would be required to account for blind faults in rifts with thick hanging-
wall sediments, and where some component of the geodetically measured strain may be
accommodated by magmatism (Bull et al., 2003; Casey et al., 2006; Ebinger, 2005; Keir et al.,
2006). This framework could also be adapted for other tectonic settings with active fault maps, an *a*
*priori* understanding of the rate and distribution of regional strain, but with few on-fault slip rate
measurements; for example the Zagros fold and thrust belt (Alipoor et al., 2012; Authemayou et al.,
2006; Molinaro et al., 2005) and Taiwan orogenic belt (Mouthereau et al., 2009; Shyu et al., 2016).

**7. Conclusions**
Here, we describe a new systems-based approach that combines geologic and geodetic data to
estimate fault slip rates and earthquake recurrence intervals. This is then applied to faults in southern
Malawi, which has led to the development of the South Malawi Active Fault Database (SMAFD), a
geospatial database designed to direct future research and aid seismic hazard assessment and





planning. The SMAFD reveals that active faults with the potential for Mw >7 earthquakes exist
across southern Malawi. That earthquakes of such magnitude can occur within this incipient section
of the East African Rift System (EARS) reflects a combination of thick (30-35 km) seismogenic
crust and fault lengthening that may have been facilitated by the exploitation of favourably oriented
pre-existing crustal weaknesses.

Slow geodetically-derived extension rates (~1 mm/yr) imply that the faults themselves have low slip
rates (0.001-5 mm/yr), and so the recurrence intervals of Mw >7 events are estimated to be on the
order of $10_2$-$10_6$ years. The large range of these estimated recurrence times reflects aleatory
uncertainty on fault rupture scenarios and epistemic uncertainties in fault-scaling relationships, fault
slip rates, and fault geometry. Sensitivity analysis suggests the biggest reduction in uncertainties
would come from improved knowledge of fault slip rates through paleoseismic investigations or
geodetic studies. Nevertheless, the combination of long, highly-coupled, low slip rate faults and a
short (<65 years) instrumental record imply that the SMAFD is an important source of information
for future seismic hazard assessments within the rift. In this respect, the development of SMAFD is
timely as the seismic risk of southern Malawi is growing due to rapid population growth.
urbanisation, and seismically vulnerable building stock. Similar challenges exist elsewhere along the
EARS, which may also be partially addressed by following the framework provided by SMAFD.

**Appendices**
**Appendix A: Hanging-wall flexure in southern Malawi**
The considerable amounts of throw (>1000 m) along a rift bounding fault can induce a significant
amount of flexure within the lithosphere either side of the fault (Muirhead et al., 2016; Olive et al.,
2014; Petit and Ebinger, 2000; Shillington et al., 2020). In the case of the hanging-wall, this is a



downward flexure that can result in intrabasinal faults accommodating additional slip to that
imparted by regional extension alone (Muirhead et al., 2016). This additional flexural strain must
therefore be accounted for when considering the distribution of strain in southern Malawi.

Here, strain due to hanging-wall flexure is estimated in profiles across southern Malawi using the
methodology described by Muirhead et al. (2016), which is based on the equations presented in
Turcotte and Schubert (1982) and Billings and Kattenhorn (2005). These flexural profiles are also
compared to those made for the North Basin of Lake Malawi using the same method (Shillington et
al., 2020). This method calculates flexure by considering a vertical line-load at the point of
maximum deflection (i.e. at the upper contact of the border fault hanging wall, Fig. A1). The
deflection (ω) across a border fault hanging wall can then be estimated as:

$$\omega = \omega_0 e^{\frac{-x}{\alpha}} \cos\left(\frac{x}{\alpha}\right)$$

(A1)

where $\omega_0$ is the maximum deflection, $x$ is the position along a hanging wall profile from the
deflecting fault (Fig. A1), and $\alpha$ is:

$$\alpha = \left[\frac{Eh^3}{(3\rho_0 g(1-v^2))}\right]^{\frac{1}{4}}$$

(A2)

where $E$ is Young's Modulus, $v$ is Poisson's ratio (0.25), $g$ is acceleration due to gravity (9.8 m/s2), $h$
is the thickness of elastic crust, which is assumed here to be the equivalent to the thickness of the
seismogenic crust (30-35 km, Fig. A1; Jackson and Blenkinsop, 1993; Craig et al., 2011; Ebinger et
al., 2019), and $\rho_0$ is crustal density, for which the average crustal density (2816 kg/m3) for the
Malawi Rift from a three layer model is used (Fagereng, 2013; Nyblade and Langston, 1995). In this





analysis, a value of *E*, such that the hanging wall deflection is restricted to a distance comparable to
the actual width of the half-graben is used (Muirhead et al., 2016; Shillington et al., 2020). Using
this principle, a comparatively low value of *E* (3 GPa) is required to fit the flexure profiles across
southern Malawi's ~50 km wide half-grabens, although this is comparable to *E* used during similar
analysis elsewhere in the East African Rift System (EARS; Muirhead et al., 2016; Shillington et al.,

2020).


In Eq. A1, $\omega_0$ can be derived through the observation from real and modelled normal faults that the
ratio (*r*) of upthrow to downthrow along a normal fault is typically 0.2 (Muirhead et al., 2016).
Therefore:

$$\omega_0 = BF_{throw}(1 - r)$$

(A3)

Where $BF_{throw}$ is border fault throw and is equivalent to the sum of the footwall escarpment height
and hanging wall sediment thickness. There are significant uncertainties in estimating sediment
thickness within southern Malawi, hence a range of values are used (Table A1). Uncertainty is
highest in the Lower Shire Graben where no boreholes have penetrated basement (Habgood et al.,
1973), and where the contribution of Karoo rifting to hanging wall flexure of the Thyolo Fault also
needs to be considered (Castaing, 1991; Chisenga et al., 2018). Castaing (1991) report throws of
1000 m for other Karoo faults in the Lower Shire Graben, and that the Thyolo Fault would have
been in transtension during the main Permian to Lower Jurassic period of Karoo extension.
Furthermore, given the overall southward propagation of the EARS (Ebinger, 1989), it is unlikely
that it would have accommodated more throw than the border faults in the Zomba and Makanjira
grabens (~1000 m, Table A1) during this phase of rifting. Therefore, it is unlikely that total throw





along the Thyolo Fault exceeds 2000 m. As a full graben, we consider the hanging-wall flexure
across both sides of the Makanjira Graben (Fig. A2).

Given a profile of hanging wall deflection, it is possible to derive the resulting flexural extensional
strain (ε) within a half-graben (Billings and Kattenhorn, 2005; Muirhead et al., 2016):


$$\varepsilon = -y\left(\frac{d^2\omega}{dx^2}\right)$$

(A4)

where *y* is the vertical distance from the centre of the plate (downward is positive, Fig. A1). The
Zomba and Lower Shire grabens are ~50 km wide (Fig. A2), therefore the mean flexural strain over
this distance is reported. For the Makanjira graben, we calculate the mean strain from the
contribution of each side of the graben over its 75 km width. From these values, the magnitude of
flexural horizontal extension over each graben is calculated, as is the extension rate (both rift wide,
and per fault average) assuming a range of graben ages (McCartney and Scholz, 2016; Roberts et al.,
2012). In calculating flexural extension rates for the Lower Shire Graben, we assume that 50% of the
flexure in this graben is a result of Karoo rifting, and so calculate rates of EARS flexure based on
half of the strain values reported in Table A1.

Results of this analysis are shown in Fig. A3 and Table A1. These demonstrate that regardless of the
simplifications, uncertainties and assumptions in this analysis, hanging-wall flexure in southern
Malawi is negligible (strains <1%, slip rates due to hanging-wall flexure <0.03 mm/yr per fault).
This reflects the thick seismogenic crust in southern Malawi (e.g. Craig et al., 2011) and relatively
small amounts of throw across its border faults (<2000 m). For example, when compared to border
faults in northern Malawi (throw ~7000 m; Accardo et al., 2018), the magnitude of hanging-wall
flexure is considerably larger (strains 2-7%, Table A1, Shillington et al., 2020) . We therefore do not





consider hanging-wall flexure further when considering the slip rate of intrabasinal faults in southern
Malawi (Sect. 4.1, main text).

**Appendix B: A multiparameter sensitivity analysis for recurrence interval estimates in the**
**South Malawi Active Fault Database**

Recurrence interval estimates in the South Malawi Active Fault Database (SMAFD) vary by over
three orders of magnitude (Fig. 9). These uncertainties are not unexpected in a region like Malawi
with no paleoseismic data and an incomplete instrumental seismic record (Cox et al., 2012; Villamor
et al., 2018), and can be accounted for in Probabilistic Seismic Hazard Assessment (PSHA) using
synthetic seismicity catalogues (Hodge et al., 2015). Nevertheless, by conducting a sensitivity
analysis on the logic tree approach used to calculate these recurrence intervals (Fig. 6), it is possible
to determine which parameters contribute most to this uncertainty, and therefore guide future
research directions that will help constrain them in future iterations of the SMAFD. This analysis is
briefly described in the main text (Sect. 5.3, Table 4), and is documented fully below.

Here, we follow the multiparameter sensitivity analysis presented by Rabinowitz and Steinberg
(1991). This study conducted sensitivity analysis for the parameters that feed into PSHA, where the
output metric is the probability of exceedance of a given level of ground shaken. For the SMAFD,
we adapt this method to test the sensitivity of seven parameters that are used to calculate earthquake
recurrence intervals ($R$, Eq. B1, Table 4). This metric is chosen as it fully incorporates the aleatory
uncertainties in rupture length, and epistemic uncertainties in fault slip rates and the Leonard (2010)
scaling relationships (Fig. 6). This analysis is performed for the Chingale Step fault central section
(Fig. 4), where like all intrabasinal faults in the SMAFD, $R$ is calculated by:

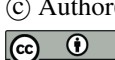



$$R = \frac{\left(\frac{5}{6}logL + \frac{1}{2}logC_1 + logC_2\right)\left(n_{if}cos\delta\right)}{\alpha_{if}vcos(\theta - \phi)}$$

(B1)

Where $L$ is rupture length and depends on whether an individual section ($L_{sec}$) or whole fault ($L_{fault}$)
rupture is considered, $C_1$ and $C_2$ are empirically derived constants from Leonard (2010), $\delta$ is fault
dip, $\theta$ is the fault slip azimuth, $v$ and $\phi$ are the rift extension rate and azimuth, $\alpha_{if}$ is a weighting of
rift extension for intrabasinal faults, and $n_{if}$ is the number of mapped intrabasinal faults ($n_{if}$) in the
graben.

Eq. B1 is essentially a combination of Eqs. 3, 5, and 6 in the main text, and its application with the
SMAFD logic tree to calculate $R$ for the Chingale Step fault central section is shown in Fig. 7. There
are 5 intrabasinal faults in the Zomba Graben where the Chingale Step fault is situated (Fig. 2), and
in this analysis, this parameter is not treated as an uncertainty. However, for simplicity, it is
combined with $\alpha_{if}$ to give the 'component of rift extensional strain' parameter, which is defined by
$\alpha_{if}/n_{if}$ (Table 4). Assuming that the Chingale Step fault is a normal fault (Wedmore et al., 2020a;
Williams et al., 2019), $\theta$ is the fault dip direction, and differs by only 4° depending on whether the
whole fault ruptures or just the central section (Fig. 7). Hence uncertainity in this parameter is not
considered here, and it is set at 290°, which is the average value for these two rupture scenarios.
When assessing the influence of $v$, we consider two geodetic models (Fig. B1; Saria et al., 2013;
Stamps et al., 2008), and perform this sensitivity analysis for both.

The method presented by Rabinowitz and Steinberg (1991) involves a two-level fractional factorial
multiparameter design, where each parameter is restricted to the two levels which will give lower or
upper estimates of $R$ (Table 4). Ideally, these levels would be symmetric about the intermediate case,
however, in the SMAFD this is not possible for the $v$, $L$, and $C_2$. Compared to a 'one at-a-time



(OAT)' parameter analysis, a multiparameter analysis allows us to assess how different parameters
interact with each other, and so more fully explore the parameter space (Rabinowitz and Steinberg,
1991). This is achieved through a factorial design, which for the seven parameters ($k$) tested here
would generate 128 (i.e. $2^7$) possible combinations in a full two-level factorial approach. However,
in a fractional factorial design, just a subset of these combinations is assessed. This approach
recognises that many of the combinations in a full factorial design offer little insight into how a
system works, and that this can instead be achieved at minimal cost to the results by considering a
carefully selected subset of these combinations (Box et al., 1978; Rabinowitz and Steinberg, 1991).
In this analysis, $2^{k-p}$ combinations are assessed where $p$ is the number of generators and is set at 1.
This results in the assessment of 64 combinations (Table B1) and a 'resolution' of 5, which means it
is possible to estimate the main effects of each parameter (Eq. B2), interactions between two
parameters (Eq. B3), but not interactions between three parameters (Box et al., 1978).

The main effect ($A$) of one parameter (e.g. fault dip, $\delta$) is quantified from the difference between the
average of the natural log of recurrence interval ($\overline{\ln R}$) for the 32 combinations in Table B1 when a
parameter was at its upper level (i.e. $\delta+ = 40°$) and $\overline{\ln R}$ for the 32 combinations when the parameter
was at its low level (i.e. $\delta- = 65°$):

$$A = \overline{\ln R}(\delta +) - \overline{\ln R}(\delta -)$$

(B2)

By applying a multiparameter approach it is also possible to the quantify parameter-parameter
interaction effects, for example, if the effect of $\delta$ depends on the choice of rift extension azimuth ($\phi$).
To do this, the results in Table B1 can be divided into two sets with $2^{k-p-1}$ combinations each
depending on which level of $\delta$ was applied. Following the table designs developed by Box et al.
(1978), each set of 32 combinations will have 16 combinations when $\phi$ was at is upper level ($\phi+$)





and 16 combinations when $\phi$ was at its lower level ($\phi$-). The effect of $\delta$ on each level of $\phi$ (i.e. $\delta\phi$) is
then calculated from the corresponding averages differences in $\overline{lnR}$ (Rabinowitz and Steinberg,

1991):


$$\delta\phi = \left( \overline{lnR}(\delta + \phi +) - \overline{lnR}(\delta - \phi +) \right) - \left( ln\bar{R}(\delta + \phi -) - \overline{lnR}(\delta - \phi -) \right)$$

(B3)

If there is no interaction effect between these two parameters, then $\delta\phi$ is 0. Otherwise, the size of the
effect is proportional to the magnitude of $\delta\phi$. In addition, we demonstrate our results in terms of an
empirical cumulative distribution function for the values of $lnR$ reported in Table B1 (Fig. B2a), and
following Rabinowitz and Steinberg (1991), values of $A$ in a normal probability plot (Fig. B1b).

If the Saria et al. (2013) model is used to provide estimates of $v$ in this sensitivity analysis, the
parameter that contributes most to uncertainties of $R$ in the SMAFD is the component of regional
extensional strain that each fault accommodates (A = 3.05, Table 4). This essentially means that $lnR$
is higher by 3.05 when this component is set at its high value compared to its lower, or that $R$ is ~21
times ($e3.05$) higher when 10% of regional extensional strain is assigned to the Chingale Step fault as
opposed to 2%. The importance of this parameter is also demonstrated by the fact that it does not
plot close to the normal distribution line in Fig. B1b. The parameters with the next highest main
effect on $R$ are $v$ and $C_2$, whilst estimates of $R$ are least sensitive to uncertainties in $\phi$ (Table 4). If,
however, estimates of $v$ are provided by the Stamps et al. (2008) model (Fig. B1), estimates of $R$ are
considerably less sensitive to uncertainites in rift extension rates, and the $C_2$ parameter has the
biggest influence on $R$ (Table B2). Multiparameter effects are all equal to zero (Table B3) regardless
of geodetic model, and thus the sensitivity of each of these parameters is independent of changes in
other parameters.





The results of the sensitivity analysis reported here are specific to estimates of $R$ for the Chingale
Step fault central section, however, results should be broadly applicable to all other faults in the
SMAFD as $R$ was calculated following the same steps. There will, however, be differences for faults
that are not segmented (where $L$ is not an uncertainty) or that have more than the three sections
mapped along the Chingale Step fault (e.g. the seven section Bilila-Mtakataka). The uncertainty in
the weighting of rift extension may also be different for border faults, as in these cases the weighting
factor ($\alpha_{bf}$) is varied between 0.5-0.9. The results of this analysis are discussed further in Sect. 5.3
and 6.3 in the main text.

**Data Availability**
The South Malawi Active Fault Database is available in the Supplement as a Shapefile.
**Author Contributions**
JW and LW led the fault mapping from TanDEM-X data, and HM led the fault mapping using
aeromagnetic data. All authors participated in the fieldwork. LW conducted analysis of geodetic
data. JW designed the method to obtain fault slip rates and earthquake source parameters with input
from all co-authors. JB and AF secured the funding for this project. All authors contributed to
manuscript preparation, but JW had primary responsibility.
**Competing interests**
The authors declare that they have no conflict of interest.
**Acknowledgements**
This work is supported by the EPSRC-Global Challenges Research Fund PREPARE project
(EP/P028233/1). TanDEM-X data were provided through DLR proposal DEM_GEOL0686. The
Geological Survey Department of Malawi kindly gave us access to the 2013 aeromagnetic surveys
across Malawi. We thank Katsu Goda and Mark Stirling for useful discussions on developing this





database, and Mike Floyd for his assistance with calculating geodetic extension rates from Euler
Poles.



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




**Figure 1**

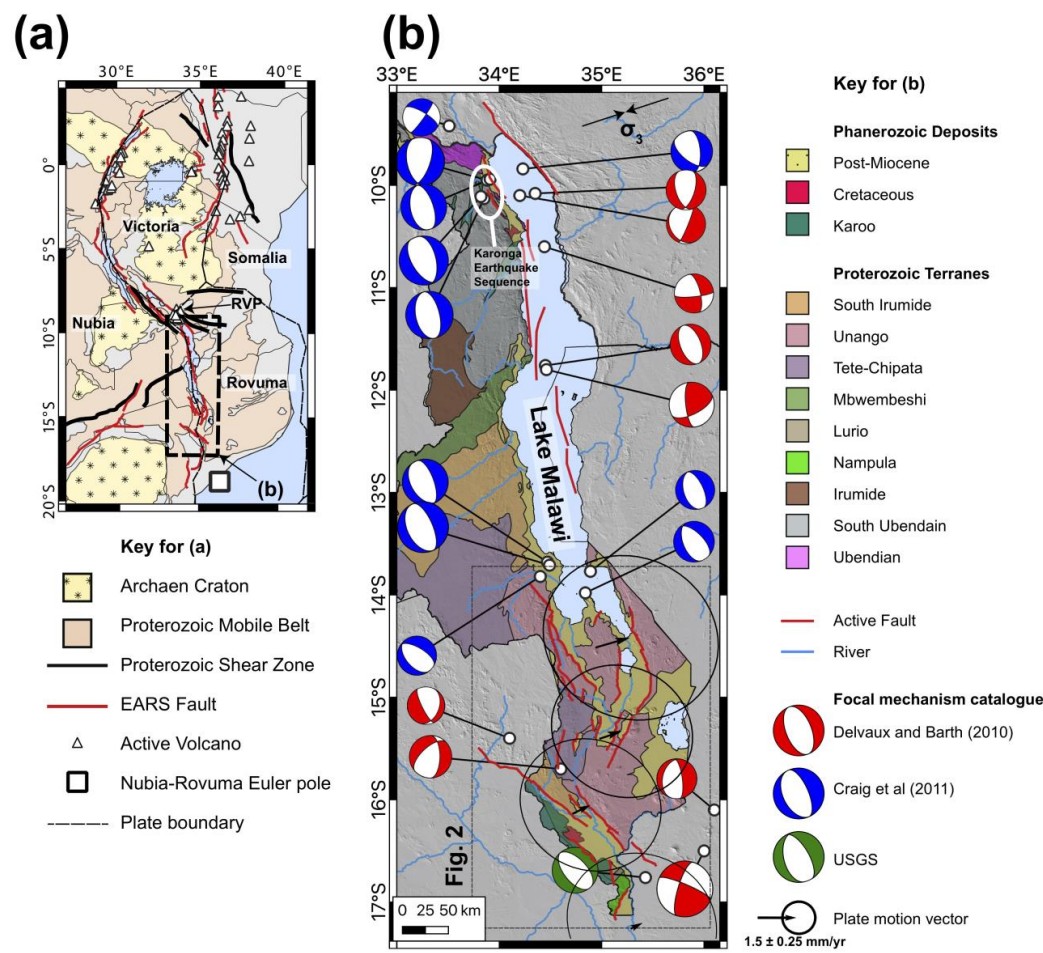


Figure 1: (a) Location of the Malawi within East Africa. Distribution of Archean Cratons,
Proterozoic Mobile Belts and shear zones modified after Fritz et al., (2013), major faults in the East
African Rift modified after Hodge et al., (2018). and Macgregor, (2015). Plate boundaries and the
Euler pole between the Nubia and Rovuma plates after Saria et al., (2013). RVP; Rungwe Volcanic
Province. (b) Simplified geological map of Malawi, with Proterozoic Terranes after Fullgraf et al.,
(2017). Map is underlain by Shuttle Radar Topography Mission (STRM) 30-m digital elevation





model (DEM; Sandwell et al., 2011). Extent of Fig. 2 also shown. Active faults within this area are
those included in the South Malawi Active Fault Database (SMAFD). Active faults outside this
region mapped as in (a). Focal mechanisms collated from Delvaux and Barth, (2010), Craig et al.,
(2011), and U.S. Department of the Interior U.S. Geological Survey, (2018). Minimum principal
compressive stress ($\sigma_3$) trend from focal mechanism stress inversion (Williams et al., 2019). Plate
motion vector for central point of each graben in southern Malawi (Fig. A2) for Nubia-Rovuma
Euler pole (Saria et al., 2013), modelled using methods described in Robertson et al., (2016).





**Figure 2**

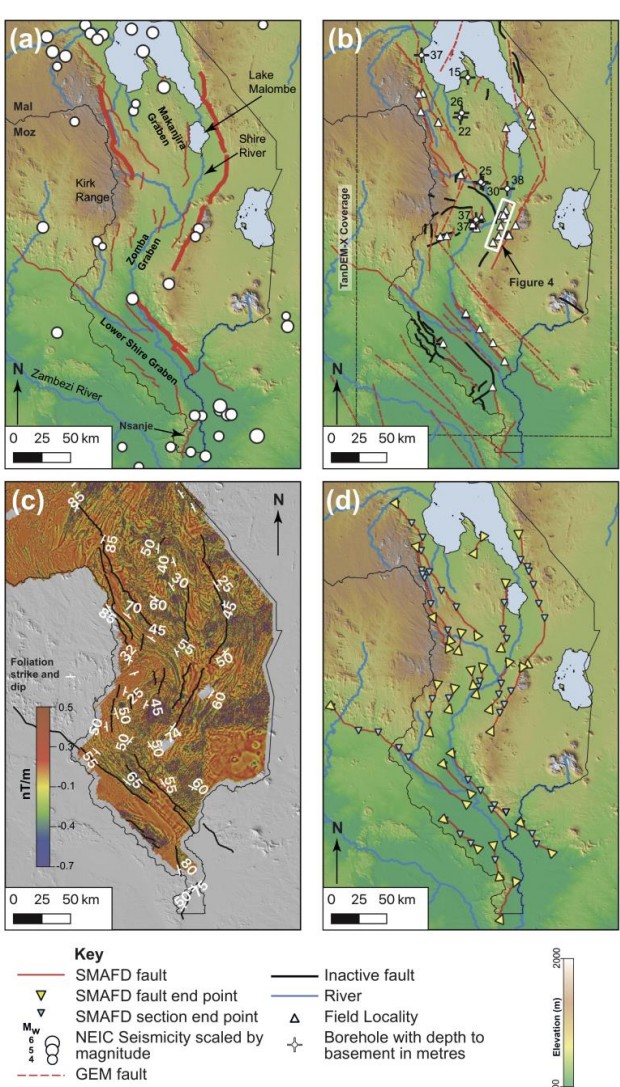


Figure 2: (a) Active fault map for south Malawi underlain by SRTM 30 m resolution DEM, division
of its principal grabens (with border faults heavily weighted), and National Earthquake Information
Centre (NEIC) record for events Mw>2.5 from 1900-February 2019 also shown. (b) Information on
methods used to collate the South Malawi Active Fault Database (SMAFD) and previous fault
mapping (c) Aeromagnetic image created from the vertical derivative. Combined with foliation



orientations digitised from geological maps (Bloomfield, 1958, 1965; Bloomfield and Garson,
1965b; Habgood et al., 1973; Walshaw, 1965), and underlain with the SMAFD faults shown in
black. For full details of the acquisition of the aeromagnetic data, see Laõ-Dávila et al., (2015). (d)
The SMAFD faults and section geometry. Extent of all maps is equivalent and is outlined in Fig. 1b.
GEM: Global Earthquake Model, Mal: Malawi, Moz: Mozambique.




**Figure 3**

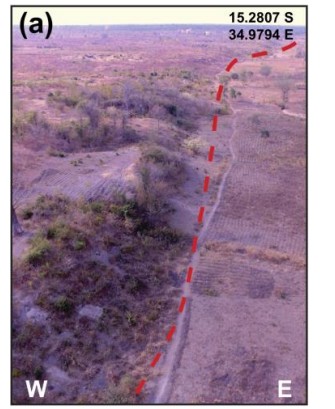
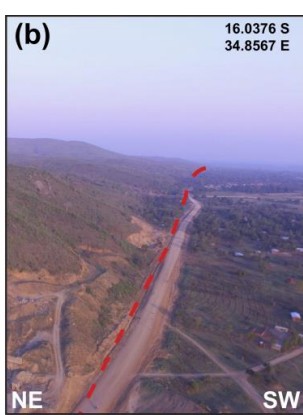

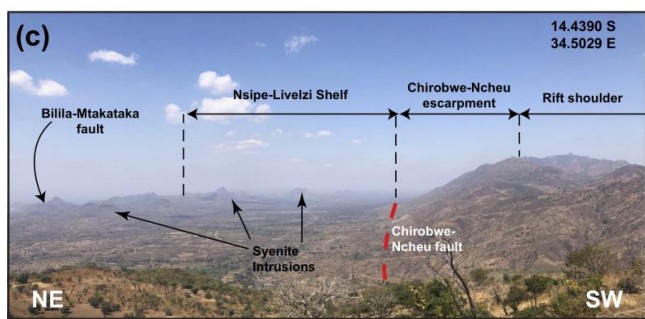

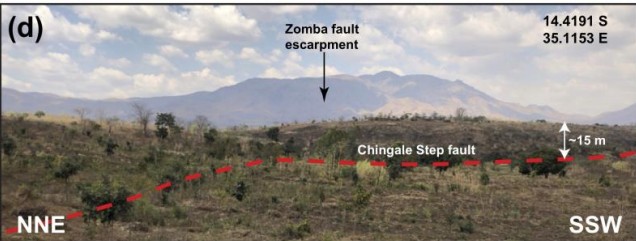

Figure 3: Field examples of border and intrabasinal faults in southern Malawi. Unmanned Aerial

Vehicle (UAV) images of scarps (dashed red line) along (a) intrabasinal Mlungusi fault in the

Zomba Graben, and (b) the Thyolo fault, the border fault for the Lower Shire Graben. (c) View

across the western edge of the Makanjira Graben showing the Chirobwe Ncheu and Bilila-

Mtakataka faults, and Proterozoic syenite intrusions (Walshaw, 1965). (d) Minor step in the scarp

along the intrabasinal Chingale Step fault, with the escarpment of the Zomba border fault behind.





**Figure 4**

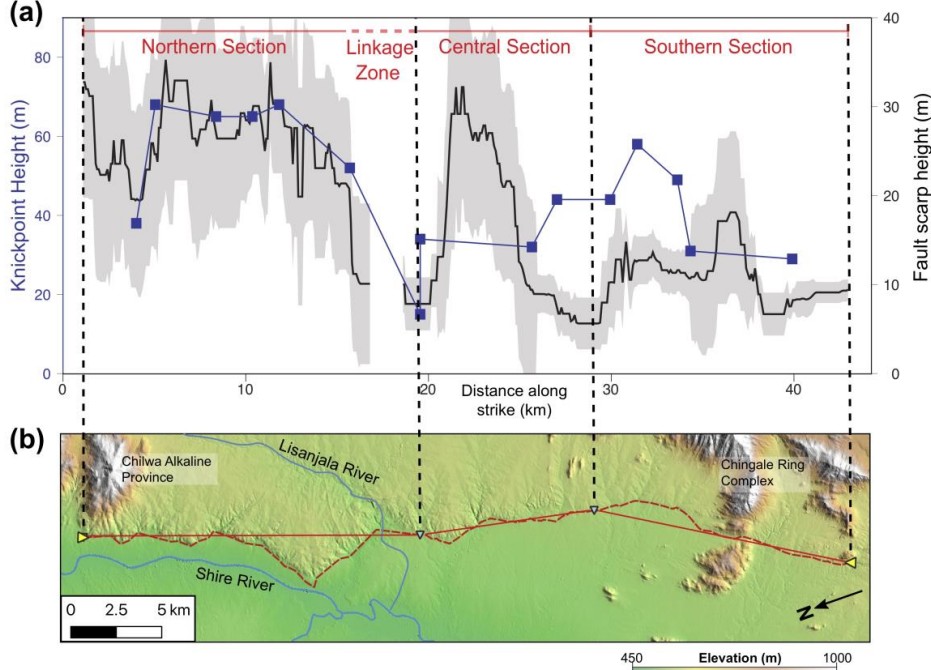


Figure 4: Fault segmentation along the Chingale Step fault, modified after Wedmore et al., (2020a).
(a) Along strike variation in stream knickpoint (blue points) and fault scarp height (black line), with
the gap due to erosion by the Lisanjala River. Grey shading represents one standard deviation error
in scarp height measurements (Wedmore et al., 2020a). (b) Map of Chingale Step fault underlain by
TanDEM-X DEM, extent of area shown in Fig. 2b. The dashed red line shows the surface trace of
the fault. The solid red line shows the geometry of the fault used in earthquake source modelling,
where it is defined by straight lines between section endpoints (blue triangles). An along-strike scarp
height minima at the boundary between the northern and central section occurs at a bend in the fault
scarp, however, there is no obvious geometrical complexity at the along strike scarp height minima
between the southern and central sections. Topography associated with the Proterozoic Chingale
Ring Structure and Chilwa Alkaline Province (Bloomfield, 1965; Manda et al., 2019) is also
indicated. For full details on (a) see Wedmore et al., (2020a).


**Figure 5**

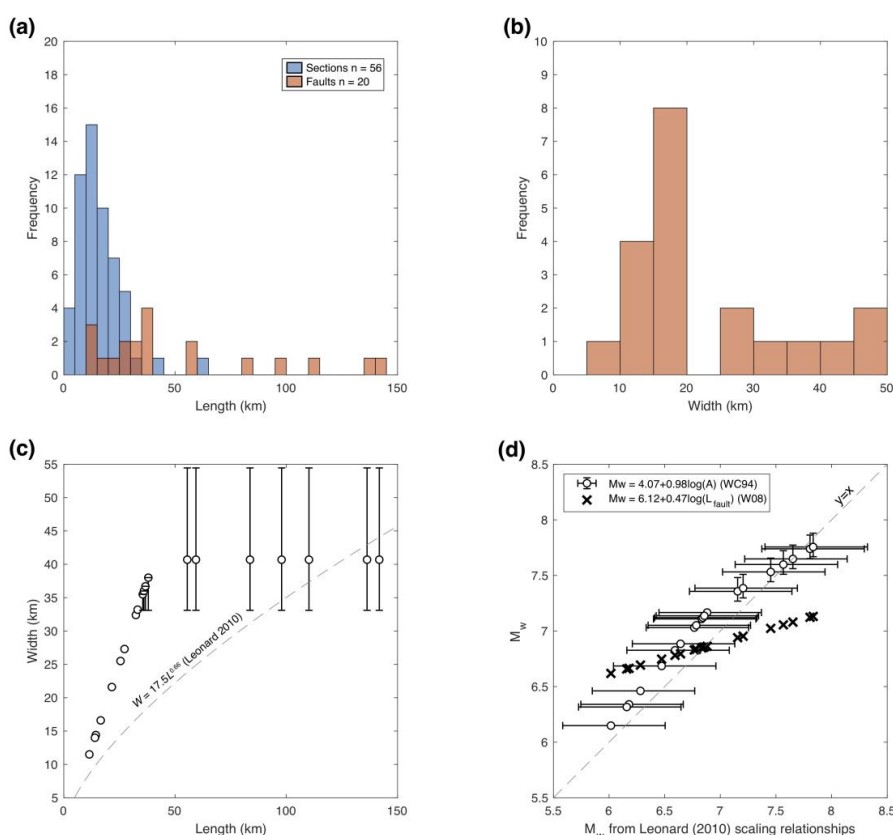


Figure 5: Assessment of fault geometry in the SMAFD. Histograms showing distribution of (a) fault
($L_{fault}$) and section ($L_{sec}$) lengths, and (b) fault widths ($W$) in the SMAFD. The latter is derived from
the Leonard, (2010) scaling relationship (Eq. (2)), and in (c) the predicted aspect ratio of faults
following this relationship (dashed grey line) is compared to an alternative method to estimate $W$
using Eq. (1) (white circles). (d) A comparison of empirical scaling relationships used to estimate
earthquake magnitudes (Mw) from fault geometry in the SMAFD. Leonard, (2010) magnitudes
estimated using Eq. (4), with error bars representing range of $C_1$ and $C_2$ values derived for interplate
dip-slip faults. $A$, fault area calculated from $L_{fault}$ and $W$ using Eq. (1); WC 94, Wells and
Coppersmith (1994); W08, Wesnousky (2008).





**Figure 6**

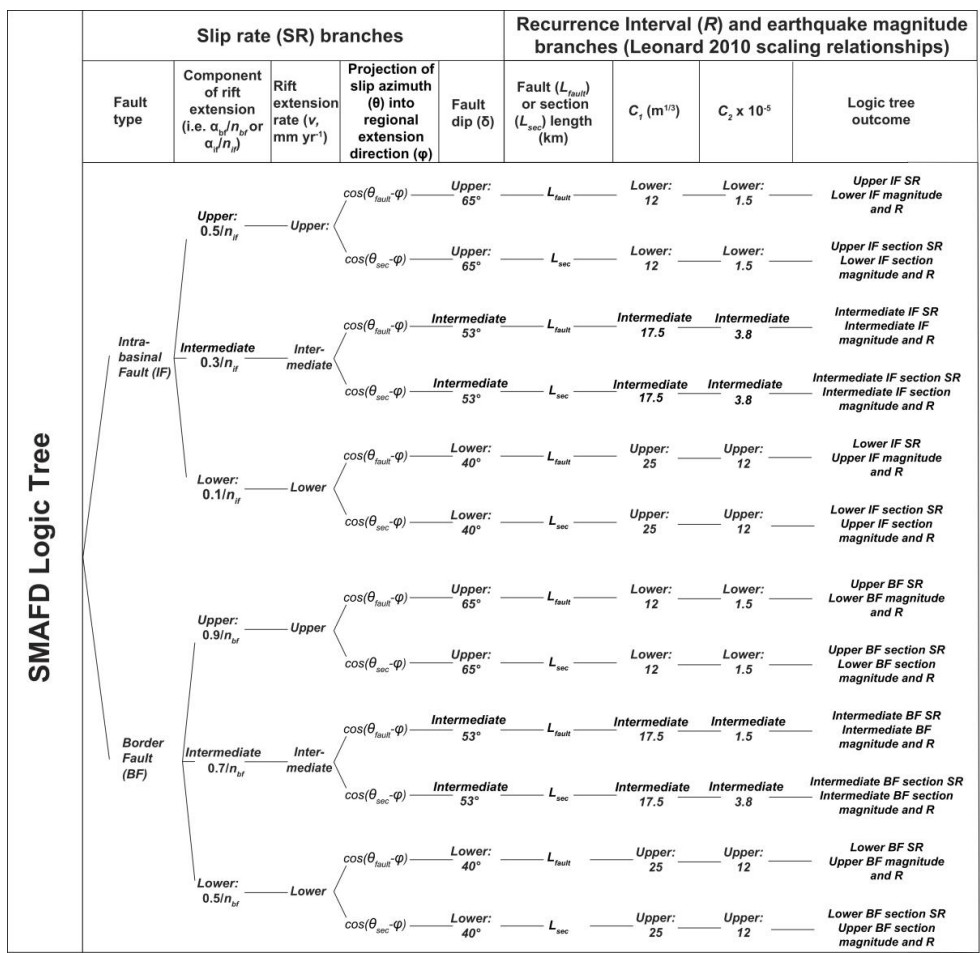


Figure 6: Logic tree for calculating lower, intermediate, and upper estimates of fault slip rates and
earthquake magnitudes and recurrence intervals in the SMAFD; $\alpha_{bf}$ and $\alpha_{if}$ are the rift extension
weighting assigned to border faults and intrabasinal faults respectively; $n_{bf}$ and $n_{if}$ are the number of
border or intrabasinal faults in a graben, $\theta_{fault}$ and $\theta_{sec}$ are whole fault and individual section slip
azimuth.





**Figure 7**

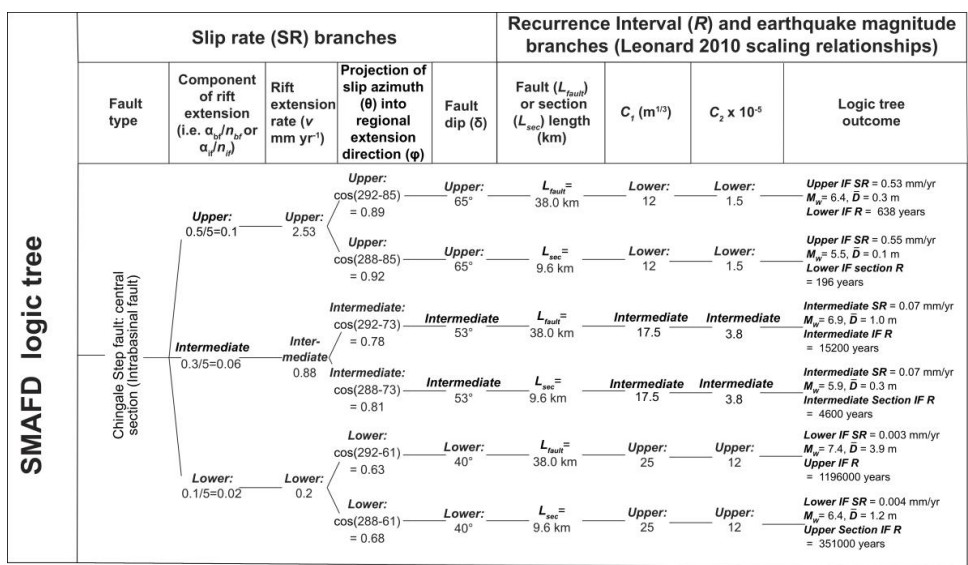


Figure 7: Example of the calculations in the SMAFD logic tree (Fig. 6), performed for the central
section of the Chingale Step Fault (Fig. 4b). This is an intrabasinal fault in the Zomba Graben, where
the number of intrabasinal faults ($n_{if}$) is five. A multiparameter sensitivity analysis for these
calculations is documented in Appendix B.



**Figure 8**

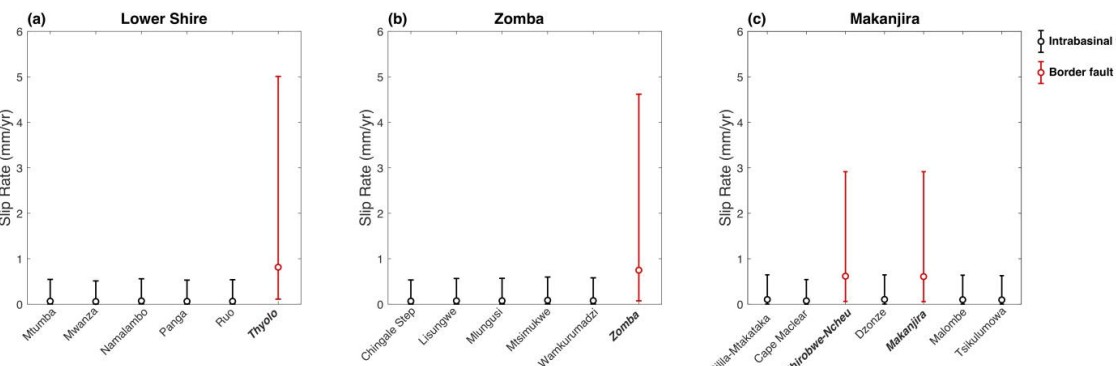


Figure 8: Fault slip rate estimates in the SMAFD, calculated following approach outlined in Fig. 6
for faults in (a) Lower Shire graben, (b) Zomba graben, and (c) Makanjira graben. Middle point
represents intermediate estimate with error bars representing upper and lower estimates. Faults with
red data points, and names that are bold and italicised are classified as border faults in the SMAFD,
the remaining faults are intrabasinal.



**Figure 9**

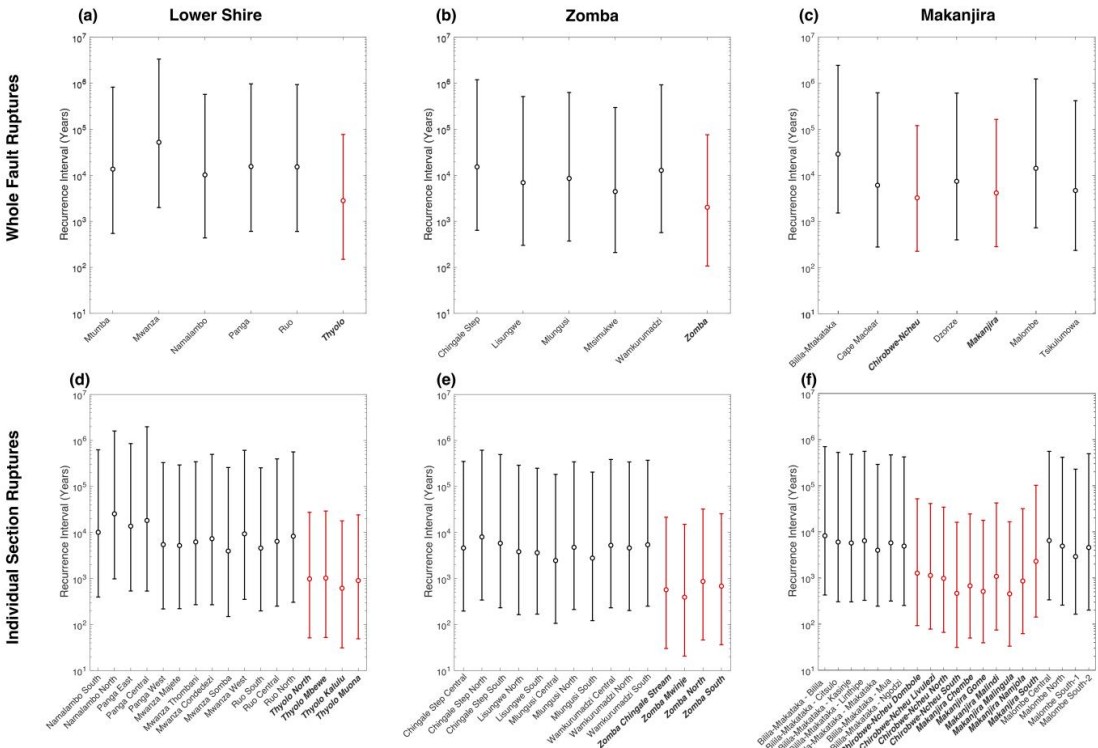


Figure 9: Recurrence interval estimates in the SMAFD for (a-c) whole fault ruptures and (d-f)
individual fault section ruptures. Calculated following approach outlined in Fig. 6, with middle point
representing intermediate estimate, and error bars representing lower and upper estimates. Faults that
are bold and italicised are classified as border faults.



**Figure 10**

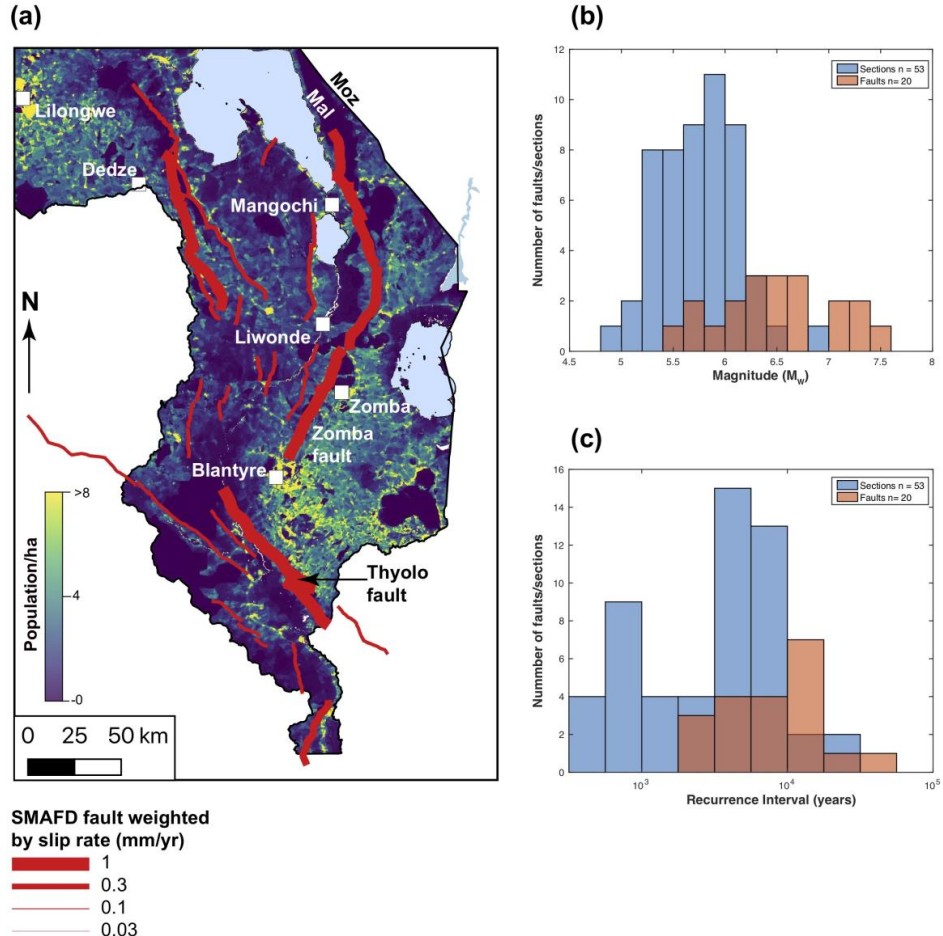


Figure 10: (a) Faults in the SMAFD with lines weighted by intermediate estimates of fault
slip rate. Fault map is underlain by population density where the pixel size is 3 arcseconds
(approximately 1 ha) as derived from WorldPop predicted 2020 datasets for Malawi
(WorldPop, 2018) with major population centres also highlighted. Note that population
density in these places may exceed 100 people/ha. Area shown is same as in Fig. 2.
Histograms to show range of (b) earthquake magnitudes and (c) recurrence interval estimates
in the SMAFD from intermediate branches in Fig. 6.



**Figure A1**

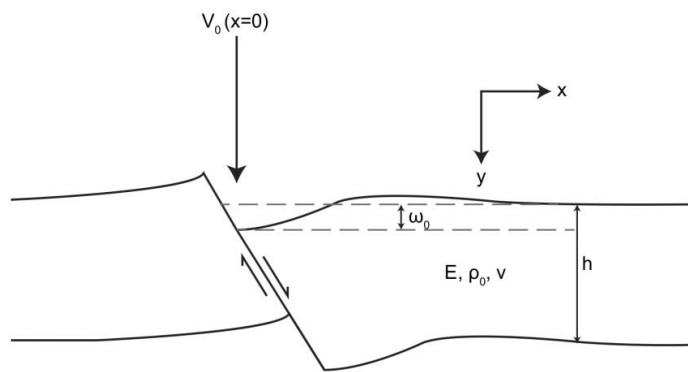


Figure A1: Set-up for hanging wall deflection equations. A vertical load ($V_0$) is applied to the
point where the hanging-wall intersects the surface (i.e. where $x=0$) and where there is a
maximum deflection ($\omega_0$). The elastic thickness, Young's Modulus, density, and Poisson's
ratio of the crust are represented by $h$, $E$, $\rho_0$, and $v$ respectively.



**Figure A2**

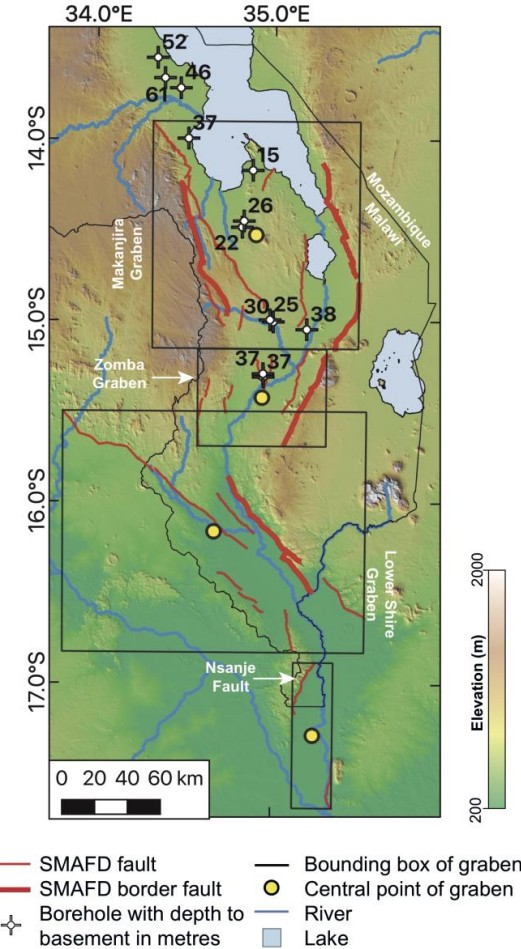


Figure A2: Active fault map for southern Malawi showing respective extent of grabens, their border
faults, and the central point of each graben from which the Nubia-Rovuma plate motion vectors were
derived. Location of boreholes that penetrate basement also shown (Bloomfield and Garson, 1965b;
Walshaw, 1965; Walter, 1972). Map underlain by 30 m resolution Shuttle Radar Topographic
Mission digital elevation model.


**Figure A3**

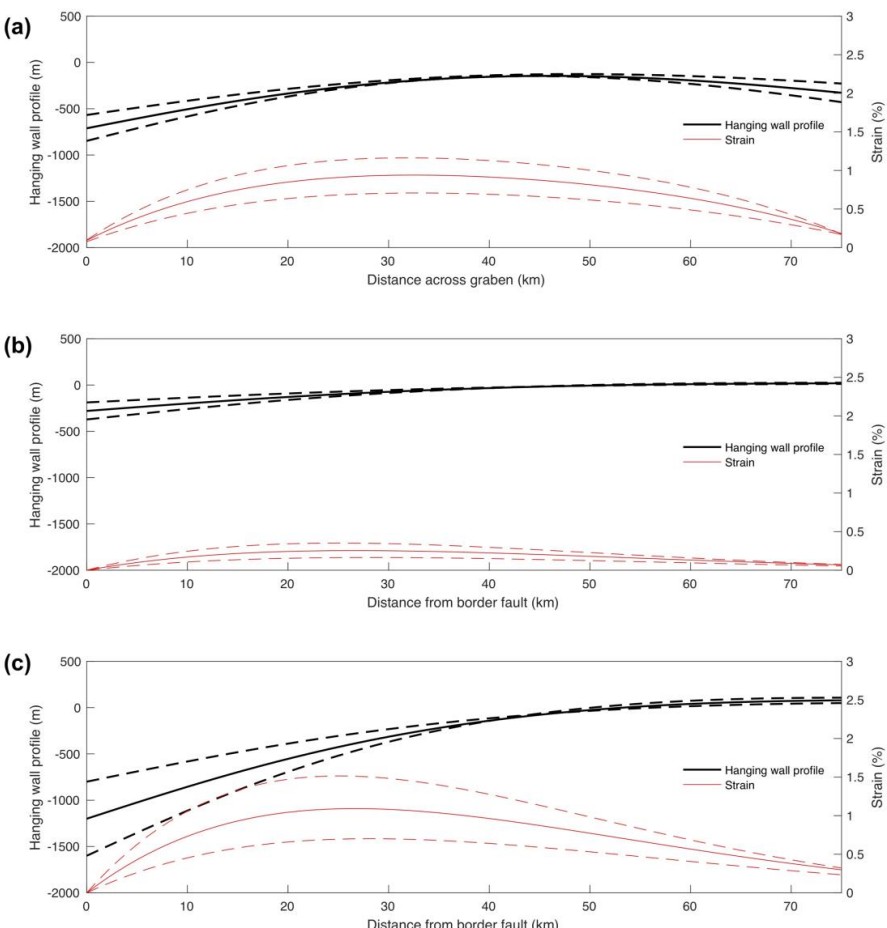


Figure A3: Flexural profiles and horizontal extensional strain for (a) Makanjira, (b) Zomba, and (c)
Lower Shire grabens, calculated following methods described in Appendix A and parameters listed
in Table A1. Solid line indicates median estimates, dashed line indicates maximum and minimum
estimates. For (a), profile shows flexure from both sides of the Makanjira graben and is shown left to
right in a WSW-ESE section. Flexural profile is relative to point of zero hanging-wall deflection, not
absolute elevation.



**Figure B1**

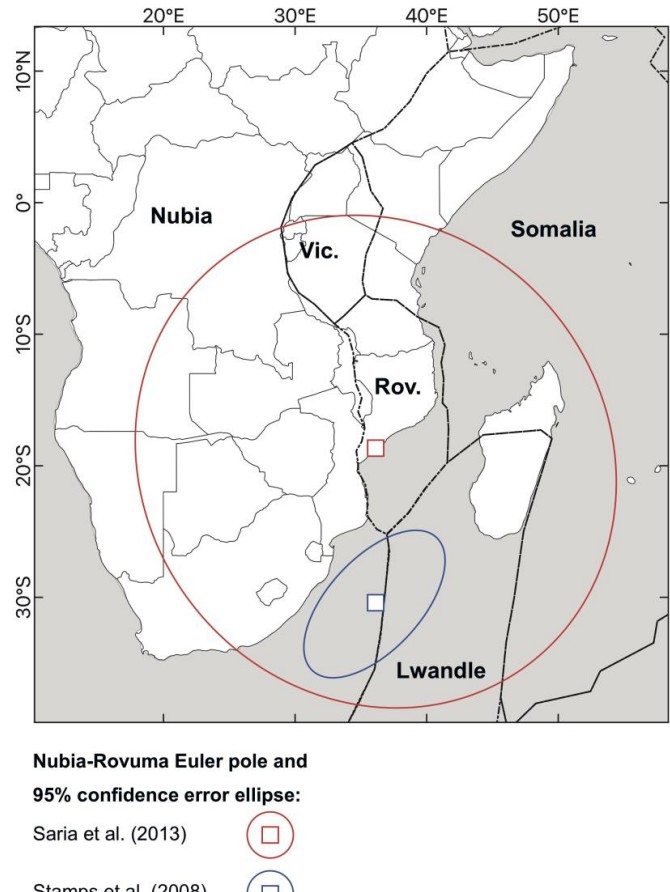


Figure B1: Plate boundaries in East Africa with location and uncertainty of the Nubia-Rovuma Euler
pole derived by Saria et al. (2013) and Stamps et al. (2008). Vic., Victoria; Rov., Rovuma. Modified
after Saria et al. (2013).




**Figure B2**

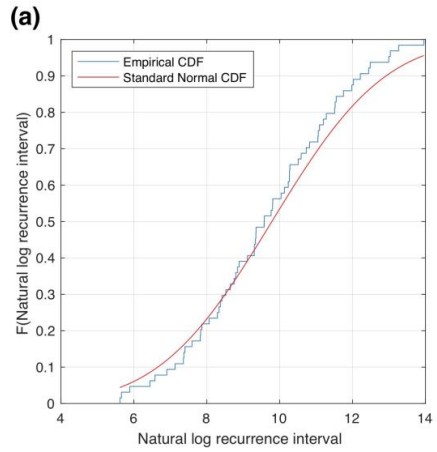
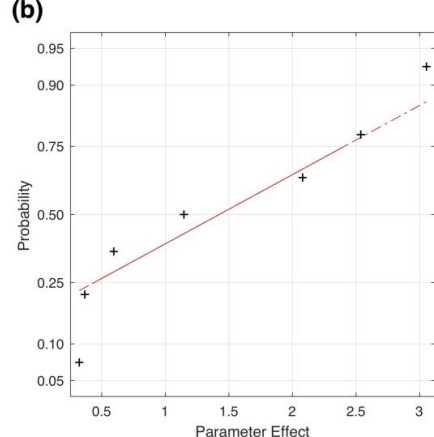


Figure B1: (a) Cumulative distribution function (CDF) of the natural log of the recurrence intervals
calculated for the Chingale Step fault central section using the various parameter combinations listed
in Table B1 (blue line). This CDF is compared to a standard normal CDF (red line) with the same
mean value and standard deviation as the values in Table B1. (b) Normal probability plot of the
parameter effects assessed in the sensitivity analysis and reported in Table 4. The most important
effects are those that plot above a standard normal distribution (red line). Line is solid when within
first and third quartiles of data and dashed when outside.





**List of Tables**
**Table 1**

| Attribute type and hierarchical assignment | Attribute | Type | Description | Notes |
|---|---|---|---|---|
| Trace: assigned by trace | Trace ID | Numeric, assigned | Unique two-digit numerical reference ID for each trace | |
| | Fault Name | Text | Fault that trace belongs to. | Assigned based on previous mapping or local geographic feature. |
| | Graben | Text | Graben that fault is located within. | Used in slip rate calculations. |
| | Geomorphic Expression | Text | Geomorphological feature used to identify and map fault trace. | E.g. scarp, escarpment |
| | Location Method | Text | Dataset used to map trace. | E.g. type of digital elevation model |
| | Scale | Numeric, assigned | Coarsest scale at which trace can be mapped. | Reflects the prominence of the fault's geomorphic expression. |
| | Confidence | Numeric, assigned | Score between 1-4 that geomorphic feature used to map trace is an active fault. | |
| | Trace notes | Text | Any remaining miscellaneous geomorphological information about fault trace. | |
| | Author | Text | Fault trace mapper | |
| Section and fault geometry: assigned by section or fault | Section Name | Text | | Assigned based on previous mapping, local geographic feature, or location along fault. |
| | Section Length ($L_{sec}$) | Numeric, assigned | Straight-line distance between section tips. | Measured in km. Except for linking sections, must be >5 km. |
| | Fault Length ($L_{fault}$) | Numeric, assigned | Straight-line distance between fault tips or sum of $L_{sec}$ for segmented faults. | Measured in km |





| | Section strike | Numeric, assigned | Measured from section tips, using bearing that is <180°. | |
|---|---|---|---|---|
| | Fault strike | Numeric, assigned | Measured from fault tips using bearing <180°. | For segmented (i.e. non-planar) this is an 'averaged' value of fault geometry, which is required for slip rate estimates (Eq. (3)). |
| | Dip ($\delta$) | Numeric, assigned | | Attribute parameterised by a set of representative values (40, 53, 65°). |
| | Dip Direction | Text | Compass quadrant that fault dips in. | |
| | Fault Width ($W$) | Numeric, calculated | Calculated from Eq. (2) from Leonard, (2010) scaling relationship using $L_{fault}$. | Not equivalent to rupture width for individual section earthquakes. |
| Kinematic and earthquake source information: assigned by section and/or fault | Section net slip rate | Numeric, calculated | Calculated from Eq. (3). | In mm/yr. All faults in the SMAFD assumed to be normal, so is equivalent to dip-slip rate. |
| | Fault net slip rate | Numeric, calculated | Calculated from Eq. (3). | In mm/yr. All faults in the SMAFD assumed to be normal, so is equivalent to dip-slip rate. Different from section net slip rate where fault strike ≠ section strike. |
| | Section earthquake magnitude | Numeric, calculated | Calculated from Leonard, (2010) scaling relationship using Eq. (4) and $L_{sec.}$ | |
| | Fault earthquake magnitude | Numeric, calculated | Calculated from Leonard, (2010) scaling relationship using Eq. (4) and $L_{fault.}$ | |
| | Section earthquake recurrence interval ($R$) | Numeric, calculated | Calculated from Eq. (6) and using $L_{sec}$ to calculate average single event displacement in Eq. (5). | |
| | Fault earthquake recurrence interval ($R$) | Numeric, calculated | Calculated from Eq. (6) and using $L_{fault}$ to calculate average single | |





| | | | event displacement in Eq. (5). | |
|---|---|---|---|---|
| Miscellaneous attributes: assigned by fault | Last event | Text | | Currently this is unknown for all faults in southern Malawi but can be updated when new information becomes available. |
| | Data completeness | Numeric, assigned | Assessment of quality of data and level of knowledge. Score between 1-4, where 1 indicates high completeness. | |
| | Fault notes | Text | Remaining miscellaneous information about fault. | Includes whether fault is classified as a border fault. |
| | References | Text | Relevant geological maps/literature where fault has been previously described. | |
| | Date last updated | Date | | |
| | Compiler | Text | | Not necessarily the same as the fault trace mapper. |

Table 1: List and brief description of attributes in the SMAFD. Representative values for numeric
attributes are reported in Table 3.



**Table 2**

| Graben | Centre of graben longitude (E) | Centre of graben latitude (S) | Geodetic Model | Velocity and velocity uncertainty of plate motion (mm/yr) | Azimuth, and azimuthal uncertainty of plate motion |
|---|---|---|---|---|---|
| Makanjira | 34.89 | 14.52 | S13 | 1.08 ± 1.66 | 075° ± 089° |
| | | | S08 | 3.01 ± 0.28 | 085º ± 002º |
| Zomba | 34.93 | 15.42 | S13 | 0.88 ± 1.65 | 072° ± 110° |
| | | | S08 | 2.84 ± 0.28 | 085º ± 002º |
| Lower Shire | 34.66 | 16.16 | S13 | 0.74 ± 1.63 | 063° ± 131° |
| | | | S08 | 2.71 ± 0.28 | 084º ± 002º |
| Nsanje | 35.23 | 17.28 | S13 | 0.46 ± 1.63 | 063° ± 212° |
| | | | S08 | 2.49 ± 0.27 | 086º ± 002º |

Table 2: Coordinates from which the Nubia-Rovuma plate motion vector for each graben in southern
Malawi was derived (Fig. 1b). The velocity, azimuth, and uncertainties of each vector is also
reported given the Nubia-Rovuma Euler poles reported in Saria et al. (2013) (S13), or in Stamps et
al., (2008) (S08; Fig. B1), and where the uncertainties associated with the Euler pole are derived
from the methods presented in Robertson et al. (2016). For justification of graben centre locations,
see Fig. A2.





**Table 3**

| Attribute | Minimum | Median | Maximum |
|---|---|---|---|
| Section Length ($L_{sec}$, km) | 3.0 | 13.9 | 62.4 |
| Fault Length ($L_{fault}$, km) | 11.5 | 35.7 | 141.8 |
| Fault Width ($W$, km) | 8.9 | 19.0 | 47.6 |
| Section net slip rate (mm/yr) | 0.03 | 0.10 | 0.84 |
| Fault net slip rate (mm/yr) | 0.06 | 0.13 | 0.81 |
| Section earthquake magnitude (Mw) | 5.4 | 6.2 | 7.2 |
| Fault earthquake magnitude (Mw) | 6.0 | 6.8 | 7.8 |
| Section earthquake recurrence interval ($R$, years) | 390 | 4580 | 25500 |
| Fault earthquake recurrence interval ($R$, years) | 2000 | 7190 | 52200 |

Table 3: Range of selected numeric attributes across all faults and sections in the SMAFD. To
demonstrate how calculated attributes vary across different faults in the SMAFD, as opposed to
variation from the set of parameters used to calculate them, the values shown are for the
intermediate branches in the SMAFD logic tree (Fig. 6).





**Table 4**

| Parameter | Lower Level | Upper Level | Parameter Main Effect ($A$) |
|---|---|---|---|
| **Component of regional extensional strain ($\alpha_{if}/n_{if}$)** | 0.1 | 0.02 | 3.05 |
| **Rift extension rate ($v$, mm/yr)** | 2.53 | 0.2 | 2.54 |
| **Rift extension azimuth ($\phi$)** | 085° | 061° | 0.32 |
| **Fault dip ($\delta$)** | 65° | 40° | 0.59 |
| **Leonard, (2010) empirically derived scaling parameter $C_1$ ($m_{1/3}$)** | 12 | 25 | 0.37 |
| **Leonard, (2010) empirically derived scaling parameter $C_2$** | 1.5 | 12 | 2.08 |
| **Rupture length ($L$, km)** | 9.6 (individual section, $L_{sec}$) | 38.0 (whole fault, $L_{fault}$) | 1.15 |

Table 4: Parameters and their associated upper and lower levels used in the sensitivity analysis for
recurrence interval ($R$) calculations for the Chingale Step fault central section. The main effect of
each parameter is then also reported. See Appendix B for full details of this analysis.

**Table A1**

| **Half graben** | **Sediment thickness (m)** | **Escarpment height (m)** | **Border fault throw (m)** | **$\omega_0$ (m)** | **Elastic plate thickness (km)** | **Mean extension (%)** | **Total horizontal extension (m)** | **Half-graben age (Ma)** | **Total flexural extension rate (mm/yr)** | **Flexural extension rate per fault (mm/yr)[1]** |
|---|---|---|---|---|---|---|---|---|---|---|
| **North Basin[2]** | | | | | | $3.8^{+2.8}_{-1.3}$ | $1280^{+6}_{-5}$ | | | |
| **Makanjira (East)** | 70±40[3] | 400±100[5] | 470±140 | 370±110 | 32.5±2.5 | 0.7±0.2 | 510±120 | 4.6-25[7] | $0.035^{+0}_{-0}$ | $0.01^{+0.02}_{-0.00}$ |
| **Makanjira (West)** | 70±40[3] | 850±150[5] | 920±190 | 740±150 | 32.5±2.5 | | | | | |
| **Zomba** | 50±15[4] | 300±100[5] | 350±115 | 280±90 | 32.5±2.5 | 0.2±0.1 | 100±40 | 4.6-25[7] | $0.007^{+0}_{-0}$ | $0.001^{+0.}_{-0.}$ |
| **Lower Shire** | | | 1500±500[6] | 1200±400 | 32.5±2.5 | 0.9±0.3 | 430±160 | 4.6-25[7] | $0.015^{+0}_{-0}$ | $0.004^{+0.}_{-0.}$ |

Table A5: Inputs and results of hanging wall flexure analysis. $\omega_0$; maximum hanging-wall deflection
calculated from Eq. A3.[1]Extension rate per fault, calculated from dividing the rift-wide flexural
extension rate by the number of intrabasinal faults along a rift perpendicular transect. [2]Values
reported are median, upper, and lower estimates of hanging-wall flexure along three profiles across





the North Basin of Lake Malawi by Shillington et al. (2020). [3]Thickness of sediments in the
Bwande-Liwawadze Valley based on electrical resistivity surveys (Mynatt et al., 2017; Walshaw,
1965) and borehole data. (Fig. A2; Bloomfield and Garson, 1965). [4]Thickness of sediments from
borehole data within the Shire Plain (Fig. A2; Bloomfield and Garson, 1965). [5]See Laõ-Dávila et al.
(2015). For the Zomba Graben, topography associated with Chilwa Alkaline Province intrusion at
the northern end of the Zomba Fault is removed. For Makanjira West, incorporates escarpment
height from Chirbowe-Ncheu and Bilila-Mtakataka Fault. [6]No boreholes in the Lower Shire Graben
penetrated basement (Habgood et al., 1973). See Appendix A for justification of throw estimates
used instead. [7]Range of ages from estimates for the onset of East African Rift Western Branch
rifting (~25 Ma; Roberts et al., 2012), and the onset of sediment accumulation in Lake Malawi (4.6
Ma; McCartney and Scholz, 2016).





**Table B1**

| Combination | Component of regional extension | $v$ (mm/yr) | $\varphi$ (°) | $\delta$ | C1 ($m_{1/3}$) | C2 | $L$ (km) | Natural log Recurrence Interval |
|---|---|---|---|---|---|---|---|---|
| 1 | 0.1 | 2.53 | 85 | 65 | 12 | 12 | 9.6 | 7.37 |
| 2 | 0.02 | 2.53 | 85 | 65 | 12 | 1.5 | 9.6 | 6.90 |
| 3 | 0.1 | 0.2 | 85 | 65 | 12 | 1.5 | 9.6 | 7.83 |
| 4 | 0.02 | 0.2 | 85 | 65 | 12 | 12 | 9.6 | 11.52 |
| 5 | 0.1 | 2.53 | 85 | 65 | 12 | 1.5 | 38 | 6.44 |
| 6 | 0.02 | 2.53 | 85 | 65 | 12 | 12 | 38 | 10.13 |
| 7 | 0.1 | 0.2 | 85 | 65 | 12 | 12 | 38 | 11.06 |
| 8 | 0.02 | 0.2 | 85 | 65 | 12 | 1.5 | 38 | 10.59 |
| 9 | 0.1 | 2.53 | 61 | 65 | 12 | 1.5 | 9.6 | 5.62 |
| 10 | 0.02 | 2.53 | 61 | 65 | 12 | 12 | 9.6 | 9.31 |
| 11 | 0.1 | 0.2 | 61 | 65 | 12 | 12 | 9.6 | 10.24 |
| 12 | 0.02 | 0.2 | 61 | 65 | 12 | 1.5 | 9.6 | 9.77 |
| 13 | 0.1 | 2.53 | 61 | 65 | 12 | 12 | 38 | 8.84 |
| 14 | 0.02 | 2.53 | 61 | 65 | 12 | 1.5 | 38 | 8.37 |
| 15 | 0.1 | 0.2 | 61 | 65 | 12 | 1.5 | 38 | 9.30 |
| 16 | 0.02 | 0.2 | 61 | 65 | 12 | 12 | 38 | 12.99 |
| 17 | 0.1 | 2.53 | 85 | 40 | 12 | 1.5 | 9.6 | 5.89 |
| 18 | 0.02 | 2.53 | 85 | 40 | 12 | 12 | 9.6 | 9.58 |
| 19 | 0.1 | 0.2 | 85 | 40 | 12 | 12 | 9.6 | 10.51 |
| 20 | 0.02 | 0.2 | 85 | 40 | 12 | 1.5 | 9.6 | 10.04 |
| 21 | 0.1 | 2.53 | 85 | 40 | 12 | 12 | 38 | 9.12 |
| 22 | 0.02 | 2.53 | 85 | 40 | 12 | 1.5 | 38 | 8.65 |
| 23 | 0.1 | 0.2 | 85 | 40 | 12 | 1.5 | 38 | 9.57 |
| 24 | 0.02 | 0.2 | 85 | 40 | 12 | 12 | 38 | 13.26 |
| 25 | 0.1 | 2.53 | 61 | 40 | 12 | 12 | 9.6 | 8.29 |
| 26 | 0.02 | 2.53 | 61 | 40 | 12 | 1.5 | 9.6 | 7.82 |
| 27 | 0.1 | 0.2 | 61 | 40 | 12 | 1.5 | 9.6 | 8.75 |
| 28 | 0.02 | 0.2 | 61 | 40 | 12 | 12 | 9.6 | 12.44 |
| 29 | 0.1 | 2.53 | 61 | 40 | 12 | 1.5 | 38 | 7.36 |
| 30 | 0.02 | 2.53 | 61 | 40 | 12 | 12 | 38 | 11.05 |
| 31 | 0.1 | 0.2 | 61 | 40 | 12 | 12 | 38 | 11.98 |
| 32 | 0.02 | 0.2 | 61 | 40 | 12 | 1.5 | 38 | 11.51 |
| 33 | 0.1 | 2.53 | 85 | 65 | 25 | 1.5 | 9.6 | 5.66 |





| Combination | Component of regional extension | $v$ (mm/yr) | $\varphi$ (°) | $\delta$ | C1 ($m^{1/3}$) | C2 | $L$ (km) | Natural log Recurrence Interval |
|---|---|---|---|---|---|---|---|---|
| 34 | 0.02 | 2.53 | 85 | 65 | 25 | 12 | 9.6 | 9.35 |
| 35 | 0.1 | 0.2 | 85 | 65 | 25 | 12 | 9.6 | 10.28 |
| 36 | 0.02 | 0.2 | 85 | 65 | 25 | 1.5 | 9.6 | 9.81 |
| 37 | 0.1 | 2.53 | 85 | 65 | 25 | 12 | 38 | 8.89 |
| 38 | 0.02 | 2.53 | 85 | 65 | 25 | 1.5 | 38 | 8.42 |
| 39 | 0.1 | 0.2 | 85 | 65 | 25 | 1.5 | 38 | 9.35 |
| 40 | 0.02 | 0.2 | 85 | 65 | 25 | 12 | 38 | 13.04 |
| 41 | 0.1 | 2.53 | 61 | 65 | 25 | 12 | 9.6 | 8.07 |
| 42 | 0.02 | 2.53 | 61 | 65 | 25 | 1.5 | 9.6 | 7.60 |
| 43 | 0.1 | 0.2 | 61 | 65 | 25 | 1.5 | 9.6 | 8.52 |
| 44 | 0.02 | 0.2 | 61 | 65 | 25 | 12 | 9.6 | 12.21 |
| 45 | 0.1 | 2.53 | 61 | 65 | 25 | 1.5 | 38 | 7.13 |
| 46 | 0.02 | 2.53 | 61 | 65 | 25 | 12 | 38 | 10.82 |
| 47 | 0.1 | 0.2 | 61 | 65 | 25 | 12 | 38 | 11.75 |
| 48 | 0.02 | 0.2 | 61 | 65 | 25 | 1.5 | 38 | 11.28 |
| 49 | 0.1 | 2.53 | 85 | 40 | 25 | 12 | 9.6 | 8.34 |
| 50 | 0.02 | 2.53 | 85 | 40 | 25 | 1.5 | 9.6 | 7.87 |
| 51 | 0.1 | 0.2 | 85 | 40 | 25 | 1.5 | 9.6 | 8.79 |
| 52 | 0.02 | 0.2 | 85 | 40 | 25 | 12 | 9.6 | 12.48 |
| 53 | 0.1 | 2.53 | 85 | 40 | 25 | 1.5 | 38 | 7.40 |
| 54 | 0.02 | 2.53 | 85 | 40 | 25 | 12 | 38 | 11.09 |
| 55 | 0.1 | 0.2 | 85 | 40 | 25 | 12 | 38 | 12.02 |
| 56 | 0.02 | 0.2 | 85 | 40 | 25 | 1.5 | 38 | 11.55 |
| 57 | 0.1 | 2.53 | 61 | 40 | 25 | 1.5 | 9.6 | 6.58 |
| 58 | 0.02 | 2.53 | 61 | 40 | 25 | 12 | 9.6 | 10.27 |
| 59 | 0.1 | 0.2 | 61 | 40 | 25 | 12 | 9.6 | 11.20 |
| 60 | 0.02 | 0.2 | 61 | 40 | 25 | 1.5 | 9.6 | 10.73 |
| 61 | 0.1 | 2.53 | 61 | 40 | 25 | 12 | 38 | 9.81 |
| 62 | 0.02 | 2.53 | 61 | 40 | 25 | 1.5 | 38 | 9.34 |
| 63 | 0.1 | 0.2 | 61 | 40 | 25 | 1.5 | 38 | 10.26 |
| 64 | 0.02 | 0.2 | 61 | 40 | 25 | 12 | 38 | 13.95 |

Table B1: Input parameter combinations and Chingale Step fault central section recurrence intervals
using upper and lower values outlined in Table 4, and a fractional factorial approach with $2_{k-p}$



combinations where $k$ is 7 and $p$ is 1. The design of this table (i.e. whether an upper or lower value
of each parameter is chosen) is derived from Box et al. (1978) and can be accessed at:
https://www.itl.nist.gov/div898/handbook/pri/section3/eqns/2to7m1.txt (date last accessed

1802    30/03/2020).





**Table B2**

| Parameter | Lower Level | Upper Level | S08 Parameter Main Effect ($A$) | S13 Parameter Main Effect ($A$) |
|---|---|---|---|---|
| **Component of regional extensional strain ($\alpha_{if}/n_{if}$)** | 0.1 | 0.02 | 1.88 | 3.05 |
| **Rift extension rate ($v$, mm/yr)** | 2.56 | 3.12 | 0.20 | 2.54 |
| **Rift extension azimuth ($\phi$)** | 085° | 061° | 0.32 | 0.32 |
| **Fault dip ($\delta$)** | 65° | 40° | 0.59 | 0.59 |
| **Leonard, (2010) empirically derived scaling parameter $C_1$ ($m_{1/3}$)** | 12 | 25 | 0.37 | 0.37 |
| **Leonard, (2010) empirically derived scaling parameter $C_2$** | 1.5 | 12 | 2.08 | 2.08 |
| **Rupture length ($L$, km)** | 9.6 (individual section, $L_{sec}$) | 38.0 (whole fault, $L_{fault}$) | 1.15 | 1.15 |



Table B2: As for Table 4 with parameters and their associated upper and lower levels used in the
sensitivity analysis for the Chingale Step fault central section recurrence interval ($R$) calculations,
however, using the Stamps et al. (2008) (S08) Nubia-Rovuma Euler pole instead (Fig. B1). For
comparison, the Parameter Main Effect reported in Table 4 for the Saria et al. (2013) (S13) Euler
Pole are also shown.





**Table B3**

| Parameter | Component of regional extension $v$ (mm/yr) | $\varphi$ | $\delta$ | C1 | C2 | L |
|---|---|---|---|---|---|---|
| Component of regional extension | - | | | | | |
| $v$ (mm/yr) | 0.00 | - | | | | |
| $\varphi$ | 0.00 | 0.00 | - | | | |
| $\delta$ | 0.00 | 0.00 | 0.00 | - | | |
| C1 | 0.00 | 0.00 | 0.00 | 0.00 | - | |
| C2 | 0.00 | 0.00 | 0.00 | 0.00 | 0.00 | - |
| L | 0.00 | 0.00 | 0.00 | 0.00 | 0.00 | 0.00 | - |

Table B3: Results of parameter-parameter interaction effects on sensitivity analysis using approach
outlined in Eq. B3. All results are zero (to two decimal places), and so there are no parameter-
parameter effects in the sensitivity analysis outlined here.