# Peer review of "A systems-based approach to parameterise seismic hazard in regions with little historical or instrumental seismicity: The South Malawi Active Fault Database 3 4 Jack N. Williamsa\*, Hassan Mdalab, Åke Fagerenga, Luke N.J. Wedmorec, Juliet Biggsc, Zuze 5 Dulanyad, Patrick Chindandalie, Felix Mphe"

_Solid Earth, 2020_

## Referee Comment (RC1) · Richard Styron (Referee) · 19 Aug 2020

The paper by Williams et al. provides a high-quality map of active faults in southern Malawi, and presents a clever method for partitioning regional deformation rates onto the rift structures, with a thorough exploration of the uncertainties.

The work (the mapping, rate estimation, and manuscript) are executed competently, and there is nothing that is strictly incorrect, although some topics would benefit from a bit more explanation if not revision. The mapping is quite high quality, and is the most

solid contribution made here.

Although the authors may not want to do this, I think that the work could benefit from being split into two different, shorter papers: one that presents the fault mapping and discusses it in a bit more detail and context (although not too much more), and another that presents the parameter estimates, ideally as a part of a PSHA. (Note that I am not asking that this be done for major revisions–it's just something to consider doing.)

Major issues:

Separation of data and estimates

The first issue is that there is no apparent separation in the fault data and attributes between observation (and associated interpretation firmly based in observations) and rough estimation based on little to no data. Although the authors are helpfully conforming to the schema laid out by the GEM Faulted Earth (GFE) project (e.g. Christophersen et al. 2015), I believe that the GFE is intended to hold observations or measurements, rather than the estimates made in this project. For example, recurrence intervals would be derived from paleoseismology rather than calculated from the slip rate and assumed magnitudes. Only considering measured recurrence intervals makes the recurrence intervals independent of assumptions of earthquake magnitude, scaling relations, or other factors. Christophersen et al (2015) state that the database is often sparse where observations don't exist.

The coupling of the fault traces (which are observations or data, as far as I am concerned, even if there is an interpretive component) with the parameter estimations crosses a traditional boundary. Typically, the fault data are considered to be somewhat objective and immutable (though subject to revision) while the derivation of earthquake rates is considered to be part of the modeling process and often done while making fault sources for a PSHA. Model results are a bit more subjective and mutable, as they are dependent on assumptions that are explored during the modeling process and/or are project-dependent. A good modeler will have a process for testing and refining some of these assumptions (i.e. magnitude scaling relations or slip partitioning) through comparisons with instrumental seismicity or other observations. However the observed data are usually not revised to improve a data-model fit. A user who wants to incorporate this dataset into a seismic hazard model, but who may not agree with some of the model assumptions used here (i.e. scaling relations, magnitude ranges, or the style of partitioning between internal and border faults) may have a hard time knowing what to keep and what to discard, without reading a long paper. This can be a big challenge for the many seismic hazard modelers who do not have a great facility with the English language. Similarly, if this data were incorporated into other fault databases, the end user may not be able to cleanly separate data from model results.

This does not mean that what the authors have done is wrong or necessarily needs to be changed. It is just to raise their awareness of a potential concern (that these estimates may be confused for observations) and that many hazard modelers would prefer to redo the estimation rather than rely on these results.

I am not sure of the best course of action. If it were me doing the work, I would separate these processes and release both 'only data' and 'data plus estimates' datasets. I would also consider publishing them independently, and perhaps incorporating the rate estimation work into a PSHA rather than going part way as is done here. But there is no 'right' or scientifically optimal decision here, and a lot depends on the particular circumstances of the authors.

Another possibility is to keep the parameter estimation through the slip rate estimation but stop there, which would avoid the problems of choosing a magnitude-frequency distribution, estimating the seismogenic thickness of the crust, etc. In a typical project, these tasks are often done by the hazard modeler rather than the geologist who prepares the fault data up through slip rate estimations.

I can state that as a fault data compiler, I am a bit hesitant to bring any of the estimated parameters into the GEM Global Active Faults Database, as they are too poorly constrained and data-limited, and I don't want users to confuse them for measurements.

Estimation of uncertainty:

The second issue is that the logic tree framework used to propagate uncertainties and explore the parameter space is perhaps more complex than it needs to be based on the lack of input data. It is a clever method and there seems to be nothing incorrect in the implementation, but I question the wisdom of using it southern Malawi where there is essentially no data to feed in. The old saying in modeling is "garbage in, garbage out"; in this case it's more like "nothing in, nothing out" (I am not suggesting the work is garbage!). The exercise seems to simply quantify the obvious, that each fault slips somewhere between 0-5 mm/yr. I am not sure that it provides much value. The further work, estimating recurrence intervals, has larger theoretical issues (discussed below) in addition to adding several more layers of uncertainty into the results. It could easily be removed from the database (though perhaps kept in the paper for discussion).

As a subordinate issue, I don't think a logic tree framework is really the most appropriate method of propagating uncertainty as used; it is more appropriate when the parameters that make up the branches in the tree are discrete variables with a few choices, rather than continuous random variables. For example, an appropriate use of logic trees is to consider different scaling relationships. With continuous random variables (i.e., extension rate or dip), the use of unweighted logic trees considers the lower, mid, and high values to all have equal probability. Do the authors consider this to be the case? Do the authors believe that the resulting low, mid and high values are equally probable? Even if the inputs are all equal, if there are no correlations between the different parameters, the middle values should be more probable (see for example the Central Limit Theorem).

In my opinion, a more appropriate method for representing the uncertainty in the results (i.e., slip rates or recurrence intervals) would be to define distributions for each continuous random variable (i.e., dip or total geodetic extension at that latitude) and

then randomly sample from these distributions, and then characterize the resulting distributions for the results parameters. This is a simple Monte Carlo method. The major strength of this method is that the sampling will cover far more of the parameter space than a coarse 'low/med/high' sampling method. It is also quite trivial to introduce distributions for each parameter that may reflect prior knowledge (i.e., a PDF of regional dips based on focal mechansisms or structural measurements).

One way to think of this is that the representation of uncertainty in the model should reflect the real uncertainty of the parameter. Continuous variables should be represented through continuous distributions, while discrete variables (i.e. the choice of scaling relationships) should be represented through discrete distributions (i.e. lists or arrays, perhaps weighted).

The strategy employed here does a good job of defining the absolute range of the results based on the inputs, but a worse job of defining the central values (broadly like the mean and one standard deviation rather than three standard deviations). If the authors believe this is the better choice, that is fine, but I would like to hear their arguments.

The calculations of recurrence rate:

The authors choose to calculate recurrence rates under the assumption that all of the seismic moment that accumulates on each fault is released during earthquakes of identical magnitude. This is essentially the "characteristic earthquake hypothesis" which featured quite prominently in mid-late 20th century paleoseismology and PSHA but was always quite contentious (for example see "Characteristic Earthquake Model, 1884-2011, RIP" by Kagan et al. 2012, Seismological Research Letters). This hypothesis is believed by fewer and fewer scientists with each passing year, as our observations of variable rupture segmentation and per-event displacement accrue. The few remaining national-level PSHA models that still use a 'pure' characteristic earthquake model (not a distribution that includes aleatoric variability) do so primarily because it

simplifies time-dependent hazard analysis.

The modern state of practice is to consider a range of earthquake sizes on each fault, and to distribute moment throughout the range of earthquake sizes by specifying the relative frequencies of different magnitudes of earthquakes, and then calculating the absolute frequencies through moment rate balancing. The canonical reference for this is Youngs and Coppersmith (1985 BSSA), which provides equations for multiple magnitude-frequency distributions. GEM's Open-Quake Engine and OQ Model Building Toolkit also has some Python code for this purpose, if the authors are interested in using or studying a functional implementation (https://github.com/gem/oq-engine/tree/master/openquake/hazardlib/mfd; https://github.com/GEMScienceTools/oq-mbtk/blob/master/openquake/mbt/tools/fault_modeler/fault_modeling_utils.py#L2

If the authors favor the pure characteristic earthquake hypothesis, then they should provide some supporting arguments. Otherwise they may either calibrate the magnitude-frequency distributions, or simply drop this part of the estimation procedure (even if they retain the estimates up through the slip rate calculations).

Minor issues:

Data license, distribution and updates:

One of the promises of 21st century science is that new technologies enable rapid and low-friction sharing, integrating, and updating of data. However, it raises some new topics that have been heretofore ignored by most. The first is the license of the data. As the creators of a nice dataset, the authors are entitled to specify the terms and conditions under which others may use it. A good "open-data" choice is the Creative Commons Attribution license, which is what the articles published by the EGU/Copernicus journals use.

However, the authors may wish to specify a different license, such as a non-commercial license (meaning that it can't be sold or used for other commercial purposes), a sharealike license (meaning that any modifications to the data, which are allowed by the Creative Commons licenses, must be redistributed under the same conditions), or various others. There are also more and less restrictive licenses, but these may start to conflict a bit with the release of the data in this journal.

It may sound like a bit of boring lawyer stuff, but it's very important to many of us that deal with others' data regularly. If the authors want the data to be most useful, please explicitly state what the license is, so the potential users can have some clarity about what they can or can't do with it. It's an easy process: just put a 'license.txt' file in the zip with the GIS data.

Similarly, the data will probably see a lot more use if it is easy to get to, and in a place where it's easy for the authors to update. The easiest here is using GitHub (github.com) which has turned into the default small data distribution channel for many, including the GEM Global Active Faults Database. GitHub, or other similar services such as GitLab, provide a great platform for licensing, distributing and updating data, in a way that makes the history of the data transparent to the users by being integrated with a version control system.

Something else to consider is whether the authors would welcome updates or extensions to the mapping (and perhaps parameter estimation). It may be that other users who are interested or have some need for a fault database over a wider area than just that covered in this dataset, and may want to expand along strike. This is the kind of collaborative science that is quite easy to do now, especially with services such as GitHub, but I don't think the academic publication process, and allotment of credit (citations etc.) has caught up. Nevertheless, if the authors support this (in principle, no need to blindly accept changes) they could write a sentence or two in the manuscript or in a text file with the data describing this.

Publication of code to perform parameter estimation:

I think that by default, any code used in a scientific work should be published with the

paper. This would definitely include any code used to perform the parameter estimation (one assumes it wasn't done on a hand calculator). There may be some extenuating circumstances where publication of code isn't a good idea, but this would involve prior intellectual property restrictions or something. I wouldn't consider messy scripts to be exempted here. Detailed inspection of methods and reproducibility is central to the scientific process, and code is perhaps the most perfect form of scientific inquiry that allows for this. Please publish the code, even if it's a messy script of zip file of them. (I also think that EGU/Copernicus asks for this but I could be wrong.)

Line edits:

Line 5 (and throughout manuscript): Superscripts are formatted as subscripts. This is particularly annoying with exponents.

Line 18: All seismically active areas on earth have instrumental records much shorter than the 'repeat times' of the larger earthquake produced in these regions (hundreds to tens of thousands of years).

Line 56: Actually, active fault databases have been developed for close to all seismically active regions; the GEM Global Active Faults database is referenced elsewhere in the paper, which has global coverage. Some areas (like the EARS) need better mapping and slip rate measurements, but active fault data does exist.

Line 79 (and elsewhere): I would be more careful with the suggestions that PSHA based on instrumental seismicity is likely to underestimate seismicity in moderately low strain rate regions. The cited references don't do a good job of backing this assertion up, which is not surprising as many earthquake scientists who are not actively involved in PSHA overestimate their knowledge of it (Stein being a prime offender). The justification that this study will provide better constraints on earthquake rates than PSHA models that incorporate instrumental seismicity (which, when done correctly, is quite capable of dealing with incomplete catalogs) is cringe-inducing in light of the extremely poor constraints on earthquake rates produced in this work.

Line 160: The GEM Global Active Faults Database has now been through peer review, and the citation should be changed to Styron, Richard, and Marco Pagani. "The GEM Global Active Faults Database." Earthquake Spectra, Aug. 2020, doi:10.1177/8755293020944182.

Line 324: Note that the GEM neotectonics database is part of the GEM Faulted Earth project, which ended around 2015, and is quite distinct from the GEM Global Active Faults Database (Styron and Pagani, 2020). Please more explicitly refer to the earlier neotectonics database as the Faulted Earth database for clarity.

Line 325: It is worth noting (but not necessarily changing the fault data or the manuscript) that the hierarchy developed by Christophersen et al (2015) as part of the GFE is a bit contentious and has been abandoned at GEM. The newer Global Active Faults database does not incorporate it, as I decided it was too cumbersome and instead chose a 'flat' system where the 'trace' units in the GFE system would be mapped as a single, continuous trace (in most cases it's somewhat obvious that the traces connect in the bedrock regardless of surface expression, as most faults in these databases have a kilometer or more displacement which can't geologically drop to zero where the traces don't quite join). This simplifies the mapping, drastically reduces the file size of the fault database, and makes for easier hazard modeling as the maximum earthquake can be calculated from the area of a single feature rather than manual joining of multiple features. Many other institutions, such as the USGS, are considering following suit if they have not done so already–the simplicity of the system allows for easier updates and more automated pipelines for incorporating faults into PSHA.

Line 383: The calculations here are another instance of what many would consider to be modeling decisions rather than something incorporated directly into fault databases.

Line 639: This is not in any way a test of the results. The comparison of very broadly estimated rates with data-based estimates for faults hundreds of kilometers away does not meaningfully indicate the validity of the rate estimates here.

Line 651: This is also not a very meaningful comparison. The reasons that the projected date of initiation of the rifting derived from geodetic data (an extrapolation of 1,000,000x) don't match geologic data are manifold to the point where it may not be worth discussing; consider removing this paragraph.

Line 669: Why exactly are only half of the 128 parameter combinations considered in this? How were these 'carefully selected' in a way that is not cherry picking? Computers are pretty fast these days and if this analysis is worth doing (it is interesting) it is worth doing with all of the combinations. Surely it wouldn't take more than a few seconds.

Lines 691-730: I don't think these bits of discussion add anything to the paper, and removing them would improve the focus of the paper.

The digression about fault growth is interesting but not very relevant. The second paragraph has some sloppy scholarship; the 30-60 km long normal faults here are not at all on the long side of normal faults worldwide, as is clearly evident in the GEM Global Active Faults Database which is cited a few times. The Jackson and White reference is very out of date.

The paragraph on seismic risk is important but could be tightened up and placed in the introduction, where it is more appropriate. The next paragraph, comparing the lengths of faults in this database to earthquakes also suffers a bit because it compares a small number of global earthquakes to a local fault database, which isn't a good point of comparison (longer normal faults exist in several orogens and generally have similarly slow slip rates, i.e. the Basin and Range in the US).

Line 758: This paragraph is troubling. It seems to discourage others from attempting to collect real data to use in PSHA, though there is no reason to think that the rough estimates provided in this work are superior to field measurements.

Line 798: The probability distributions listed here describe aleatory variability in recurrence, but the topic under discussion is epistemic uncertainty. In this case these are not comparable.

---

## Referee Comment (RC2) · FOLARIN KOLAWOLE (Referee) · 23 Aug 2020

Title: A systems-based approach to parameterise seismic hazard in regions with little historical or instrumental seismicity: The South Malawi Active Fault Database Manuscript Number: se-2020-104 Authors: Williams et al. ————————————— ———————————————————————————————————-

This manuscript presents a new systematic approach useful for parametrizing seismic hazards in areas with limited instrumental seismicity. The study was carried out in the

southern part of Malawi, and documents the large faults that are capable of accommodating medium-large magnitude earthquakes in the region, as well as the attributes of these faults that are relevant for the hazard analysis. Also, the study discusses both the seismic hazard and tectonic implications of the results, as well as the uncertainties in the estimates. I believe that this approach is great and useful in active plat boundary settings where there is poor earthquake monitoring infrastructure. Such settings abound in several continents, and seismic instrumentation is expensive; thus, necessitating a need for creative, less expensive approaches as presented in this study. The manuscript is well written and easy to read. I believe that this manuscript contains material fit for publication in EGU Solid Earth. However, I believe this manuscript could appropriate for publication in the journal after moderate revisions to the paper. I'm recommending moderate revision because of the issues I consider to be major flaws in the interpretation of the tectonic domains and associated structural elements in the study area, which directly impact either the input data or specific features of the implementation of the analysis performed in the study. Here below, are the 7 major issues I have with the manuscript, and 2 comments/questions that I think the authors could consider incorporating into the discussion part of the manuscript. Also, I made comments in different parts of the text that are either minor corrections/comments, or are related to the major issues stated below (see attached an annotated pdf).

Regards, Folarin Kolawole

Major Issue 1: The interchanging use of "southern Malawi" and "southern Malawi Rift". These two terms should not be used interchangeably in this text as it can bring confusion. "Southern Malawi" refers to a geopolitical region that hosts rift segments of different tectonic affiliations; whereas, "Southern Malawi Rift" refers to the southernmost segment of the Malawi Rift which includes only the Makanjita Trough & Malombe Graben (bifurcation around the Shire Horst), and the Zomba Graben. This interchanging use occurs at too many parts of the manuscript, so I decided to just mention it here instead of commenting on it in the text (attached pdf). This issue also leads to and is related to my Major Issue 2...see below.

Major Issue 2: Definition of principal grabens of the southern Malawi Rift. The authors identified the graben "Lower Shire Graben" as a principal graben of southern Malawi Rift (pg 19 lines 458-459). I have issue with the characterization of this graben as a tectonic element of the Malawi Rift. This is very misleading as this Lower Shire Graben is a sub-basin in the Shire Rift, not the Malawi Rift (Castaing, 1991). I understand that this graben is located within the Malawi geopolitical boundary, whereas most of the other sections of the Shire Rift are located in Mozambique. However, geopolitical location does not automatically make this graben a part of the Malawi Rift. Moreover, the Shire Rift has a distinctly different structure, orientation, and tectonic history from those of the southern Malawi Rift. Shire Rift is a multiphase rift basin (Mesozoic-Cenozoic; Castaing, 1991), whereas, southern Malawi Rift is Late Cenozoic (e.g., Wedmore et al., 2019; Scholz et al., 2020). Infact, exposed basement highs separate the Zomba Graben (which is at the southernmost tip of the Malawi Rift) from this Lower Shire Graben and the other sections of the Shire Rift. Therefore, in order not to confuse a reader, I'll suggest that the authors use the term "southern Malawi" in the context of describing the location of the 'Lower Shire graben' (i.e. use geographical description), rather than the term "southern Malawi Rift".

Major Issue 3: The descriptions of the "Makanjira Graben" in the manuscript and the modelling done in Fig.A3a shows that the authors consider that term to incorporate both the Makanjira Trough and Malombe Graben. The Malawi Rift bifurcates around the Shire Horst into these two segments and further south, they link-up and transition into the Zomba Graben. The Makanjira Trough is bounded to the west by Chirobwe-Ncheu Fault, and to the East by Shire Horst, whereas the Malombe Graben is bounded to the west by the Malombe Fault and to the east by the Mwanjage Fault. The surface+subsurface structure of this section of the Malawi Rift (Lao-Davila et al., 2015) shows that the Malombe Graben has a greater hanging wall subsidence and thus, border fault offset than the Makanjira Trough. Here are the evidences: 1.) the floor of the

Makanjira trough is mostly dominated by exposed basement, whereas, the Malombe Graben is relatively better developed graben structure with a wider-spread sediment accumulation and even a lake development at the foot of its border fault. The zone of sediment accumulation on the northern half of the Makanjira trough is associated with subsidence along the southern extension of the N-S trending eastern border fault of the Nkhotakota Segment of the Malawi Rift (for location of Nkhotakota Segment see Lao-Davila et al., 2015; for the described subsidence and fault location, see Fig.5b of Scholz et al., 2020). 2.) the floor of the Makanjira half-graben is at a higher elevation compared to that of the Malombe Graben, indicating that subsidence is most-likely greater in the Malombe Graben. For reference see the across-rift profiles in Figs.4L-4M of Lao-Davila et al. (2015). I am guessing that the authors consider that because the Chirobwe-Nchue Fault has a higher footwall elevation/escarpment along the rift section, therefore, it must have the largest throw. If that is the consideration upon which the border fault definition and Fig.A3a model are based, I refer the authors to the Rukwa Rift where border fault footwall elevation/uplift is not representative of the subsurface fault throw (Morley et al., 1999). Thus, based on the observed geologic structure, I think the model in Fig.A3a is problematic because it ignores the presence of the fault with the larger offset and hanging wall subsidence at the so-called "Makanjira Graben". Also, the model assumes the Chirobwe-Ncheu to has the greatest offset/hanging wall subsidence along the profile which is not representative of the distribution of subsidence across this section of the rift (Figs. 4L-4M in Lao-Davila et al., 2015). Therefore, in my opinion, if possible, I think this model needs to be revised. If impossible due to modelling limitations, then it should be stated.

Major Issue 4: Age of the Thyolo Fault and modelling of strain in Lower Shire graben (Fig. A3c and pg 37 lines 895-896, pg 38 lines914-915). The authors suggest that the Thyolo Fault is Karoo age. There is no evidence suggesting that there exists karoo-age sedimentary or volcaniclastic deposits on the hanging wall of the Thyolo Fault (Habgood, 1963; Habgood et al., 1973). Mesozoic activity along the Thyolo Fault would require subsidence of its hanging wall and creation of accommodation space for the deposition of volcanic and sedimentary sequences. Both Habgood (1963) and Castaing (1991) suggested that the Mwanza-Namalmbo Fault system is the eastern border fault of the Mesozoic Shire Rift. Castaing (1991) suggested that the Thyolo Fault is Cenozoic, bounding the currently active eastern domain of the Shire Rift. Therefore, I think this idea of Thyolo Fault being a Karoo fault needs to be revised. . .except the authors provide data showing the presence of Mesozoic deposits on the hanging wall of the Thyolo Fault.

Major Issue 5: Definition of "border fault" in southern Malawi (Fig. 2a). The Lisungwe Fault, Malombe Fault, and Mwanza Faults are excluded from the 'border fault' definition and I am not particularly sure why. This is an issue for me, particularly because the structure of the basins point directly to the essence of these faults. For example, the Malombe Fault is the principal border fault of the Malombe Graben, not the Mwanjage Fault which you've assigned as the main border fault. Even the distribution of the amplitudes and wavelengths of the magnetic fabrics beneath the Malombe Graben in Fig. 2c (Laõ-Dávila et al., 2015) clearly shows that the hanging wall of the Malombe Fault has significantly larger subsidence than that of the Mwanjage Fault. Also, the Mwanza Fault is a major border fault of the NW half of the Shire Rift (as shown in the maps in Figure 2). There is also evidence that the exposed segment of the Mwanza fault has been reactivated in the Cenozoic given by accumulation of Quaternary sediments on its hanging wall (Habgood, 1963).

Major Issue 6: Related to "Issue 4" above, the authors classified Namalambo Fault as an active Fault (e.g., Figs. 1b & 2). Namalambo Fault cannot be classified as an East African Rift System Fault because there is no evidence supporting its Cenozoic reactivation (see Bloomfield, 1958; Habgood, 1963; Castaing, 1991). The mentioned geological reports specifically stated that there is no Quaternary sediment deposition on top of the karoo sedimentary units at the base of its scarp, suggesting it has not been reactivated in the Cenozoic. Also, this fault does not satisfy the criteria stated by the author in Pg11 lines 260-267. Are the authors including it because they're assuming that it could be reactivated sometime in the future? If yes, I think this should be stated in the relevant figure captions and in the manuscript (particularly because this fault is a prominent fault in the area, and could be confusing to a reader without this additional information).

Major Issue 7 (minor): A 'declaration' that I think is not clearly made in the set-up of the premise of the manuscript is that the parameterization approach focuses on tectonically active continental settings. I do not think that the authors imply that the approach is applicable to relatively more stable intraplate settings where much lower strain rates generally abound and potentially dangerous faults are buried, although some of those areas could be seismically active (e.g., intraplate induced seismicity). Therefore, I'll suggest that the authors state this clearly, at least in the abstract and introduction sections of the paper. I noticed that in different parts of the text, it is subtly implied with phrases relating to plate boundary, interplate setting etc., however, I think it will be beneficial to the reader if this is stated clearer from the onset.

Question/Comment 1: Pg29 lines 703-707. The authors highlighted the anomalously large seismogenic thickness of the southern Malawi area, with continuous 30-60 km-long fault sections. Crustal thickness map of southern Malawi (Njinju et al., 2019a) shows that an unusually thick crust dominates the area. In addition, heat flow map of the same area (Njinju et al., 2019b) shows an anomalous thermal gap in the area. Both of these have been associated it with an eastern extension of the Niassa Craton. Do you think that there is a possibility that these have an impact on the observed seismogenic thickness?

Question/Comment 2: It is well-known that the patterns of seismogenic fault reactivation are influenced by the frictional stability of faults. Asides from strain rate and lithology/mineralogical composition, another important factor that influence the frictional stability is geothermal gradient/heat flow. Well-constrained heat flow & geothermal gradient maps of southern Malawi (Njinju et al., 2019b) show interesting thermal anomalies within the areas analyzed in this study. I am curious as to how the heat flow distribution in the area may affect the results/conclusions of this study.

References: Bloomfield, K., 1958. The geology of the Port Herald area. Malawi Geological Survey Department Bulletin 9, Zomba. Castaing, C. (1991), Post-Pan-African tectonic evolution of South Malawi in relation to the Karroo and recent East African rift systems. Tectonophysics, 191(1-2), pp.55-73. Fonseca, J.F.B.D., Chamussa, J., Domingues, A., Helffrich, G., Antunes, E., van Aswegen, G., Pinto, L.V., Custódio, S. and Manhiça, V.J., 2014. MOZART: A seismological investigation of the East African Rift in central Mozambique. Seismological Research Letters, 85(1), pp.108-116. Gawthorpe, R.L. and Leeder, M.R., 2008. Tectono-sedimentary evolution of active extensional basins. Basin Research, 12(3-4), pp.195-218. Habgood, F. (1963). The geology of the country west of the Shire River between Chikwawa and Chiromo. Malawi Geological Survey Department Bulletin No. 14, Zomba. Habgood, F., Holt, D. N., and Walshaw, R.D., 1973. The Geology of the Thyolo Area. Malawi Geological Survey Department Bulletin No. 22, Zomba. Morley, C.K., Wescott, W.A., Harper, R.M. and Cunningham, S.M., 1999. Geology and geophysics of the Rukwa Rift. Geoscience of Rift Systems-Evolution of East Africa. AAPG Studies in Geology, 44, pp.91-110. Mueller, C.O. and Jokat, W., 2019. The initial Gondwana break-up: a synthesis based on new potential field data of the Africa-Antarctica Corridor. Tectonophysics, 750, pp.301-328. Njinju, E. A., Atekwana, E. A., Stamps, D. S., Abdelsalam, M. G., Atekwana, E. A., Mickus, K. L., et al. (2019a). Lithospheric structure of the Malawi Rift: Implications for magma-poor rifting processes. Tectonics, 38. https://doi.org/10.1029/2019TC005549. Njinju, E.A., Kolawole, F., Atekwana, E.A., Stamps, D.S., Atekwana, E.A., Abdelsalam, M.G. and Mickus, K.L. (2019b). Terrestrial heat flow in the Malawi Rifted Zone, East Africa: Implications for tectono-thermal inheritance in continental rift basins. Journal of Volcanology and Geothermal Research, 387, p.106656. Salman, G. and Abdula, I., 1995. Development of the Mozambique and Ruvuma sedimentary basins, offshore Mozambique. Sedimentary Geology, 96(1-2), pp.7-41. Scholz, C.A., Shillington, D.J., Wright, L.J., Accardo, N., Gaherty, J.B. and Chindandali, P., 2020. Intrarift fault fabric, segmentation, and basin evolution of the Lake Malawi (Nyasa) Rift, East Africa. Geosphere. Wedmore, L. N. J., Biggs, J., Williams, J. N., Fagereng, Å., Dulanya, Z., Mphepo, F. and Mdala, H.: 1571 Active fault scarps in southern Malawi and their implications for the distribution of strain in 1572 incipient continental rifts, Tectonics, e2019TC005834,

Please also note the supplement to this comment:
https://se.copernicus.org/preprints/se-2020-104/se-2020-104-RC2-supplement.pdf

**Supplement:**

[revised manuscript text omitted]

---

## Author Comment (AC1) · 21 Sep 2020

REPLY TO REVIEWER 1

In this document, we have replied to comments on our *Solid Earth* Discussion article *'A systems-based systems-based approach to parameterise seismic hazard in regions with little historical or instrumental seismicity: The South Malawi Active Fault Database'* (SE-2020-104) by Reviewer 1 (Richard Styron). In italics are the reviewer's comments, which have been copied from the original review, and in blue cambria text are our replies to them with how we would like to incorporate them into a revised manuscript to submit to *Solid Earth.*

Kind regards
Jack Williams (on behalf of all coauthor's)

*The paper by Williams et al. provides a high-quality map of active faults in southern Malawi, and presents a clever method for partitioning regional deformation rates onto the rift structures, with a thorough exploration of the uncertainties. The work (the mapping, rate estimation, and manuscript) are executed competently, and there is nothing that is strictly incorrect, although some topics would benefit from a bit more explanation if not revision. The mapping is quite high quality, and is the most solid contribution made here.*

*Although the authors may not want to do this, I think that the work could benefit from being split into two different, shorter papers: one that presents the fault mapping and discusses it in a bit more detail and context (although not too much more), and another that presents the parameter estimates, ideally as a part of a PSHA. (Note that I am not asking that this be done for major revisions–it's just something to consider doing.)*

We thank the reviewer for their suggestion, which we have considered very seriously. Our preference is to keep the mapping and parameter estimates together in one paper, as we feel they are inherently linked and that a strength of the current study is the fact that it spans traditional discipline boundaries, Nevertheless, we see the importance of more distinctly separating between our observations and interpretations. As outlined below with respect to major issue #1, we will do this by providing two distinct GIS file databases, the 'South Malawi Active Fault Database' and 'South Malawi Seismogenic Sources Database'. In this way, the reader will be more able to separate between the fault mapping and the parameter estimates, and any user has a choice between adapting the fault mapping only or also including our parameter estimates.

**Major issues:**

**1. Separation of data and estimates**

*The first issue is that there is no apparent separation in the fault data and attributes between observation (and associated interpretation firmly based in observations) and rough estimation based on little to no data. Although the authors are helpfully conforming to the schema laid out by the GEM Faulted Earth (GFE) project (e.g. Christophersen et al. 2015), I believe that the GFE is intended to hold observations or measurements, rather than the estimates made in this project. For example, recurrence intervals would be derived from paleoseismology rather than calculated from the slip rate and assumed magnitudes. Only considering measured recurrence intervals makes the recurrence intervals independent of assumptions of earthquake magnitude, scaling relations, or other factors. Christophersen et al (2015) state that the database is often sparse where observations don't exist.*

*The coupling of the fault traces (which are observations or data, as far as I am concerned, even if there is an interpretive component) with the parameter estimations crosses a traditional boundary. Typically, the fault data are considered to be somewhat objective and immutable (though subject to*

*revision) while the derivation of earthquake rates is considered to be part of the modeling process and often done while making fault sources for a PSHA. Model results are a bit more subjective and mutable, as they are dependent on assumptions that are explored during the modeling process and/or are project-dependent. A good modeler will have a process for testing and refining some of these assumptions (i.e. magnitude scaling relations or slip partitioning) through comparisons with instrumental seismicity or other observations. However the observed data are usually not revised to improve a data-model fit. A user who wants to incorporate this dataset into a seismic hazard model, but who may not agree with some of the model assumptions used here (i.e. scaling relations, magnitude ranges, or the style of partitioning between internal and border faults) may have a hard time knowing what to keep and what to discard, without reading a long paper. This can be a big challenge for the many seismic hazard modelers who do not have a great facility with the English language. Similarly, if this data were incorporated into other fault databases, the end user may not be able to cleanly separate data from model results.*

*This does not mean that what the authors have done is wrong or necessarily needs to be changed. It is just to raise their awareness of a potential concern (that these estimates may be confused for observations) and that many hazard modelers would prefer to redo the estimation rather than rely on these results. I am not sure of the best course of action. If it were me doing the work, I would separate these processes and release both 'only data' and 'data plus estimates' datasets. I would also consider publishing them independently, and perhaps incorporating the rate estimation work into a PSHA rather than going part way as is done here. But there is no 'right' or scientifically optimal decision here, and a lot depends on the particular circumstances of the authors.*

*Another possibility is to keep the parameter estimation through the slip rate estimation but stop there, which would avoid the problems of choosing a magnitude-frequency distribution, estimating the seismogenic thickness of the crust, etc. In a typical project, these tasks are often done by the hazard modeler rather than the geologist who prepares the fault data up through slip rate estimations. I can state that as a fault data compiler, I am a bit hesitant to bring any of the estimated parameters into the GEM Global Active Faults Database, as they are too poorly constrained and data-limited, and I don't want users to confuse them for measurements.*

As noted above, to follow the advice to release both 'only data' and 'data plus estimates' datasets, we propose that in the resubmitted manuscript we will describe and include two separate GIS databases of faults in South Malawi:

1. The South Malawi Active Fault Database (SMAFD): this database will incorporate the objective mapping and geomorphic 'trace' attributes included in the current version of the SMAFD. In addition, individual GIS features will represent faults (see minor comment for Line 325). In this way, the SMAFD will conform to the 'only data' dataset requested by the reviewer and will also be readily comparable to the newly published GEM Global Active Fault Database (Styron and Pagani, 2020).

2. The South Malawi Seismogenic Source Database (SMSSD): this database will incorporate the modelling derived attributes included in the original version of the SMAFD that are required to turn the mapped faults into earthquake sources for PSHA (e.g. fault segmentation, fault width, slip rates, earthquake magnitudes, and recurrence intervals). Here, individual GIS features will represent fault segments that can each be considered distinct sources for PSHA. Furthermore, in line with other earthquake source databases (Basili et al., 2008; Field et al., 2014; Stirling et al., 2012) faults in the SMSSD will be mapped as straight lines that connect segment tips, as opposed to truly honouring the surface trace of the faults (see for example the comparison for the Chingale Step Fault in Figure 4b of the discussion manuscript). In this way, the SMSSD will conform to the 'data plus estimate' dataset requested by the reviewer.

By distinctly describing these two datasets in the revised manuscript, whilst also clearly outlining how they are linked, we will thus address the reviewer's concern about how users of these databases may be confused by which fault attributes are objective measurements, and which are model-driven. Furthermore, we will maintain the multidisciplinary aspect of this manuscript, and allow seismic hazard modellers to understand the limitations for geologists investigating active faults in a region where data is sparse and vice versa. Work to include the SMAFD and SMSSD into PSHA in southern Malawi is ongoing, and this will be the topic of a subsequent study.

**2. *Estimation of uncertainty:**

*The second issue is that the logic tree framework used to propagate uncertainties and explore the parameter space is perhaps more complex than it needs to be based on the lack of input data. It is a clever method and there seems to be nothing incorrect in the implementation, but I question the wisdom of using it southern Malawi where there is essentially no data to feed in. The old saying in modeling is "garbage in, garbage out"; in this case it's more like "nothing in, nothing out" (I am not suggesting the work is garbage!). The exercise seems to simply quantify the obvious, that each fault slips somewhere between 0-5 mm/yr. I am not sure that it provides much value. The further work, estimating recurrence intervals, has larger theoretical issues (discussed below) in addition to adding several more layers of uncertainty into the results. It could easily be removed from the database (though perhaps kept in the paper for discussion).*

*As a subordinate issue, I don't think a logic tree framework is really the most appropriate method of propagating uncertainty as used; it is more appropriate when the parameters that make up the branches in the tree are discrete variables with a few choices, rather than continuous random variables. For example, an appropriate use of logic trees is to consider different scaling relationships.*

*With continuous random variables (i.e., extension rate or dip), the use of unweighted logic trees considers the lower, mid, and high values to all have equal probability. Do the authors consider this to be the case? Do the authors believe that the resulting low, mid and high values are equally probable? Even if the inputs are all equal, if there are no correlations between the different parameters, the middle values should be more probable (see for example the Central Limit Theorem).*

*In my opinion, a more appropriate method for representing the uncertainty in the results (i.e., slip rates or recurrence intervals) would be to define distributions for each continuous random variable (i.e., dip or total geodetic extension at that latitude) and then randomly sample from these distributions, and then characterize the resulting distributions for the results parameters. This is a simple Monte Carlo method. The major strength of this method is that the sampling will cover far more of the parameter space than a coarse 'low/med/high' sampling method. It is also quite trivial to introduce distributions for each parameter that may reflect prior knowledge (i.e., a PDF of regional dips based on focal mechanisms or structural measurements).*

*One way to think of this is that the representation of uncertainty in the model should reflect the real uncertainty of the parameter. Continuous variables should be represented through continuous distributions, while discrete variables (i.e. the choice of scaling relationships) should be represented through discrete distributions (i.e. lists or arrays, perhaps weighted).*

*The strategy employed here does a good job of defining the absolute range of the results based on the inputs, but a worse job of defining the central values (broadly like the mean and one standard deviation rather than three standard deviations). If the authors believe this is the better choice, that is fine, but I would like to hear their arguments.*

We can appreciate why the reviewer might question the usefulness of the large uncertainties in our slip rate and recurrence intervals estimates. However, the fact remains that even being able to quantify slip rate estimates of 0-5 mm/yr in southern Malawi represents a significant advance, given that prior to this study, no slip rate estimates have been previously made. Furthermore, as highlighted in the manuscript (Lines 732-741) the large range of values we obtain in our recurrence interval estimates are not unusual compared to other low strain rate regions with limited paleoseismic information (Villamor et al., 2018) and can still be incorporated into PSHA (Hodge et al., 2015). Indeed, these large ranges can be considered as an important outcome of the study, as they can be used to the highlight the parameters that need further study to reduce uncertainty (Sect. 5.3 of discussion manuscript).

We note that logic trees have been used elsewhere to propagate uncertainty in seismic hazard in regions with little paleoseismic data (Villamor et al., 2018). However, we do agree with the reviewer's comments on how we should consider our lower, intermediate, and upper estimates; indeed, we noted in the discussion manuscript that the upper and lower values obtained from the logic tree required an unlikely set of parameter combinations (Lines 589-592), and that treating these values as a continuous variable and assigning a probability distribution function to them would be a more appropriate method of treating them in PSHA (lines 797-801).

A more complex treatment than the intermediate, lower, and upper slip rate and recurrence interval estimates obtained from the logic tree are beyond the scope of this study, but already under consideration for our next paper. In the resubmitted manuscript, we propose that we would have a distinct section on how the fault data included in the SMAFD and SMSSD could be incorporated into PSHA, where we can more explicitly discuss some of the reviewer's excellent suggestions. This would also address Major Issue #3 (see below).

**3. *The calculations of recurrence rate:**

*The authors choose to calculate recurrence rates under the assumption that all of the seismic moment that accumulates on each fault is released during earthquakes of identical magnitude. This is essentially the "characteristic earthquake hypothesis" which featured quite prominently in mid-late 20th century paleoseismology and PSHA but was always quite contentious (for example see "Characteristic Earthquake Model, 1884-2011, RIP" by Kagan et al. 2012, Seismological Research Letters). This hypothesis is believed by fewer and fewer scientists with each passing year, as our observations of variable rupture segmentation and per-event displacement accrue. The few remaining national-level PSHA models that still use a 'pure' characteristic earthquake model (not a distribution that includes aleatoric variability) do so primarily because it simplifies time-dependent hazard analysis. The modern state of practice is to consider a range of earthquake sizes on each fault, and to distribute moment throughout the range of earthquake sizes by specifying the relative frequencies of different magnitudes of earthquakes, and then calculating the absolute frequencies through moment rate balancing.*

*The canonical reference for this is Youngs and Coppersmith (1985 BSSA), which provides equations for multiple magnitude-frequency distributions. GEM's Open-Quake Engine and OQ Model Building Toolkit also has some Python code for this purpose, if the authors are interested in using or studying a functional implementation (https://github.com/gem/oq-engine/tree/master/openquake/hazardlib/mfd; https://github.com/GEMScienceTools/oqmbtk/blob/master/openquake/mbt/tools/fault_modeler/fault_modeling_utils.py#L2379). If the authors favor the pure characteristic earthquake hypothesis, then they should provide some supporting arguments. Otherwise they may either calibrate the magnitude frequency distributions, or simply drop this part of the estimation procedure (even if they retain the estimates up through the slip rate calculations).*

We recognise that we make a large simplification by considering characteristic whole-fault or segment ruptures only, and that fully incorporating the SMAFD into PSHA requires further evaluation of individual faults' magnitude-frequency distribution. We will ensure that this simplification is clearly spelled out in a revision, but also note that by considering both whole fault or segmented fault ruptures, our study already partially recognises that the characteristic earthquake hypothesis does not necessarily apply in southern Malawi, because the faults will not always host similar sized events. Furthermore, some recent studies have noted that the characteristic earthquake hypothesis, and its use in PSHA, is not necessarily 'dead' (Stirling and Gerstenberger, 2018).

It is also worth reflecting that in studies where faults are allowed to host a range of earthquake magnitudes in PSHA, these are built from many years of detailed geological mapping, historical, instrumental and paleo-, seismicity data, and hazard modelling (Basili et al., 2008; Field et al., 2014). Conversely, prior to this study, there has been very little systematic investigation of possible earthquake magnitudes and recurrence intervals at all in southern Malawi. Hodge et al., (2015) is an exception, and this was based on a very limited active fault mapping. So even though the recurrence interval and earthquake magnitude estimates in this study are poorly constrained compared to other countries' seismic hazard assessments, we still consider them to represent a step change in our understanding of southern Malawi's seismic hazard.

Therefore, similar to Major Issue #2 above, our preference is to incorporate the reviewer's comments in a distinct section where we discuss how this study could be used into PSHA.

*Minor issues:*

*Data license, distribution and updates:*

*One of the promises of 21st century science is that new technologies enable rapid and low-friction sharing, integrating, and updating of data. However, it raises some new topics that have been heretofore ignored by most. The first is the license of the data. As the creators of a nice dataset, the authors are entitled to specify the terms and conditions under which others may use it. A good "open-data" choice is the Creative Commons Attribution license, which is what the articles published by the EGU/Copernicusjournals use.*

*However, the authors may wish to specify a different license, such as a non-commercial license (meaning that it can't be sold or used for other commercial purposes), a share-alike license (meaning that any modifications to the data, which are allowed by the Creative Commons licenses, must be redistributed under the same conditions), or various others. There are also more and less restrictive licenses, but these may start to conflict a bit with the release of the data in this journal.*

*It may sound like a bit of boring lawyer stuff, but it's very important to many of us that deal with others' data regularly. If the authors want the data to be most useful, please explicitly state what the license is, so the potential users can have some clarity about what they can or can't do with it. It's an easy process: just put a 'license.txt' file in the zip with the GIS data.*

*Similarly, the data will probably see a lot more use if it is easy to get to, and in a place where it's easy for the authors to update. The easiest here is using GitHub (github.com) which has turned into the default small data distribution channel for many, including the GEM Global Active Faults Database. GitHub, or other similar services such as GitLab, provide a great platform for licensing, distributing and updating data, in a way that makes the history of the data transparent to the users by being integrated with a version control system.*

*Something else to consider is whether the authors would welcome updates or extensions to the mapping (and perhaps parameter estimation). It may be that other users who are interested or have some need for a fault database over a wider area than just that covered in this dataset, and may want to expand along strike. This is the kind of collaborative science that is quite easy to do now, especially with services such as GitHub, but I don't think the academic publication process, and allotment of credit (citationsetc.) has caught up. Nevertheless, if the authors support this (in principle, no need to blindly accept changes) they could write a sentence or two in the manuscript*
*or in a text file with the data describing this.*

We recognise the importance of findable, accessible, interoperable, and reusable data (i.e. the FAIR principles), and thank the reviewer for their advice. Indeed, it is these principles that partly guided our decision to submit this study to *Solid Earth*. As the reviewer recommends, when resubmitting the SMAFD and SMSSD, we will be explicit that this will be licenced under Creative Commons (CC-BY-4.0) Licence (in keeping with the GEM Global Fault Active Database).

**Publication of code to perform parameter estimation:**

*I think that by default, any code used in a scientific work should be published with the paper. This would definitely include any code used to perform the parameter estimation (one assumes it wasn't done on a hand calculator). There may be some extenuating circumstances where publication of code isn't a good idea, but this would involve prior intellectual property restrictions or something. I wouldn't consider messy scripts to be exempted here. Detailed inspection of methods and reproducibility is central to the scientific process, and code is perhaps the most perfect form of scientific inquiry thatallows for this. Please publish the code, even if it's a messy script of zip file of them. (Ialso think that EGU/Copernicus asks for this but I could be wrong.)*

As outlined above with respect to sharing our GIS file, we appreciate the importance of data that follows the FAIR principle. In this case we will include the excel file where our earthquake source estimates are calculated with our resubmitted file.

**Line edits:**

*Line 5 (and throughout manuscript): Superscripts are formatted as subscripts. This is particularly annoying with exponents.*

This was a formatting error when converting the word document to a pdf file, and will of course be corrected in the typeset version of the manuscript

*Line 18: All seismically active areas on earth have instrumental records much shorter than the 'repeat times' of the larger earthquake produced in these regions (hundreds to tens of thousands of years).*

We recognise that the phrasing of this sentence was not precise. The reviewer is correct in what they say, but we would argue that this problem is particularly acute in low strain rate regions where earthquake recurrence intervals may be ~10,000-100,000 years (i.e. the ~100 year long instrumental record covers 0.1-1% of a fault's seismic cycle). as opposed to 100-1000's of years in high strain rate regions (where the instrumental record covers 10-100% of a fault's seismic cycle). We will revise this sentence to reflect this.

*Line 56: Actually, active fault databases have been developed for close to all seismically active regions; the GEM Global Active Faults database is referenced elsewhere in the paper, which has global coverage. Some areas (like the EARS) need better mapping and slip rate measurements, but active fault data does exist.*

We will correct for this in our revised submission, but as the reviewer acknowledges, emphasis that though there is global coverage of active fault maps, the mapping in many regions, including Africa is still patchy, and many of the underlying attributes required to use these faults in PSHA is still lacking.

*Line 79 (and elsewhere): I would be more careful with the suggestions that PSHA based on instrumental seismicity is likely to underestimate seismicity in moderately low strain rate regions. The cited references don't do a good job of backing this assertion up, which is not surprising as many earthquake scientists who are not actively involved in PSHA overestimate their knowledge of it (Stein being a prime offender). The justification that this study will provide better constraints on earthquake rates than PSHA models that incorporate instrumental seismicity (which, when done correctly, is quite capable of dealing with incomplete catalogs) is cringe-inducing in light of the extremely poor constraints on earthquake rates produced in this work.*

The reviewer makes excellent points, and we did not wish to assert that instrumental data provide less constraint than our estimates or that corrections cannot be made to incorporate incomplete catalogs in PSHA. We will therefore clarify and be more careful with our wording in a revised manuscript. In particular, we will more specifically relate it to the EARS, where, to compensate for the incomplete instrumental catalogue, the maximum possible earthquake within each source zone (Mmax) is calculated as the magnitude of the largest event across the entire EAR Western Branch (the 1910 M7.4 Rukwa Earthquake) + 0.5 (Poggi et al., 2017). However, we would suggest that fault mapping in the SMAFD provides much robust constrains on this important parameter.

There are two other points that we also wish to make clearer on the advantage of adding geological and geodetic constraints to hazard models based on instrumental data in Malawi.

Firstly, in the context of the Malawi, if the moment rate from the most complete instrumental record for Malawi (Poggi et al., 2017) is converted to a strain rate using a Kostrov Summation (following the methods outlined in Kostrov, (1974) and Molnar, (1979), a seismogenic crustal thickness of 35 km and faults dipping 53°), the resulting strain rate ($1.4 \times 10^{-9}$ $s^{-1}$) is less than the strain rate from geodesy ($5\text{-}10 \times 10^{-9}$ $s^{-1}$; Stamps et al., 2018) by a factor of > 3. Indeed, in a study specific to northern Malawi, the moment release estimated from geodesy is >25x greater than the moment rate from seismology (Ebinger et al., 2019). Of course, some of this strain could be accommodated aseismically, but there is little evidence that this is the case in Malawi (lines 622-627 in the discussion paper). Hence, at least in the EARS, the point remains that incomplete seismicity catalogues may underestimate the seismic hazard. In the revised manuscript, we will provide more details of these moment rate comparisons.

Secondly, the source zones in the PSHA study of Poggi et al., (2017) covers the Rukwa and Malawi Rifts, and so the seismic hazard (in terms of Peak Ground Acceleration) is essentially uniform across a ~300,000 km² area. It therefore neglects the observation that the strain (and hence seismic hazard) in this part of the rift is likely to be highest around its border faults (Accardo et al., 2018; Shillington et al., 2020; Wedmore et al., 2020) or that the strain rate increases from south to north along this section of the EAR (Saria et al., 2014).

Of course, the instrumental catalogue is a still a very point source of information for characterising seismic hazard, and the most robust PSHA should consider all these sources of information. In the revised submitted fault map manuscript, we will more carefully outline these points.

*Line 160: The GEM Global Active Faults Database has now been through peer review, and the citation should be changed to Styron, Richard, and Marco Pagani. "The GEM Global Active Faults Database." Earthquake Spectra, Aug. 2020, doi:10.1177/8755293020944182.*

We will update this reference in the revised manuscript.

*Line 324: Note that the GEM neotectonics database is part of the GEM Faulted Earth project, which ended around 2015, and is quite distinct from the GEM Global Active Faults Database (Styron and Pagani, 2020). Please more explicitly refer to the earlier neotectonics database as the Faulted Earth database for clarity.*

We thank the reviewer for clarifying the distinction between the GEM Faulted Earth Project and the GEM Global Active Faults Database, and in the revised manuscript will carefully distinguish between these projects.

*Line 325: It is worth noting (but not necessarily changing the fault data or the manuscript) that the hierarchy developed by Christophersen et al (2015) as part of the GFE is a bit contentious and has been abandoned at GEM. The newer Global Active Faults database does not incorporate it, as I decided it was too cumbersome and instead chose a 'flat' system where the 'trace' units in the GFE system would be mapped as a single, continuous trace (in most cases it's somewhat obvious that the traces connect in the bedrock regardless of surface expression, as most faults in these databases have a kilometer or more displacement which can't geologically drop to zero where the traces don't quite join). This simplifies the mapping, drastically reduces the file size of the fault database, and makes for easier hazard modeling as the maximum earthquake can be calculated from the area of a single feature rather than manual joining of multiple features. Many other institutions, such as the USGS, are considering following suit if they have not done so already–the simplicity of the system allows for easier updates and more automated pipelines for incorporating faults into PSHA.*

It is our intention that the fault databases we produce are as consistent with the GEM Global Active Fault database as possible. Therefore, in the revised manuscript we will revise the South Malawi Active Fault Database (SMAFD) data-only GIS file so that each fault is a single GIS 'feature' (though as outlined for our response to Major Comment #1, individual faults can consist of multiple GIS features in the seismogenic source database, SMSSD)

*Line 383: The calculations here are another instance of what many would consider to be modeling decisions rather than something incorporated directly into fault databases.*

As outlined for Major Comment #1, by more clearly distinguishing between our observations and modelling parameters, we will address this comment. In this case, by placing fault width as a parameter in the data + estimates SMSSD, but not the data-only SMAFD.

*Line 639: This is not in any way a test of the results. The comparison of very broadly estimated rates with data-based estimates for faults hundreds of kilometers away does not meaningfully indicate the validity of the rate estimates here.*

We recognise our use of the term 'test' here was misguided but consider this a meaningful 'comparison.' Though these estimates are 100's of km away from southern Malawi, the tectonic setting (i.e. amagmatic continental rift with border faults and intrabasinal faults). and extension rates (1-2 mm/yr; Saria et al., 2014) between these two regions are comparable. In the revised manuscript, we would therefore omit the use of 'test,' and more explicitly justify why this is a meaningful comparison.

Furthermore, we would also argue that when taking a heavily model dependent approach to estimate slip rates and seismic hazard, any 'real-world' constraints that can support our approach, even within an order of magnitude, are useful; it would be worrying, for example, if the intrabasinal faults in northern Malawi had slip rates of <0.01 or >1 mm/yr.

*Line 651: This is also not a very meaningful comparison. The reasons that the projected date of initiation of the rifting derived from geodetic data (an extrapolation of 1,000,000x) don't match geologic data are manifold to the point where it may not be worth discussing; consider removing this paragraph.*

Unlike the comparison above which deals with fault slip rates measured over 75 Ka, we recognise this comparison is much more uncertain and dependent on several poorly constrained parameters. We will remove it in the revised manuscript.

*Line 669: Why exactly are only half of the 128 parameter combinations considered in this? How were these 'carefully selected' in a way that is not cherry picking? Computers re pretty fast these days and if this analysis is worth doing (it is interesting it is worth doing with all of the combinations. Surely it wouldn't take more than a few seconds.*

As described fully in Appendix B, these combinations are not 'cherry-picked' but selected based on a rigorous statistical analysis (Box et al., 1978; Rabinowitz and Steinberg, 1991) such that they provide comparable results to an exploration of all the parameter space. We will clarify this in the main text of the revised manuscript.

*Lines 691-730: I don't think these bits of discussion add anything to the paper, and removing them would improve the focus of the paper. The digression about fault growth is interesting but not very relevant. The second paragraph has some sloppy scholarship; the 30-60 km long normal faults here are not at all on the long side of normal faults worldwide, as is clearly evident in the GEM Global Active Faults Database which is cited a few times. The Jackson and White reference is very out of date.*

We accept the reviewer's comments that this section (Section 6.1 of the discussion paper) detracts from the main focus of the study, and so will remove it from the resubmitted manuscript. This will also create space for further discussion on incorporating the databases into PSHA, without lengthening the paper.

*The paragraph on seismic risk is important but could be tightened up and placed in the introduction, where it is more appropriate. The next paragraph, comparing the lengths of faults in this database to earthquakes also suffers a bit because it compares a small number of global earthquakes to a local fault database, which isn't a good point of comparison (longer normal faults exist in several orogens and generally have similarly slow slip rates, i.e. the Basin and Range in the US).*

As requested by the reviewer, in the resubmitted manuscript we will revise a comparison of fault lengths in Malawi to those of other normal faults from the Global Earthquake Model Global Active Fault Database (Styron and Pagani, 2020), as opposed to specifically just normal fault earthquake ruptures.

*Line 758: This paragraph is troubling. It seems to discourage others from attempting to collect real data to use in PSHA, though there is no reason to think that the rough estimates provided in this work are superior to field measurements.*

It was certainly never our intention to discourage the collection of on-fault data to feed into active fault databases, albeit such data still carries large uncertainties, particularly in low strain rate regions such as Malawi (as outlined at Lines 767-769). In the revised manuscript, we will emphasise that our approach is most appropriate in places like southern Malawi precisely because there is currently no paleoseismic data. Furthermore, a motivation of this study was to identify where future data collection should be targeted, with the collection of paleoseismic data clearly highlighted, along with tighter geodetic constraints, as a priority area.

*Line 798: The probability distributions listed here describe aleatory variability in recurence, but the topic under discussion is epistemic uncertainty. In this case these are not comparable.*

We will correct this statement in the revised manuscript, where there will be a distinct section on incorporating the SMAFD/SMSSD into PSHA.

**References**

[revised manuscript text omitted]

---

## Author Comment (AC2) · 21 Sep 2020

REPLY TO REVIEWER 2

In this document, we have replied to comments on our Solid Earth Discussion article 'A systems-based approach to parameterise seismic hazard in regions with little historical or instrumental seismicity: The South Malawi Active Fault Database' (SE-2020-104) by Reviewer 2 (Folarin Kolawole). In italics are the reviewer's comments, which have been copied from the original review, and in blue cambria text are our replies to them with how we would like to incorporate them into a revised manuscript to submit to Solid Earth.

Kind regards
Jack Williams (on behalf of all coauthors)

*This manuscript presents a new systematic approach useful for parametrizing seismic hazards in areas with limited instrumental seismicity. The study was carried out in the southern part of Malawi, and documents the large faults that are capable of accommodating medium-large magnitude earthquakes in the region, as well as the attributes of these faults that are relevant for the hazard analysis. Also, the study discusses both the seismic hazard and tectonic implications of the results, as well as the uncertainties in the estimates. I believe that this approach is great and useful in active plat boundary settings where there is poor earthquake monitoring infrastructure. Such settings abound in several continents, and seismic instrumentation is expensive; thus, necessitating a need for creative, less expensive approaches as presented in this study. The manuscript is well written and easy to read. I believe that this manuscript contains material fit for publication in EGU Solid Earth.*

*However, I believe this manuscript could appropriate for publication in the journal after moderate revisions to the paper. I'm recommending moderate revision because of the issues I consider to be major flaws in the interpretation of the tectonic domains and associated structural elements in the study area, which directly impact either the input data or specific features of the implementation of the analysis performed in the study. Here below, are the 7 major issues I have with the manuscript, and 2 comments/questions that I think the authors could consider incorporating into the discussion part of the manuscript. Also, I made comments in different parts of the text that are either minor corrections/comments, or are related to the major issues stated below (see attached an annotated pdf).*

*Regards, Folarin Kolawole*

**Major Issues**

*Major Issue 1: The interchanging use of "southern Malawi" and "southern Malawi Rift". These two terms should not be used interchangeably in this text as it can bring confusion. "Southern Malawi" refers to a geopolitical region that hosts rift segments of different tectonic affiliations; whereas, "Southern Malawi Rift" refers to the southernmost segment of the Malawi Rift which includes only the Makanjita Trough & Malombe Graben (bifurcation around the Shire Horst), and the Zomba Graben. This interchanging use occurs at too many parts of the manuscript, so I decided to just mention it here instead of commenting on it in the text (attached pdf). This issue also leads to and is related to my Major Issue 2…see below.*

We acknowledge our interchangeable use of 'southern Malawi' and 'the southern Malawi Rift' will be confusing to readers not familiar with the area, and further adds to the confusion in the various ways that the southern end of the Malawi Rift has been defined previously (Chapola and Kaphwiyo, 1992; Chorowicz and Sorlien, 1992; Ebinger et al., 1987; Laõ-Dávila et al., 2015). In our revised submission, we will carefully outline that the database covers the geopolitical region of southern Malawi, as opposed to the southern 'Malawi Rift' (albeit with the necessity that it will include some faults that extend into Mozambique). This choice also reflects that seismic hazard is typically considered at a national level, and so it makes sense that active fault

databases are defined by national, and not geological, boundaries, with the necessary exception that faults that cross geopolitical boundaries are included.

*Major Issue 2: Definition of principal grabens of the southern Malawi Rift. The authors identified the graben "Lower Shire Graben" as a principal graben of southern Malawi Rift (pg 19 lines 458-459). I have issue with the characterization of this graben as a tectonic element of the Malawi Rift. This is very misleading as this Lower Shire Graben is a sub-basin in the Shire Rift, not the Malawi Rift (Castaing, 1991). I understand that this graben is located within the Malawi geopolitical boundary, whereas most of the other sections of the Shire Rift are located in Mozambique. However, geopolitical location does not automatically make this graben a part of the Malawi Rift. Moreover, the Shire Rift has a distinctly different structure, orientation, and tectonic history from those of the southern Malawi Rift. Shire Rift is a multiphase rift basin (Mesozoic-Cenozoic; Castaing, 1991), whereas, southern Malawi Rift is Late Cenozoic (e.g., Wedmore et al., 2019; Scholz et al., 2020). Infact, exposed basement highs separate the Zomba Graben (which is at the southernmost tip of the Malawi Rift) from this Lower Shire Graben and the other sections of the Shire Rift. Therefore, in order not to confuse a reader, I'll suggest that the authors use the term "southern Malawi" in the context of describing the location of the 'Lower Shire graben' (i.e. use geographical description), rather than the term "southern Malawi Rift".*

As discussed above with reference to Major Issue 1, by explicitly outlining that the South Malawi Active Fault Database (SMAFD) and the South Malawi Seismogenic Source Database (SMSSD) cover the political region of southern Malawi, not the southern 'Malawi Rift,' we will address this comment.

*Major Issue 3: The descriptions of the "Makanjira Graben" in the manuscript and the modelling done in Fig.A3a shows that the authors consider that term to incorporate both the Makanjira Trough and Malombe Graben. The Malawi Rift bifurcates around the Shire Horst into these two segments and further south, they link-up and transition into the Zomba Graben. The Makanjira Trough is bounded to the west by Chirobwe-Ncheu Fault, and to the East by Shire Horst, whereas the Malombe Graben is bounded to the west by the Malombe Fault and to the east by the Mwanjage Fault. The surface+subsurface structure of this section of the Malawi Rift (Lao-Davila et al., 2015) shows that the Malombe Graben has a greater hanging wall subsidence and thus, border fault offset than the Makanjira Trough. Here are the evidences:*

1.) *The floor of the Makanjira trough is mostly dominated by exposed basement, whereas, the Malombe Graben is relatively better developed graben structure with a wider-spread sediment accumulation and even a lake development at the foot of its border fault. The zone of sediment accumulation on the northern half of the Makanjira trough is associated with subsidence along the southern extension of the N-S trending eastern border fault of the Nkhotakota Segment of the Malawi Rift (for location of Nkhotakota Segment see Lao-Davila et al., 2015; for the described subsidence and fault location, see Fig.5b of Scholz et al., 2020).*

2.) *The floor of the Makanjira half-graben is at a higher elevation compared to that of the Malombe Graben, indicating that subsidence is most-likely greater in the Malombe Graben. For reference see the across-rift profiles in Figs.4L-4M of Lao-Davila et al. (2015). I am guessing that the authors consider that because the Chirobwe-Nchue Fault has a higher footwall elevation/escarpment along the rift section, therefore, it must have the largest throw. If that is the consideration upon which the border fault definition and Fig.A3a model are based, I refer the authors to the Rukwa Rift where border fault footwall elevation/uplift is not representative of the subsurface fault throw (Morley et al., 1999). Thus, based on the observed geologic structure, I think the model in Fig.A3a is problematic because it ignores the presence of the fault with the larger offset and hanging wall subsidence at the so-called "Makanjira Graben". Also, the model assumes the Chirobwe-Ncheu to has the greatest offset/hanging wall subsidence along the profile which is not representative of the distribution*

*of subsidence across this section of the rift (Figs. 4L-4M in Lao-Davila et al., 2015). Therefore, in my opinion, if possible, I think this model needs to be revised. If impossible due to modelling limitations, then it should be stated.*

Firstly, please note that the purpose of the profile in Figure A3a is not to construct a realistic across-rift cross-section. Instead, and as conducted by Shillington et al., (2020) for the northern Malawi Rift, it is to explore the range of flexural profiles across the rift that *may* have formed given the significant uncertainties in each variable that we must test. We recognise that this was not clearly indicated in the initial discussion paper, and we will clarify the purpose of this modelling in the revised manuscript.

We address the reviewer's concerns on how we define border faults and intrabasinal faults further below (Major Issue #4). In the context of the Malombe Fault, as the reviewer correctly points out, the Shire Horst divides the rift section that we term the Makanjira Graben. However, though this structure may have been important in the tectonic evolution of the rift (Laõ-Dávila et al., 2015), for the reasons outlined below we do not consider that it strongly influences the current distribution of extensional strain in the Makanjira Graben.

With respect to the reviewer's first set of concerns, we disagree that the 'Makanjira trough' is dominated by exposed basement with any sediment accumulation necessarily related to subsidence along the Nkhotakota rift segment: (1) geological maps and boreholes indicate that sediments have not just accumulated along the northern half of the Makanjira trough, but along its entire length (Dawson and Kirkpatrick, 1968; Walshaw, 1965; Figure 1b of our discussion paper) and including to the south of the Nkhotakota fault as mapped by Scholz et al., (2020), (2) there is geomorphic evidence of recent multiple earthquakes along the Bilila-Mtakataka Fault (BMF; Hodge et al., 2018, 2019, 2020; Jackson and Blenkinsop, 1997), indicating that this is a highly active part of the rift capable of creating accommodation space for sediment accumulation, (3) boreholes indicate that these sediments in this section of the rift thicken to the west against the BMF (Dawson and Kirkpatrick, 1968; Walshaw, 1965), indicating that it is the BMF not the Nkhotakota fault that is primarily generating accommodation space, and (4), where there is exposed basement in the hanging-wall of the Chirobwe-Ncheu fault, this can be related to footwall uplift of the interior BMF (Lines 263-267 of the discussion manuscript).

With respect to the reviewer's second concerns, the higher elevation of the Makanjira trough relative to the Malombe trough does not require that these should be separated as distinct basins. There are other (albeit smaller) horst structures and basement highs that the Malawi Rift bifurcates around in the Central and North Basins of Lake Malawi (Ebinger et al., 1987; Scholz et al., 2020; Shillington et al., 2020). These also result in complex across-rift topography; however, they have not necessitated the division of the rift across strike. We suggest too that the formation of Lake Malombe does not require that the Malombe Fault has accommodated considerable throw as: (1) it only extends across the northern section of the fault and (2) it has a maximum depth of 5 m (Weyl et al., 2004). Indeed, this part of the rift has very little variation in subsidence with the Shire River experiencing only a 1.5 m drop in elevation in the 85 km distance between Mangochi and Liwonde; (Dulanya, 2017).

We agree with the reviewer that ideally subsurface data should be used to characterise the structure of the Malawi Rift. However, south of Lake Malawi, such data are scarce, and in their absence, we prefer to use the data we *do* have from the rift's topography and basement-penetrating boreholes to characterise its structure (Figure 2b; Bloomfield, 1965; Bloomfield and Garson, 1965; Walshaw, 1965). Cumulatively, we suggest that these observations indicate that the Malombe Fault should be considered as an intrabasinal fault (see Major Issue #4), albeit it could be one with considerable displacement (>500 m). In this context, it could be considered similar to some of the high displacement horst-forming intrabasinal faults (up to 2.5 km throw) in Lake Malawi (Scholz et al., 2020; Shillington et al., 2020).

To further demonstrate how we have used topography and borehole data to characterise the rift's structure, we propose that in the revised manuscript we would include across-rift cross sections for each basin in southern Malawi. For the Makanjira Graben cross section, as highlighted by the reviewer, we will note the Shire Horst structure and the lower elevation in the eastern side of the graben, as these structures were not described in sufficient detail in the discussion paper. Also note that Figure 4L-M in Lao-Davila et al. 2015 would not be a good reference for such a figure, as although they suggest a ~500 m thick sequence of sediments in the Malombe Trough, it is not clear what evidence they have for this assertion.

*Major Issue 4: Age of the Thyolo Fault and modelling of strain in Lower Shire graben (Fig. A3c and pg 37 lines 895-896, pg 38 lines914-915). The authors suggest that the Thyolo Fault is Karoo age. There is no evidence suggesting that there exists karoo-agesedimentary or volcaniclastic deposits on the hanging wall of the Thyolo Fault (Habgood, 1963; Habgood et al., 1973). Mesozoic activity along the Thyolo Fault would require subsidence of its hanging wall and creation of accommodation space for the deposition of volcanic and sedimentary sequences. Both Habgood (1963) and Castaing (1991) suggested that the Mwanza-Namalmbo Fault system is the eastern border fault of the Mesozoic Shire Rift. Castaing (1991) suggested that the Thyolo Fault is Cenozoic, bounding the currently active eastern domain of the Shire Rift. Therefore, I think this idea of Thyolo Fault being a Karoo fault needs to be revised except the authors provide data showing the presence of Mesozoic deposits on the hanging wall of the Thyolo Fault.*

In the context of a paper on active fault databases, the timing of Thyolo Fault activation is not important (the important part is that it is currently active). Nevertheless, in the context of our hanging-wall flexural modelling, we accept that we cannot prove the Thyolo Fault was active during the Karoo. However, equally, it cannot be proved that it was inactive during the Karoo, as its hanging-wall stratigraphy is poorly constrained.

We therefore suggest that in our revised hanging-wall flexure strain modelling, we will now also consider a scenario where the Thyolo Fault has only been active during East African Rifting and so has a hanging-wall sedimentary thickness equivalent to the maximum proven thickness of EAR sediments in its hanging-wall (64 m; Habgood, 1963). In this case, the flexural strain in its hanging-wall will be even less than previously modelled, and so if anything, will further support this analysis main finding; that hanging-wall flexural strain in southern Malawi is negligible.

*Major Issue 5: Definition of "border fault" in southern Malawi (Fig. 2a). The Lisungwe Fault, Malombe Fault, and Mwanza Faults are excluded from the 'border fault' definition and I am not particularly sure why. This is an issue for me, particularly because the structure of the basins point directly to the essence of these faults. For example, the Malombe Fault is the principal border fault of the Malombe Graben, not the Mwanjage Fault which you've assigned as the main border fault. Even the distribution of the amplitudes and wavelengths of the magnetic fabrics beneath the Malombe Graben in Fig.2c (Laõ-Dávila et al., 2015) clearly shows that the hanging wall of the Malombe Fault has significantly larger subsidence than that of the Mwanjage Fault. Also, the Mwanza Fault is a major border fault of the NW half of the Shire Rift (as shown in the maps in Figure 2). There is also evidence that the exposed segment of the Mwanza fault has been reactivated in the Cenozoic given by accumulation of Quaternary sediments on its hanging-wall (Habgood 1963).*

We thank the reviewer for bringing these points to our attention, and for demonstrating that the classification of border faults and intrabasinal faults is not always as clear-cut as the manuscript suggests in its current form. In the resubmitted manuscript, we will include a section where we more explicitly define the difference in border and intrabasinal faults, and how this was applied to southern Malawi (including the use of geological cross sections as outlined for Major Comment #3). We define border faults using the simplest geometric criteria: the fault at the

edge of the rift's surface expression. In other words, this definition is purely based on the geometry and distribution of brittle deformation across the rift. This definition is not inconsistent with differences between border and intrabasinal faults noted in previous studies (e.g. cumulative offset, slip rate, length; Agostini et al., 2011; Ebinger, 1989; Gawthorpe and Leeder, 2000; Muirhead et al., 2019; Wedmore et al., 2020b), but equally this definition is not dependent on these factors.

To specifically reply to the reviewer's comments on individual faults: the justification for not including the Malombe Fault as a border fault is discussed in Major Comment #3. With regards to the Mwanza Fault, we agree that it has been active during the East African Rifting, hence its inclusion in the SMAFD. Nevertheless, on the basis of the reviewer's comments and more recent mapping by Daly et al.,( 2020) that suggests this part of the rift may extend further into Mozambique and the Lower Zambezi Rift where it forms a different microplate boundary (Angoni-San), we will consider the Mwanza Fault as the border fault of a different rift section to the Lower Shire Graben. Unfortunately, in this case there are no geodetic constraints on the extension rate across the Zambezi Rift. In this case, we will calculate fault slip rates using values of between 0.2-1 mm/yr, where the lower estimate represents the minimum strain accrual measurable by geodesy (Calais et al., 2016) and the upper estimate represents that extension rates in the Zambezi Rift are unlikely to be higher than in the Lower Shire Graben given that the Mwanza Fault has only accumulated EAR sediments along its south-eastern most extent (Habgood, 1963).

The topography at the western edge of the Zomba Graben, where it grades into the Kirk Plateau, is very complex and so it is difficult to fit the Zomba Graben into conventional half-graben/graben models (Wedmore et al., 2020a). In particular, there are a number of N-S trending deeply incised valleys that lie to the west of the Lisungwe Fault and which have been previously interpreted as 'rift valley faults' (Bloomfield and Garson, 1965, see Figure 1 below). In addition, there are a number ENE-WSW trending valleys that are interpreted as 'cross faults' (i.e. strike-slip) faults (Bloomfield and Garson, 1965). Though only one of these faults (the Wamkurumadzi Fault) meet our definition of being active and is included in the SMAFD (section 3.1 of the discussion paper), inclusion of this fault, and the generally complex topography, requires that the Lisungwe Fault does not meet the definition of the border fault as outlined above. In the revised manuscript, we will more carefully outline how we have come to this decision and hope that this study will stimulate further studies into the question of fault activity at the western edge of the Zomba Graben.

[Figure]

Figure 1: Fault map for Zomba Graben underlain by TanDEM-X 12 m resolution digital elevation model. 'Inactive faults' as mapped by Bloomfield and Garson, (1965). TanDEM-X data was obtained via DLR proposal DEM_GEOL0686.

*Major Issue 6: Related to "Issue 4" above, the authors classified Namalambo Fault as an active Fault (e.g., Figs. 1b & 2). Namalambo Fault cannot be classified as an East African Rift System Fault because there is no evidence supporting its Cenozoic reactivation (see Bloomfield, 1958; Habgood, 1963; Castaing, 1991). The mentioned geological reports specifically stated that there is no Quaternary sediment deposition on top of the karoo sedimentary units at the base of its scarp, suggesting it has not been reactivated in the Cenozoic. Also, this fault does not satisfy the criteria stated by the author in Pg11 lines 260-267. Are the authors including it because they're assuming that it could be reactivated sometime in the future? If yes, I think this should be stated in the relevant figure captions and in the manuscript (particularly because this fault is a prominent fault in the area, and could be confusing to a reader without this additional information).*

Although we noted in the database that the Namlambo Fault formed during Karoo-age rifting, we accept the reviewers comments that there is very little evidence for its reactivation during East African Rifting, and so in the resubmitted manuscript we will remove this fault from the SMAFD and include it in the 'Malawi Rift Inactive Fault' database instead.

*Major Issue 7 (minor): A 'declaration' that I think is not clearly made in the set-up of the premise of the manuscript is that the parameterization approach focuses on tectonically active continental settings. I do not think that the authors imply that the approach is applicable to relatively more stable intraplate settings where much lower strain rates generally abound and potentially dangerous faults are buried, although some of those areas could be seismically active (e.g., intraplate induced seismicity). Therefore, I'll suggest that the authors state this clearly, at least in the abstract and introduction sections of the paper. I noticed that in different parts of the text, it is subtly implied with phrases relating to plate boundary, interplate setting etc., however, I think it will be beneficial to the reader if this is stated clearer from the onset.*

The reviewer is correct that the approach outlined for characterising seismic hazard is most applicable to low strain rate regions (plate boundary slip rates 0.1-10 mm/yr; Scholz et al 1986), which are distinct from both high strain rate regions (plate boundary slip rates >10

mm/yr) *and* stable cratons (plate boundary slip rates <0.1 mm/yr). We will correct for this in the revised manuscript.

*Question/Comment 1: Pg29 lines 703-707. The authors highlighted the anomalously large seismogenic thickness of the southern Malawi area, with continuous 30-60 km long fault sections. Crustal thickness map of southern Malawi (Njinju et al., 2019a) shows that an unusually thick crust dominates the area. In addition, heat flow map of the same area (Njinju et al., 2019b) shows an anomalous thermal gap in the area. Both of these have been associated it with an eastern extension of the Niassa Craton. Do you think that there is a possibility that these have an impact on the observed seismogenic thickness?*

The contribution of low heat flow to the anomalously thick seismogenic crust in southern Malawi is acknowledged (Lines 183-186), however, we thank the reviewer for the suggested reference, which we will include in the revised manuscript. It is also worth noting that Fagereng, (2013) demonstrated that an anomalously low heat flow alone (63 mW/m$^{-2}$) is not sufficient to explain the thick seismogenic crust in this region, with other factors such as strain rate and crustal composition also important. Furthermore, the seismogenic crust is unusually thick throughout Malawi (>30 km), even in places where the heat flow is higher (e.g. 65-70 mW/m$^{-2}$ in Karonga; Ebinger et al 2019).

*Question/Comment 2: It is well-known that the patterns of seismogenic fault reactivation are influenced by the frictional stability of faults. Asides from strain rate and lithology/mineralogical composition, another important factor that influence the frictional stability is geothermal gradient/heat flow. Well-constrained heat flow & geothermal gradient maps of southern Malawi (Njinju et al., 2019b) show interesting thermal anomalies within the areas analyzed in this study. I am curious as to how the heat flow distribution in the area may affect the results/conclusions of this study.*

Although the reviewer raises an interesting point, as noted above there are other factors beyond heat flow that will control fault reactivation in this region. We consider that to discuss them all will go beyond the scope of this study, which is focussed on active faulting, not crustal rheology. Furthermore, classic Mohr-Coulomb theory suggests that brittle fault reactivation is temperature independent (Sibson 1985), although it will partially influence thickness of the seismogenic crust (as discussed above).

**Comments on Reviewer Supplement**

*Line 31: This sentence, as it is written here, implies circular logic. This is because measurements of fault length is one of the inputs into your model estimates of EQ magnitude. How about "These potentially high magnitudes for continental normal faults are compartible with the observed 11-140 km-long faults and thick (30-35 km) seismogenic crust of southern Malawi."*

We will revise this sentence as the reviewer suggests

*Line 61: The repetition of "estimate" sound awkward...could reword as "fault slip rates can be estimated using geodetic constraints"?*

We will correct this in the revised manuscript as advised

*Line 87: I suggest a rewording of this text. Southern Malawi lies NEAR the southern incipient end of the EARS, not "AT" the end.*
*Rather, it is Central/Southern Mozambique that lies AT the southern incipient end of the EARS and has all those characteristics you've mentioned...see papers on the MOZART project (e.g., Fonseca et*

*al. (2014), Urema Graben, Mazenga Graben, Chissenga-Urema Graben System, and the Changani Graben System (Mueller & Jokat, 2019).*

We will correct this in the revised manuscript and outline that the Malawi Rift lies *near* the southern end of the EARS

*Line 112: I think the authors should include the Salambidwe Igneous structure (Cooper, 1961) and the flood basalts associated with the Lupata Volcanic Complex (footwall and hanging wall of the Panga Fault; Habgood, 1963).*

We thank the reviewer for noting these omissions and will include them in the revised manuscript

*Line 123: By definition, can a graben can be bounded by 1 border fault? I suggest need rewording*

We will replace 'graben' with 'basin' in the revised manuscript

*Lines 136-144: This paragraph is written with a mixed context that could create confusion...i.e. written with a context of a political territory (i.e. southern Malawi) and a rift basin (Malawi Rift). I will say this is very 'dangerous' as it propogates a very common problem with the way geology has been carried out in Africa for a very long time. Therefore, I will suggest that the authors stick to "Malawi Rift" since this study and this particular paragraph focuses more on tectonic history.*

*On the aspect of the evolution of the Malawi Rift, I will refer the author to Scholz et al. (2020) which demonstrates clearly the southward episodic propagation of the Malawi Rift. In addition, studies have showed that Karoo sandstones outcrop along the Karonga border fault of the Karonga Basin, and Accardo et al. 2018 observed an anomalous velocity interval that suggests highly lithified Karoo sedimentary rocks directly overlying the basement beneath the Karonga and Usisya Basin fill.*

As outlined for Major Issue #1 we will more carefully define the term 'Malawi Rift' in our revised manuscript, and note that the databases cover the political region of southern Malawi, not the Malawi Rift. We will incorporate the Scholz et al (2020) reference into the revised manuscript.

*Lines 146-150: The inferences made in this section, as written, sounds highly speculative, and since it is placed in the "geologic setting" section, I will suggest rewording. First, the text totally ignores the Malombe Graben as the primary hydrologic linkage between the Lake Malawi and the Zomba Graben (i.e. hosts the axial stream). Second, the text implies that sedimentation in the grabens referred to are only associated with flooding episodes. This is very strange to me. The area described is defined by the southward bifurcation of the Malawi Rift into two narrower branches: a graben in the east (Malombe Graben), another graben to the west (Makanjira Graben). The Malombe Graben has a well-developed lake from which the axial stream Shire River flows southwards. South of the bifurcation, the two branches merge back into a weakly-extended graben (Zomba Graben) south of which the basement is exposed and faulting is diffused. The point here is than the bifurcation troughs are actively subsiding, fault-bounded tectonic elements with structurally-controlled axial (Shire River) and transverse streams (from the rift flanks) channelling sediments into the subsiding basins. Thus, it is more likely that both faulting and climate control sedimentation within these basins, and not 'only climate' (as it is described here). For general reference on interactions between faulting and climate in humid rift settings, I refer the authors to Gawthorpe & Leeder (2000).*

In the revised manuscript we will note that faulting in the rift will have influenced sedimentation. Nevertheless, it should be noted that base level changes in Lake Malawi are thought to be primarily driven by climatic forcing (e.g. Scholz et al 2007; Lyons et al 2015).

*Lines 151-152: What does this sentence mean? ...you mean the steep gradient of the rift floor does not correspond to a fault escarpment? As this sentence is written, it precludes every form of structural control on the gradient...and I am highlighting this because this particular zone is a transfer zone between the Malawi Rift and the currently active part of the Shire Rift. Transfer zones in areas of incipient rifting are typically characterized by elevated/exposed basement...for reference, see Heilman et al. (2019) and Gawthorpe & Leeder (2000).*

This sentence refers to the point that no active faults were identified in the region between the Zomba and Lower Shire Graben from fieldwork and analysis of high resolution digital elevation models (although we of course cannot exclusively prove that there are no active faults in this region). We will clarify this in the resubmitted manuscript.

*Lines 157-158: First, I think it should also be added that the Castaing (1991) mapping was infact, only limited to the Shire Rift part of southern Malawi...the mapped faults in the paper did not extend into Southern Malawi Rift.*

*Second, I have seen a lot of these faults mapped previously in the Malawi Geological Survey reports (e.g., Bloomfield, 1958; Habgood, 1963; Habgood et al., 1973)... some of which were cited in this manuscript. I could see that the authors mentioned some of these faults on pg 12, however, I think the contributions should also be acknowledged here.*

We will make it clearer in the revised manuscript that Castiang (1991) only considered EARS faults in the Lower Shire. With respect to the second point, we are discussing the mapping of *active* faults. The previous Malawi Geological Survey reports, although an excellent resource, made few attempts to differentiate active and inactive faults (as described at Lines 284-291).

*Line 161: The mapping in the Geological Survey reports looks pretty fine-scale to me because they were done on the field. I think the detailed info on slip rates and recent faulting are the parts that are missing which this current study provide.*

We are referring here specifically to the Global Earthquake Model active fault database here (Christophersen et al., 2015; Styron and Pagani, 2020), which do not incorporate the faults mapped by the Malawi Geological Survey, but those mapped by Macgregor, (2015). We will clarify this in the revised manuscript

*Line 178-179: This sentence sounds weird with the "to 1965". Pls check*

We will revise this sentence in the resubmitted manuscript as requested.

*Lines 181-188: Wondering if it will also be worth mentioning the crustal thickness in southern Malawi (Njinju et al., 2019a; Tectonics), and heat flow distribution in southern Malawi (Njinju et al., 2019b; J. Volc. & Geoth. Res.)*

*Njinju, E.A., Atekwana, E.A., Stamps, D.S., Abdelsalam, M.G., Atekwana, E.A., Mickus, K.L., Fishwick, S., Kolawole, F., Rajaonarison, T.A. and Nyalugwe, V.N. (2019a). Lithospheric structure of the Malawi Rift: Implications for magma-poor rifting processes. Tectonics, 38(11), pp.3835-3853.*

*Njinju, E.A., Kolawole, F., Atekwana, E.A., Stamps, D.S., Atekwana, E.A., Abdelsalam, M.G. and Mickus, K.L. (2019b). Terrestrial heat flow in the Malawi Rifted Zone, East Africa: Implications for tectono-thermal inheritance in continental rift basins. Journal of Volcanology and Geothermal Research, 387, p.106656.*

As outlined for Question/Comment 1 we will incorporate these references into the revised manuscript.

*Lines 202-203: True. However, another possibility that could be mentioned here is local stress rotations (see Morley, 2010...case study on the Rukwa Rift).*

As discussed in Williams et al., (2019) stress rotations do not explain the discrepancy in extension direction when inferred from geodesy or earthquake focal mechanisms in southern Malawi, as the orientation of recent joint sets in the region is uniform across the rift suggesting that the regional stress field is uniform. Furthermore, the hypothesis of Morley, (2010) is that that stress rotations reflect changes in foliation orientation, in which case it would not account faults locally cross cutting the foliation in southern Malawi.

*Line 237: also depth of medium-large magnitude EQ ruptures?*

We will include focal depth as a factor of whether an earthquake ruptures to the surface in the revised manuscript

*Line 250: "little" sounds awkward here, because how little is "little"? I'll suggest "limited" instead*

We will clarify this in the revised manuscript, and note that there is some limited dating from <10 ka sediments around Lake Malombe (Van Bocxlaer et al., 2012) and <50 Ka sediments 20-30 km east of the rift around Lake Chilwa (Thomas et al., 2009).

*Lines 458-459: See comments in 'major issues'.*

See reply to comment on Major Issue 3

*Lines 464-464: Isn't this graben is known as the Urema Graben/Rift (Castaing, 1991; Fonseca etal. 2014; Lloyed et al., 2019). Why give it a different name?*

*Castaing, C. (1991), Post-Pan-African tectonic evolution of South Malawi in relation to the Karroo and recent East African rift systems. Tectonophysics, 191(1-2), pp.55-73.*

*Fonseca, J.F.B.D., Chamussa, J., Domingues, A., Helffrich, G., Antunes, E., van Aswegen, G., Pinto, L.V., Custódio, S. and Manhiça, V.J., 2014. MOZART: A seismological investigation of the East African Rift in central Mozambique. Seismological Research Letters, 85(1), pp.108-116.*

*Lloyd, R., Biggs, J. and Copley, A., 2019. The decade-long Machaze–Zinave aftershock sequence in the slowly straining Mozambique Rift. Geophysical Journal International, 217(1), pp.504-531.*

As for the Malawi Rift, there is little consensus on the extent of the Urema Graben (see also, Steinbruch, (2010) which suggest it refers to mainly the basins around the river Urema 150 km along strike to the south). Our preference here is to avoid using this term as it may imply that our fault mapping covers the full extent of the Urema Graben, and this will be clarified in the revised manuscript.

*Lines 473-474: Does this estimate include subsurface measurement of the total throw at the hanging wall cut-off of the faults? Please, provide reference. Also, based on the wording, does this refer to southern Malawi Rift (excluding the 'Lower Shire Graben') or does it refer to all the faults in southern Malawi geopolitical boundary?*

Yes, this estimate includes the limited subsurface data that does exist in southern Malawi (i.e. groundwater boreholes; Bloomfield, 1965; Bloomfield and Garson, 1965; King and Dawson, 1976; Walshaw, 1965), and we will clarify this in the revised manuscript.

*Lines 687-689: For the sake of the reader, I think the hazard part should be stated before implications for continental rift as seismic hazard is the primary focus of this study. Thus, I'll suggest a rewording to: "In the following section, we examine some key results of the SMAFD in terms of its implications for seismic hazard in southern Malawi, its contribution to our understanding of fault growth in continental rifts, and future strategies to...".*

As discussed with respect to the comments for Lines 690-711 from Reviewer #1, we will be removing this section on 'controls on fault growth in southern Malawi' in the revised manuscript.

*Line691: This relates to my comment above... While this section is important and relevant, for the reader, it seems like a sudden digression from the seismic hazard story to bring it in at this early part of the discussion. Pls consider swaping it with '6.2 Implications for seismic hazard in southern Malawi'.*

See our reply to the previous comment.

*Line 694: Could also include Scholz et al. (2020)*

*Scholz, C.A., Shillington, D.J., Wright, L.J., Accardo, N., Gaherty, J.B. and Chindandali, P., 2020. Intrarift fault fabric, segmentation, and basin evolution of the Lake Malawi (Nyasa) Rift, East Africa. Geosphere.*

We thank the reviewer for bringing this article to our attention, and will incorporate it into the revised manuscript.

*Line 843: growth,*

We will correct this grammatical error in the revised manuscript

*Line 895-896: This statement, as written, is misleading. Please revise. Castaing (1991) argued that the Thyolo Fault is Cenozoic and along with the reactivated Mwanza Fault, is accommodating strike-slip in present day. He suggested that the Mwanza Fault was the primary eastern Karoo and Cretaceous border fault of the Shire Rift. See pages 65-66 and figs. 7,9,&10 of Castaing et al. (1999).*

In the revised manuscript, we will correct the Castaing, (1991) reference to state that he did not consider the Thyolo Fault to have been active during the Karoo.

*Lines 897-899: This statement, as written, is speculative. As I explained in my "Major issues 1 & 2", the Shire Rift is a multiphase rift of which the Lower Shire graben is the easternmost sub-basin. The Malawi Rift and Shire Rift have different tectonic histories and structure, and only linked up at the current location of southern Zomba Graben. Although the development of the Lower Shire graben could be syn-tectonic with the southward propagation of the Malawi Rift, the pre-rift and early-phase structures in the Shire Rift (which are absent in the Zomba and Makanjira grabens) could greatly impact the strain in the Lower-Shire Graben...e.g., consider the influence of the mechanical load of the thick sequences of volcanic flows in the hanging wall of the Panga Fault and buried Mwanza Fault (all beneath a part of the Lower-Shire graben), which could impact the throw on the Thyolo Fault. Thus, in the absence of actual subsurface data on Thyolo fault throw, I think it is speculative to make a statement like this. I understand that there is a need to assume a maximum throw limit for the*

*modelling, thus, I'll suggest that the authors revise this sentence by stating that the estimate is an assumption.*

We will correct the revised manuscript to state that these are assumptions that we have made about the throw of the Thyolo Fault, which though speculative do not invalidate the results of our modelling. See also our reply to Major Issue #4.

*Lines 900-901: See my comments in "Major Issue 3" related to the structure of this rift segment as implemented in the model.*

See our reply to Major Issue #3

*Line 909: The lower Shire graben is no more than 38-40 km wide (at its widest). I'm curious to know how the extent of this basin is estimated?*

This was estimated from the combined width of the Karoo and EARS sections of the Lower Shire Graben (i.e. it incorporates the Karoo deposits in the footwall of the Panga Fault). We will correct this in the revised flexural modelling, so that for the modelling where the Thyolo Fault is assumed to be an EARS-only fault (see Major Issue #4), we only considers the extensional strain over a 40 km wide rift, which is the width of EARS sediments in the Lower Shire Graben.

*Line 914-915: This statement assumes that the Thyolo Fault is a Karoo-age border fault, which I think is problematic considering the observations in Castaing (1991). See "Major Issue 4".*

See our reply to Major Issue #4

*Line 1620/Figure 1b: For figure 1b: It will be very helpful if you can add symbols or number the colored polygons of the different terranes shown. The colored polygons are faded into the grey scale hillshade map which makes the color slightly different from the ones shown in the legend....by adding symbols or numbering, it makes it easier for the reader to identify where what terrane is.*

We will add text labels for the terranes in the revised manuscript

*Line 1621/Figure 1a: I'll suggest that you include the Aswa Shear Zone to this map as it is one of the most well-known lithospheric-scale shear zones in East Africa (see Daly et al., 1989; Ruotoistenmäki et al., 2014; Katumwehe et al., 2016; Saalmann et al., 2016). Other ones you could also include are the Lurio Shear Zone and the Sanangoe Shear Zone.*

We included the Aswa Shear Zone, as mapped by Fritz et al., (2013) in Figure 1a, but will revise the extent in the resubmitted manuscript as per the updated references the reviewer suggests. We will also include the other shear zones noted by the reviewer.

*Line 1639/Figure 2c: You might want to check the N-striking (80deg-dip) foliation anotated on the Namalambo Fault in Fig2c. The foliation trends in the Nsanje horst are NE-trending. The Namalambo Fault cuts the foliation. For reference, see "Structual Map of the Northern PortHerald Hills" in Bloomfield (1958).*

As suggested by the reviewer, we will remove this strike and dip measurements to reflect the regional NE-striking fabrics in the Nsanje Horst (albeit with local variations).

*Line 1732/Figure A3: The orientation of the Zomba and Lower Shire profiles should also be stated in this caption.*

We will include the orientation of these profiles in the revised manuscript.

*Line 1736/Figure A3: You mean WSW-ENE?*

Yes, this will be corrected in the revised manuscript

**References**

[revised manuscript text omitted]

---

## Author Response (AR1)

$\begin{array}{c}
\\
\\
\\
\\
\\
\\
\\
\\
\\
\\
\\
\\
\\
\\
\\
\\
\\
\\
\\
\\
\\
\\
\\
\\
\end{array}$

Dear Editor

Kind regards

Der Mrs without tracked changes unless.

Jack Williams (on behalf of all co-authors)

We thank you for soliciting two thorough, informed, and constructive reviews on our Solid Earth

Discussion article by Richard Styron and Folarin Kolawole, which will undoubtedly improve our

Solid Earth. Unless otherwise stated, line numbers refer to the clean version of the manuscript

We thank you for consideration of this manuscript and look forward to hearing from you.

manuscript. Below, we have copied the reviewer's comments in italics and replied to them in blue, cambria text with how we would like to incorporate them into a revised manuscript to submit to

**26 Reviewer 127**

The paper by Williams et al. provides a high-quality map of active faults in southern Malawi, and presents a clever method for partitioning regional deformation rates onto the rift structures, with a thorough exploration of the uncertainties. The work (the mapping, rate estimation, and manuscript) are executed competently, and there is nothing that is strictly incorrect, although some topics would benefit from a bit more explanation if not revision. The mapping is quite high quality, and is the most solid contribution made here.

Although the authors may not want to do this, I think that the work could benefit from being split into two different, shorter papers: one that presents the fault mapping and discusses it in a bit more detail and context (although not too much more), and another that presents the parameter estimates, ideally as a part of a PSHA. (Note that I am not asking that this be done for major revisions–it's just something to consider doing.)

We thank the reviewer for their suggestion, which we have considered very seriously. Our preference is 42 to keep the mapping and parameter estimates together in one paper, as we feel they are inherently 43 linked and that a strength of the current study is the fact that it spans traditional discipline boundaries, 44 Nevertheless, we see the importance of more distinctly separating between our observations and 45 interpretations. As outlined below with respect to major issue #1, and in the introduction in the main 46 text (Lines 68-94) we have recognised this by describing and providing two distinct GIS file databases 47 in this study, the 'South Malawi Active Fault Database' (Sect. 3) and 'South Malawi Seismogenic Sources 48 Database' (Sect. 4). In this way, the reader will be more able to separate between the fault mapping and 49 the parameter estimates, and any user has a choice between adapting the fault mapping only or also 50 including our parameter estimates. 51

Major issues:53

**1. Separation of data and estimates**

The first issue is that there is no apparent separation in the fault data and attributes between 57 observation (and associated interpretation firmly based in observations) and rough estimation based 58 on little to no data. Although the authors are helpfully conforming to the schema laid out by the 59 GEM Faulted Earth (GFE) project (e.g. Christophersen et al. 2015), I believe that the GFE is 60 intended to hold observations or measurements, rather than the estimates made in this project. For 61 example, recurrence intervals would be derived from paleoseismology rather than calculated from the slip rate and assumed magnitudes. Only considering measured recurrence intervals makes the 62 63 recurrence intervals independent of assumptions of earthquake magnitude, scaling relations, or 64 other factors. Christophersen et al (2015) state that the database is often sparse where observations 65 don't exist. 66

The coupling of the fault traces (which are observations or data, as far as I am concerned, even if there is an interpretive component) with the parameter estimations crosses a traditional boundary. 68 69 Typically, the fault data are considered to be somewhat objective and immutable (though subject to 70 revision) while the derivation of earthquake rates is considered to be part of the modeling process 71 and often done while making fault sources for a PSHA. Model results are a bit more subjective and 72 mutable, as they are dependent on assumptions that are explored during the modeling process 73 and/or are project-dependent. A good modeler will have a process for testing and refining some of 74 these assumptions (i.e. magnitude scaling relations or slip partitioning) through comparisons with 75 instrumental seismicity or other observations. However the observed data are usually not revised to improve a data-model fit. A user who wants to incorporate this dataset into a seismic hazard model, 77 but who may not agree with some of the model assumptions used here (i.e. scaling relations, 78 magnitude ranges, or the style of partitioning between internal and border faults) may have a hard 79 time knowing what to keep and what to discard, without reading a long paper. This can be a big 80 challenge for the many seismic hazard modelers who do not have a great facility with the English 81 language. Similarly, if this data were incorporated into other fault databases, the end user may not 82 be able to cleanly separate data from model results. 83 84 This does not mean that what the authors have done is wrong or necessarily needs to be changed. It 85 is just to raise their awareness of a potential concern (that these estimates may be confused for 86 observations) and that many hazard modelers would prefer to redo the estimation rather than rely on these results. I am not sure of the best course of action. If it were me doing the work, I would 87 88 separate these processes and release both 'only data' and 'data plus estimates' datasets. I would 89 also consider publishing them independently, and perhaps incorporating the rate estimation work 90 into a PSHA rather than going part way as is done here. But there is no 'right' or scientifically 91 optimal decision here, and a lot depends on the particular circumstances of the authors. 92 93 Another possibility is to keep the parameter estimation through the slip rate estimation but stop 94 there, which would avoid the problems of choosing a magnitude-frequency distribution, estimating 95 the seismogenic thickness of the crust, etc. In a typical project, these tasks are often done by the 96 hazard modeler rather than the geologist who prepares the fault data up through slip rate 97 estimations. 98 I can state that as a fault data compiler, I am a bit hesitant to bring any of the estimated parameters 99 into the GEM Global Active Faults Database, as they are too poorly constrained and data-limited, 100 and I don't want users to confuse them for measurements. 101 102 As noted above, to follow the advice to release both 'only data' and 'data plus estimates' datasets, in this 103 revised manuscript we have described and included two separate GIS databases of faults in South 104 Malawi: 105 1. The South Malawi Active Fault Database (SMAFD, Sect. 3, Table 1): this database incorporates 106 107 the objective mapping and geomorphic 'trace' attributes included in the previous version of the 108 SMAFD. In addition, individual GIS features represent faults (see minor comment for Reviewer #1 Line 325). In this way, the SMAFD conforms to the 'only data' dataset requested by the 109 110 reviewer and will also be readily comparable to the newly published GEM Global Active Fault 111 Database (Styron and Pagani, 2020). 112 The South Malawi Seismogenic Source Database (SMSSD, Sect. 4, Table 2): this database 2. 113 incorporates the modelling derived attributes included in the original version of the SMAFD 114 that are required to turn the mapped faults into earthquake sources for PSHA (e.g. fault segmentation, fault width, slip rates, earthquake magnitudes, and recurrence intervals). Here, 115 116 individual GIS features represent fault segments, which can each be considered distinct sources 117 for PSHA. Furthermore, in line with other earthquake source databases (Basili et al., 2008; Field et al., 2014; Stirling et al., 2012) faults in the SMSSD are mapped as straight lines that connect 118 119 segment tips, as opposed to truly honouring the surface trace of the faults (see for example the 120 comparison for the Chingale Step Fault in Fig. 4). In this way, the SMSSD will conform to the 121 'data plus estimate' dataset requested by the reviewer. 122 123 By distinctly describing these two datasets in the revised manuscript, whilst also clearly outlining how they are linked, we thus address the reviewer's concern about how users of these databases may be confused by which fault attributes are objective measurements, and which are model-driven.

Furthermore, we maintain the multidisciplinary aspect of this manuscript, and allow seismic hazard modellers to understand the limitations for geologists investigating active faults in a region where data
is sparse and vice versa (Lines 96-98). Work to include the SMAFD and SMSSD into PSHA in southern
Malawi is ongoing, and this will be the topic of a subsequent study.

**2. Estimation of uncertainty:**

The second issue is that the logic tree framework used to propagate uncertainties and explore the 134 parameter space is perhaps more complex than it needs to be based on the lack of input data. It is a clever method and there seems to be nothing incorrect in the implementation, but I question the 135 136 wisdom of using it southern Malawi where there is essentially no data to feed in. The old saying in modeling is "garbage in, garbage out"; in this case it's more like "nothing in, nothing out" (I am 137 138 not suggesting the work is garbage!). The exercise seems to simply quantify the obvious, that each fault slips somewhere between 0-5 mm/yr. I am not sure that it provides much value. The further 139 140 work, estimating recurrence intervals, has larger theoretical issues (discussed below) in addition to 141 adding several more layers of uncertainty into the results. It could easily be removed from the 142 database (though perhaps kept in the paper for discussion). 143

As a subordinate issue, I don't think a logic tree framework is really the most appropriate method of
propagating uncertainty as used; it is more appropriate when the parameters that make up the
branches in the tree are discrete variables with a few choices, rather than continuous random
variables. For example, an appropriate use of logic trees is to consider different scaling
relationships.

With continuous random variables (i.e., extension rate or dip), the use of unweighted logic trees
considers the lower, mid, and high values to all have equal probability. Do the authors consider this
to be the case? Do the authors believe that the resulting low, mid and high values are equally
probable? Even if the inputs are all equal, if there are no correlations between the different
parameters, the middle values should be more probable (see for example the Central Limit
Theorem).

In my opinion, a more appropriate method for representing the uncertainty in the results (i.e., slip
rates or recurrence intervals) would be to define distributions for each continuous random variable
(i.e., dip or total geodetic extension at that latitude) and then randomly sample from these
distributions, and then characterize the resulting distributions for the results parameters. This is a simple Monte Carlo method. The major strength of this method is that the sampling will cover far
more of the parameter space than a coarse 'low/med/high' sampling method. It is also quite trivial
to introduce distributions for each parameter that may reflect prior knowledge (i.e., a PDF of

- 164 regional
- 165 *dips based on focal mechansisms or structural measurements).*

One way to think of this is that the representation of uncertainty in the model should reflect the real
uncertainty of the parameter. Continuous variables should be represented through continuous
distributions, while discrete variables (i.e. the choice of scaling relationships) should be represented
through discrete distributions (i.e. lists or arrays, perhaps weighted).

The strategy employed here does a good job of defining the absolute range of the results based on 173 the inputs, but a worse job of defining the central values (broadly like the mean and one standard 174 deviation rather than three standard deviations). If the authors believe this is the better choice, that 175 is fine, but I would like to hear their arguments.

We can appreciate why the reviewer might question the usefulness of the large uncertainties in our slip 178 rate and recurrence intervals estimates. However, the fact remains that even being able to quantify slip 179 rate estimates of 0-5 mm/yr in southern Malawi represents a significant advance, given that prior to 180 this study, no slip rate estimates have been previously made (Lines 78-79). Furthermore, as highlighted 181 in the manuscript (Lines 672-675) the large range of values we obtain in our recurrence interval 182 estimates are not unusual compared to other low strain rate regions with limited paleoseismic 183 information (Villamor et al., 2018) and can still be incorporated into PSHA (Hodge et al., 2015). Indeed, 184 these large ranges can be considered as an important outcome of the study, as they can be used to the 185 highlight the parameters that need further study to reduce uncertainty (Sect. 5.4 of the manuscript). 186 187 We note that logic trees have been used elsewhere to propagate uncertainty in seismic hazard in 188 regions with little paleoseismic data (Villamor et al., 2018). However, we do agree with the reviewer's 189 comments on how we should consider our lower, intermediate, and upper estimates; indeed, and note 190 in the revised manuscript that the upper and lower values obtained from the logic tree required an 191 unlikely set of parameter combinations (Lines 675-678), and that treating these values as a continuous 192 variable and assigning a probability distribution function to them would be a more appropriate method

A more complex treatment than the intermediate, lower, and upper slip rate and recurrence interval
estimates obtained from the logic tree are beyond the scope of this study, but already under
consideration for our next paper. In this revised manuscript, we have therefore included a distinct
section on how the fault data included in the SMAFD and SMSSD could be incorporated into PSHA (Sect.
6.3), where we more explicitly discuss some of the reviewer's excellent suggestions. This would also
address Major Issue #3 (see below).

**3. The calculations of recurrence rate:**

of treating them in PSHA (lines 1599-1600).

The authors choose to calculate recurrence rates under the assumption that all of the seismic moment that accumulates on each fault is released during earthquakes of identical magnitude. This 205 206 is essentially the "characteristic earthquake hypothesis" which featured quite prominently in midlate 20th century paleoseismology and PSHA but was always quite contentious (for example see 207 "Characteristic Earthquake Model, 1884-2011, RIP" by Kagan et al. 2012, Seismological Research 208 209 Letters). This hypothesis is believed by fewer and fewer scientists with each passing year, as our 210 observations of variable rupture segmentation and per-event displacement accrue. The few remaining national-level PSHA models that still use a 'pure' characteristic earthquake model (not a 211 212 distribution that includes aleatoric variability) do so primarily because it simplifies time-dependent 213 hazard analysis. The modern state of practice is to consider a range of earthquake sizes on each 214 fault, and to distribute moment throughout the range of earthquake sizes by specifying the relative 215 frequencies of different magnitudes of earthquakes, and then calculating the absolute frequencies 216 through moment rate balancing. 217 The canonical reference for this is Youngs and Coppersmith (1985 BSSA), which provides equations 218 219 for multiple magnitude-frequency distributions. GEM's Open-Quake Engine and OQ Model

Building Toolkit also has some Python code for this purpose, if the authors are interested in using or 221 studying a functional implementation (https://github.com/gem/oq- engine/tree/master/openquake/hazardlib/mfd; https://github.com/GEMScienceTools/oqmbtk/blob/master/openquake/mbt/tools/fault\_modeler/fault\_

modeling\_utils.py#L2379). If the authors favor the pure characteristic earthquake hypothesis, then they should provide some supporting arguments. Otherwise they may either calibrate the magnitude

- 226 frequency distributions, or simply drop this part of the estimation procedure (even if
- 227 they retain the estimates up through the slip rate calculations).
- 228

We recognise that we make a large simplification by considering characteristic whole-fault or segment 230 ruptures only, and that fully incorporating the SMAFD into PSHA requires further evaluation of 231 individual faults' magnitude-frequency distribution. We have now ensured that this simplification is 232 spelled out in the revised manuscript (Lines 682-685), but also noted that by considering both whole 233 234 fault or segmented fault ruptures, our study already partially recognises that the characteristic earthquake hypothesis does not necessarily apply in southern Malawi, because the faults will not 235 always host similar sized events Furthermore, some recent studies have noted that the characteristic 236 earthquake hypothesis, and its use in PSHA, is not necessarily 'dead' (Stirling and Gerstenberger, 2018). 237 238 239 It is also worth reflecting that in studies where faults are allowed to host a range of earthquake magnitudes in PSHA, these are built from many years of detailed geological mapping, historical, 240 instrumental and paleo-, seismicity data, and hazard modelling (Basili et al., 2008; Field et al., 2014). 241 Conversely, prior to this study, there has been very little systematic investigation of possible 242 earthquake magnitudes and recurrence intervals at all in southern Malawi. Hodge et al., (2015) is an 243 exception, and this was based on a very limited active fault mapping. So even though the recurrence 244 interval and earthquake magnitude estimates in this study are poorly constrained compared to other 245 countries' seismic hazard assessments, we still consider them to represent a step change in our 246 understanding of southern Malawi's seismic hazard. 247

Therefore, similar to Major Issue #2 above, our preference has been to incorporate the reviewer's 249 comments in a distinct section where we discuss how this study could be used into PSHA (Sect. 6.3).

Minor issues:

**252 Data license, distribution and updates: 253**

One of the promises of 21st century science is that new technologies enable rapid and low-friction 255 sharing, integrating, and updating of data. However, it raises some new topics that have been 256 heretofore ignored by most. The first is the license of the data. As the creators of a nice dataset, the 257 authors are entitled to specify the terms and conditions under which others may use it. A good 258 "open-data" choice is the Creative Commons Attribution license, which is what the articles 259 published by the EGU/Copernicusjournals use. 260

However, the authors may wish to specify a different license, such as a non-commercial license 262 (meaning that it can't be sold or used for other commercial purposes), a share-alike license 263 (meaning that any modifications to the data, which are allowed by the Creative Commons licenses, 264 must be redistributed under the same conditions), or various others. There are also more and less 265 restrictive licenses, but these may start to conflict a bit with the release of the data in this journal. 266

It may sound like a bit of boring lawyer stuff, but it's very important to many of us that deal with 268 others' data regularly. If the authors want the data to be most useful, please explicitly state what the 269 license is, so the potential users can have some clarity about what they can or can't do with it. It's 270 an easy process: just put a 'license.txt' file in the zip with the GIS data. 271

Similarly, the data will probably see a lot more use if it is easy to get to, and in a place where it's 273 easy for the authors to update. The easiest here is using GitHub (github.com) which has turned into the default small data distribution channel for many, including the GEM Global Active Faults

Database. GitHub, or other similar services such as GitLab, provide a great platform for licensing, distributing and updating data, in a way that makes the history of the data transparent to the users

by being integrated with a version control system. 278 279 Something else to consider is whether the authors would welcome updates or extensions to the 280 mapping (and perhaps parameter estimation). It may be that other users who are interested or have 281 some need for a fault database over a wider area than just that covered in this dataset, and may 282 want to expand along strike. This is the kind of collaborative science that is quite easy to do now, 283 especially with services such as GitHub, but I don't think the academic publication process, and 284 allotment of credit (citationsetc.) has caught up. Nevertheless, if the authors support this (in 285 principle, no need to blindly accept changes) they could write a sentence or two in the manuscript 286 or in a text file with the data describing this. 287

We recognise the importance of findable, accessible, interoperable, and reusable data (i.e. the FAIR
principles), and thank the reviewer for their advice. Indeed, it is these principles that partly guided our
decision to submit this study to *Solid Earth*. As the reviewer recommends, when resubmitting the
SMAFD and SMSSD, we have been explicit that this is licenced under Creative Commons Attribution
ShareAlike (CC-BY-SA 4.0) Licence (in keeping with the GEM Global Fault Active Database).

**293 Publication of code to perform parameter estimation:294**

I think that by default, any code used in a scientific work should be published with the paper. This 296 would definitely include any code used to perform the parameter estimation (one assumes it wasn't 297 done on a hand calculator). There may be some extenuating circumstances where publication of 298 code isn't a good idea, but this would involve prior intellectual property restrictions or something. I wouldn't consider messy scripts to be exempted here. Detailed inspection of methods and 299 300 reproducibility is central to the scientific process, and code is perhaps the most perfect form of 301 scientific inquiry thatallows for this. Please publish the code, even if it's a messy script of zip file of 302 them. (Ialso think that EGU/Copernicus asks for this but I could be wrong.)

As outlined above with respect to sharing our GIS file, we appreciate the importance of data that
 follows the FAIR principle. In this case we will include the excel file where our earthquake source
 estimates are calculated with our resubmitted file.

**308 Line edits:**

Line 5 (and throughout manuscript): Superscripts are formatted as subscripts. This is particularly
 annoying with exponents.

This was a formatting error when converting the word document to a pdf file, and will of course be
 corrected in the typeset version of the manuscript

Line 18: All seismically active areas on earth have instrumental records much shorter than the
'repeat times' of the larger earthquake produced in these regions (hundreds to tens of thousands of
years).

We recognise that the phrasing of this sentence was not precise, and this has been corrected in the abstract (Line 17-18). However, we also note that tough the reviewer is correct in what they say, we would argue that this problem is particularly acute in low strain rate regions where earthquake recurrence intervals may be ~10,000-100,000 years (i.e. the ~100 year long instrumental record covers 0.1-1% of a fault's seismic cycle). as opposed to 100-1000's of years in high strain rate regions (where the instrumental record covers 10-100% of a fault's seismic cycle). We discuss this further in the revised manuscript at lines 82-84.

Line 56: Actually, active fault databases have been developed for close to all seismically active 329 regions; the GEM Global Active Faults database is referenced elsewhere in the paper, which has 330 global coverage. Some areas (like the EARS) need better mapping and slip rate measurements, but 331 active fault data does exist 332 333 We have correct for this in our revised submission (Lines 53-55), but as the reviewer acknowledges, 334 emphasis that though there is global coverage of active fault maps, the mapping in many regions, 335 including Africa is still patchy, and many of the underlying attributes required to use these faults in 336 PSHA is still lacking (Lines 56-58). 337 338 Line 79 (and elsewhere): I would be more careful with the suggestions that PSHA based on 339 instrumental seismicity is likely to underestimate seismicity in moderately low strain rate regions. 340 The cited references don't do a good job of backing this assertion up, which is not surprising as 341 many earthquake scientists who are not actively involved in PSHA overestimate their knowledge of it 342 (Stein being a prime offender). The justification that this study will provide better constraints on 343 earthquake rates than PSHA models that incorporate instrumental seismicity (which, when done 344 correctly, 345 is quite capable of dealing with incomplete catalogs) is cringe-inducing in light of the extremely 346 poor constraints on earthquake rates produced in this work. 347 348 The reviewer makes excellent points, and we did not wish to assert that instrumental data provide less 349 constraint than our estimates or that corrections cannot be made to incorporate incomplete catalogues 350 in PSHA. We have therefore removed the sentences that discuss whether instrumental seismicity can 351 be reliably used as a PSHA source in low strain rate settings (Lines 79-83 in the original discussion 352 paper). 353 354 Instead, we have more specifically related this section to the East African Rift (Lines 92-94), where it is 355 noted more generally that although previous PSHA has typically considered the instrumental record 356 alone (Poggi et al., 2017), prelimary studies by Hodge et al. (2015) suggest that fault source data can 357 improve the assessment of the magnitude and location of future earthquakes. Indeed, it is now 358 becoming increasingly routine for PSHA to consider fault sources (Gerstenberger et al., 2020). 359 360 Line 160: The GEM Global Active Faults Database has now been through peer review, and the citation should be changed to Styron, Richard, and Marco Pagani. "The GEM Global Active Faults 361 Database." Earthquake Spectra, Aug. 2020, doi:10.1177/8755293020944182. 362 363 364 We have updated this reference in the revised manuscript (e.g. Line 55). 365 366 Line 324: Note that the GEM neotectonics database is part of the GEM Faulted Earth project, which 367 ended around 2015, and is quite distinct from the GEM Global Active Faults Database (Styron and 368 Pagani, 2020). Please more explicitly refer to the earlier neotectonics database as the Faulted Earth 369 database for clarity. 370 371 We thank the reviewer for clarifying the distinction between the GEM Faulted Earth Project and the 372 GEM Global Active Faults Database, and in the revised manuscript we have carefully distinguish 373 between these projects (e.g. Lines 53-55). 374

Line 325: It is worth noting (but not necessarily changing the fault data or the manuscript) that the
 hierarchy developed by Christophersen et al (2015) as part of the GFE is a bit contentious and has been abandoned at GEM. The newer Global Active Faults database does not incorporate it, as I 378 decided it was too cumbersome and instead chose a 'flat' system where the 'trace' units in the GFE system would be mapped as a single, continuous trace (in most cases it's somewhat obvious that the 379 380 traces connect in the bedrock regardless of surface expression, as most faults in these databases 381 have a kilometer or more displacement which can't geologically drop to zero where the traces don't 382 quite join). This simplifies the mapping, drastically reduces the file size of the fault database, and makes for easier hazard modeling as the maximum earthquake can be calculated from the area of a 383 384 single feature rather than manual joining of multiple features. Many other institutions, such as the 385 USGS, are considering following suit if they have not done so already-the simplicity of the system allows for easier updates and more automated pipelines for incorporating faults into PSHA. 386 387 388 It is our intention that the fault databases we produce are as consistent with the GEM Global Active 389 Fault database as possible. Therefore, in the revised manuscript we have revised the South Malawi 390 Active Fault Database (SMAFD) data-only GIS file so that each fault is a single continuous GIS 'feature' 391 (Sect. 3.4; though as outlined for our response to Major Comment #1, individual faults can consist of 392 multiple GIS features in the seismogenic source database, SMSSD). 393 394 Line 383: The calculations here are another instance of what many would consider to be modeling 395 decisions rather than something incorporated directly into fault databases. 396 397 As outlined for Major Comment #1, by more clearly distinguishing between our observations and 398 modelling parameters, we will address this comment. In this case, by placing fault width as a parameter 399 in the data + estimates SMSSD (Sect. 4.1), but not the data-only SMAFD (Table 2, Sect. 4.1) 400 401 Line 639: This is not in any way a test of the results. The comparison of very broadly estimated rates 402 with data-based estimates for faults hundreds of kilometers away does not meaningfully indicate the 403 validity of the rate estimates here. 404 We recognise our use of the term 'test' here was misguided and have removed this term (Lines 594-405 406 596). Nevertheless, though these estimates are 100's of km away from southern Malawi, the tectonic 407 setting (i.e. amagmatic continental rift with border faults and intrabasinal faults) and extension rates 408 (1-3 mm/yr; Saria et al., 2014) between these two regions are comparable. Furthermore, we would also 409 argue that when taking a heavily model dependent approach to estimate slip rates and seismic hazard, 410 any 'real-world' constraints that can support our approach, even within an order of magnitude, are 411 useful; it would be worrying, for example, if the intrabasinal faults in northern Malawi had slip rates of 412 413 <0.01 or >1 mm/yr. Therefore, we still consider this a useful 'comparison.' 414 Line 651: This is also not a very meaningful comparison. The reasons that the projected date of 415 initiation of the rifting derived from geodetic data (an extrapolation of 1,000,000x) don't match 416 geologic data are manifold to the point where it may not be worth discussing; consider removing 417 this paragraph.

Unlike the comparison above which deals with fault slip rates measured over 75 Ka, we recognise this
comparison is much more uncertain and dependent on several poorly constrained parameters. We
have removed it in the revised manuscript.

Line 669: Why exactly are only half of the 128 parameter combinations considered in this? How 424 were these 'carefully selected' in a way that is not cherry picking? Computers re pretty fast these 425 days and if this analysis is worth doing (it is interesting it is worth doing with all of the

*combinations. Surely it wouldn't take more than a few seconds.*

428 As described fully in Appendix A, these combinations are not 'cherry-picked' but selected based on a 429 rigorous statistical analysis (Box et al., 1978; Rabinowitz and Steinberg, 1991) such that they provide 430 comparable results to an exploration of all the parameter space. We have clarified this in the main text 431 of the revised manuscript (Lines 605-609). 432 433 Lines 691-730: I don't think these bits of discussion add anything to the paper, and removing them would improve the focus of the paper. The digression about fault growth is interesting but not very 434 435 relevant. The second paragraph has some sloppy scholarship; the 30-60 km long normal faults here 436 are not at all on the long side of normal faults worldwide, as is clearly evident in the GEM 437 Global Active Faults Database which is cited a few times. The Jackson and White reference is very 438 out of date. 439 440 We accept the reviewer's comments that this section (Section 6.1 of the discussion paper) detracts 441 from the main focus of the study, and so will remove it from the resubmitted manuscript. This has also 442 created space for further discussion on incorporating the databases into PSHA (Sect. 6.3), without 443 lengthening the paper. 444 445 The paragraph on seismic risk is important but could be tightened up and placed in the introduction, 446 where it is more appropriate. The next paragraph, comparing the lengths of faults in this database 447 to earthquakes also suffers a bit because it compares a small number of global earthquakes to a 448 local fault database, which isn't a good point of comparison (longer normal faults exist in several 449 orogens and generally have similarly slow slip rates, i.e. the Basin and Range in the US). 450 451 As requested by the reviewer, in the resubmitted manuscript we have revised a comparison of fault 452 lengths in Malawi to those of other normal faults from the Global Earthquake Model Global Active Fault 453 Database (Styron and Pagani, 2020), as opposed to specifically just normal fault earthquake ruptures 454 (Lines 636-638). 455 456 Line 758: This paragraph is troubling. It seems to discourage others from attempting to collect real 457 data to use in PSHA, though there is no reason to think that the rough estimates provided in this 458 work are superior to field measurements. 459 460 It was certainly never our intention to discourage the collection of on-fault data to feed into active fault 461 databases. In the revised manuscript, we have emphasised that such data should still be collected 462 (Lines 655-656) and that our approach is most appropriate in places like southern Malawi precisely 463 because there is currently no paleoseismic data (e.g. Line 317). Furthermore, a motivation of this study 464 was to identify where future data collection should be targeted, with the collection of paleoseismic data 465 clearly highlighted, along with tighter geodetic constraints, as a priority area (Sect. 6.2). 466 467 Line 798: The probability distributions listed here describe aleatory variability in recurence, but the 468 topic under discussion is epistemic uncertainty. In this case these are not comparable. 469 470 We have removed the discussion on these types of probabilities distributions 471

**473 Reviewer 2474**

This manuscript presents a new systematic approach useful for parametrizing seismic hazards in 475 476 areas with limited instrumental seismicity. The study was carried out in the southern part of Malawi, 477 and documents the large faults that are capable of accommodating medium-large magnitude 478 earthquakes in the region, as well as the attributes of these faults that are relevant for the hazard 479 analysis. Also, the study discusses both the seismic hazard and tectonic implications of the results, 480 as well as the uncertainties in the estimates. I believe that this approach is great and useful in active 481 plat boundary settings where there is poor earthquake monitoring infrastructure. Such settings abound in several continents, and seismic instrumentation is expensive; thus, necessitating a need 482 483 for creative, less expensive approaches as presented in this study. The manuscript is well written 484 and easy to read. I believe that this manuscript contains material fit for publication in EGU Solid 485 Earth.

However, I believe this manuscript could appropriate for publication in the journal after moderate 488 revisions to the paper. I'm recommending moderate revision because of the issues I consider to be 489 major flaws in the interpretation of the tectonic domains and associated structural elements in the 490 study area, which directly impact either the input data or specific features of the implementation of 491 the analysis performed in the study. Here below, are the 7 major issues I have with the manuscript, 492 and 2 comments/questions that I think the authors could consider incorporating into the discussion 493 part of the manuscript. Also, I made comments in different parts of the text that are either minor 494 corrections/comments, or are related to the major issues stated below (see attached an annotated 495 pdf).

Regards, Folarin Kolawole

**499 Major Issues**

Major Issue 1: The interchanging use of "southern Malawi" and "southern Malawi Rift". These two 502 terms should not be used interchangeably in this text as it can bring confusion. "Southern Malawi" 503 refers to a geopolitical region that hosts rift segments of different tectonic affiliations; whereas, 504 "Southern Malawi Rift" refers to the southernmost segment of the Malawi Rift which includes only 505 the Makanjita Trough & Malombe Graben (bifurcation around the Shire Horst), and the Zomba 506 Graben. This interchanging use occurs at too many parts of the manuscript, so I decided to just 507 mention it here instead of commenting on it in the text (attached pdf). This issue also leads to and is 508 related to my Major Issue 2...see below. 509

We acknowledge our interchangeable use of 'southern Malawi' and 'the southern Malawi Rift' will be 511 confusing to readers not familiar with the area, and further adds to the confusion in the various ways 512 that the southern end of the Malawi Rift has been defined previously (Chapola and Kaphwiyo, 1992; 513 Chorowicz and Sorlien, 1992; Ebinger et al., 1987; Laõ-Dávila et al., 2015). In our revised submission, 514 we have carefully outlined that the database covers the geopolitical region of southern Malawi, as 515 opposed to the southern 'Malawi Rift' (albeit with the necessity that it will include some faults that 516 extend into Mozambique, Lines 107-113). This choice also reflects that seismic hazard is typically 517 considered at a national level, and so it makes sense that active fault databases are defined by national, 518 and not geological, boundaries, with the necessary exception that faults that cross geopolitical 519 boundaries are included. 520

Major Issue 2: Definition of principal grabens of the southern Malawi Rift. The authors identified 522 the graben "Lower Shire Graben" as a principal graben of southern Malawi Rift (pg 19 lines 458-523 459). I have issue with the characterization of this graben as a tectonic element of the Malawi Rift. 524 This is very misleading as this Lower Shire Graben is a sub-basin in the Shire Rift, not the Malawi 525 Rift (Castaing, 1991). I understand that this graben is located within the Malawi geopolitical 526 boundary, whereas most of the other sections of the Shire Rift are located in Mozambique. However, geopolitical location does not automatically make this graben a part of the Malawi Rift. Moreover, 527 528 the Shire Rift has a distinctly different structure, orientation, and tectonic history from those of the 529 southern Malawi Rift. Shire Rift is a multiphase rift basin (Mesozoic-Cenozoic; Castaing, 1991), 530 whereas, southern Malawi Rift is Late Cenozoic (e.g., Wedmore et al., 2019; Scholz et al., 2020). 531 Infact, exposed basement highs separate the Zomba Graben (which is at the southernmost tip of the 532 Malawi Rift) from this Lower Shire Graben and the other sections of the Shire Rift. Therefore, in order not to confuse a reader, I'll suggest that the authors use the term "southern Malawi" in the 533 534 context of describing the location of the 'Lower Shire graben' (i.e. use geographical description), 535 rather than the term "southern Malawi Rift". 536 537 As discussed above with reference to Major Issue 1, by explicitly outlining that the South Malawi Active 538 Fault Database (SMAFD) and the South Malawi Seismogenic Source Database (SMSSD) cover the 539 political region of southern Malawi, not the southern 'Malawi Rift,' we have addressed this comment 540 (Lines 107-113). Major Issue 3: The descriptions of the "Makanjira Graben" in the manuscript and the modelling 541 542 done in Fig.A3a shows that the authors consider that term to incorporate both the Makanjira 543 Trough and Malombe Graben. The Malawi Rift bifurcates around the Shire Horst into these two 544 segments and further south, they link-up and transition into the Zomba Graben. The Makanjira 545 Trough is bounded to the west by Chirobwe-Ncheu Fault, and to the East by Shire Horst, whereas 546 the Malombe Graben is bounded to the west by the Malombe Fault and to the east by the Mwanjage 547 Fault. The surface+subsurface structure of this section of the Malawi Rift (Lao-Davila et al., 2015) 548 shows that the Malombe Graben has a greater hanging wall subsidence and thus, border fault offset than the Makanjira Trough. Here are the evidences: 549 550 551 1.) The floor of the Makanjira trough is mostly dominated by exposed basement, whereas, the 552 Malombe Graben is relatively better developed graben structure with a wider-spread 553 sediment accumulation and even a lake development at the foot of its border fault. The zone 554 of sediment accumulation on the northern half of the Makanjira trough is associated with 555 subsidence along the southern extension of the N-S trending eastern border fault of the 556 Nkhotakota Segment of the Malawi Rift (for location of Nkhotakota Segment see Lao-Davila 557 et al., 2015; for the described subsidence and fault location, see Fig.5b of Scholz et al., 558 2020). 559 560 2.) The floor of the Makanjira half-graben is at a higher elevation compared to that of the 561 Malombe Graben, indicating that subsidence is most-likely greater in the Malombe Graben. 562 For reference see the across-rift profiles in Figs.4L-4M of Lao-Davila et al. (2015). I am 563 guessing that the authors consider that because the Chirobwe-Nchue Fault has a higher guessing that the dathors consider that because the Chirobwe-Nende Fault has a higher
 footwall elevation/escarpment along the rift section, therefore, it must have the largest
 throw. If that is the consideration upon which the border fault definition and Fig.A3a model
 are based, I refer the authors to the Rukwa Rift where border fault footwall elevation/uplift is
 not representative of the subsurface fault throw (Morley et al., 1999). Thus, based on the observed geologic structure, I think the model in Fig.A3a is problematic because it ignores 569 the presence of the fault with the larger offset and hanging wall subsidence at the so-called 570 "Makanjira Graben". Also, the model assumes the Chirobwe-Ncheu to has the greatest 571 offset/hanging wall subsidence along the profile which is not representative of the 572 distribution of subsidence across this section of the rift (Figs. 4L-4M in Lao-Davila et al., 573 2015). Therefore, in my opinion, if possible, I think this model needs to be revised. If 574 impossible due to modelling limitations, then it should be stated. 575 576 We firstly recognise that the hanging-wall flexural modelling in a discussion manuscript was based on 577 several large assumptions and have removed this from the revised manuscript. Nevertheless, the point 578 remains due to thick elastic crust, and comparatively small border fault throws (<1000 m), the amount 579 of hanging-wall flexural extensional strain in southern Malawi is negligible (Lines 426-429; Wedmore 580 et al., 2020a) 581 582 We address the reviewer's concerns on how we define border faults and intrabasinal faults further 583 below (Major Issue #4). In the context of the Malombe Fault, as the reviewer correctly points out, the Shire Horst divides the rift section that we term the Makanjira Graben. However, though this structure 584 585 may have been important in the tectonic evolution of the rift (Laõ-Dávila et al., 2015), for the reasons 586 outlined below we do not consider that it strongly influences the current distribution of extensional 587 strain in the Makaniira Graben. 588 589 With respect to the reviewer's first set of concerns, we disagree that the 'Makanjira trough' is 590 dominated by exposed basement with any sediment accumulation necessarily related to subsidence 591 along the Nkhotakota rift segment: (1) geological maps and boreholes indicate that sediments have not 592 just accumulated along the northern half of the Makanjira trough, but along its entire length (Dawson 593 and Kirkpatrick, 1968; Walshaw, 1965; Fig. 1b of our discussion paper) and including to the south of the Nkhotakota fault as mapped by Scholz et al., (2020), (2) there is geomorphic evidence of recent 594 595 multiple earthquakes along the Bilila-Mtakataka Fault (BMF; Hodge et al., 2018, 2019, 2020; Jackson 596 and Blenkinsop, 1997), indicating that this is a highly active part of the rift capable of creating 597 accommodation space for sediment accumulation, (3) boreholes indicate that these sediments in this 598 section of the rift thicken to the west against the BMF (Dawson and Kirkpatrick, 1968; Walshaw, 1965), 599 indicating that it is the BMF not the Nkhotakota fault that is primarily generating accommodation 600 space, and (4), where there is exposed basement in the hanging-wall of the Chirobwe-Ncheu fault, this 601 can be related to footwall uplift of the interior BMF (Lines 231-235). 602 603 With respect to the reviewer's second concerns, the higher elevation of the Makanjira trough relative to 604 the Malombe trough does not require that these should be separated as distinct basins. There are other 605 (albeit smaller) horst structures and basement highs that the Malawi Rift bifurcates around in the 606 Central and North Basins of Lake Malawi (Ebinger et al., 1987; Scholz et al., 2020; Shillington et al., 607 2020). These also result in complex across-rift topography; however, they have not necessitated the division of the rift across strike. We suggest too that the formation of Lake Malombe does not require 608 609 that the Malombe Fault has accommodated considerable throw as: (1) it only extends across the 610 northern section of the fault and (2) it has a maximum depth of 5 m (Weyl et al., 2004). Indeed, this 611 part of the rift has very little variation in subsidence with the Shire River experiencing only a 1.5 m drop in elevation in the 85 km distance between Mangochi and Liwonde; (Dulanya, 2017). 612 613 614 We agree with the reviewer that ideally subsurface data should be used to characterise the structure of we agree with the reviewer that therein substrate data should be used to that acterise the structure of the Malawi Rift. However, south of Lake Malawi, such data are scarce, and in their absence, we prefer to use the data we *do* have from the rift's topography and basement-penetrating boreholes to characterise
its structure (Fig. S1; Bloomfield, 1965; Bloomfield and Garson, 1965; Walshaw, 1965). Cumulatively,
we suggest that these observations indicate that the Malombe Fault should be considered as an
intrabasinal fault (see Major Issue #4), albeit it could be one with considerable displacement (>500 m).

In this context, it could be considered similar to some of the high displacement horst-forming 621 intrabasinal faults (up to 2.5 km throw) in Lake Malawi (Scholz et al., 2020; Shillington et al., 2020). 622 623 To further demonstrate how we have used topography and borehole data to characterise the rift's 624 structure, in the revised manuscript we have included across-rift cross sections for each basin in 625 southern Malawi (Fig. 8). For the Makanjira Graben cross section, as highlighted by the reviewer, we 626 have also noted the Shire Horst structure and the lower elevation in the eastern side of the graben, as 627 these structures were not described in sufficient detail in the discussion paper (Lines  $524_{426}$ ). Also note that Figure 4L-M in Lao-Davila et al. 2015 would not be a good reference for such a figure, as 628 629 although they suggest a  $\sim$ 500 m thick sequence of sediments in the Malombe Trough, it is not clear 630 what evidence they have for this assertion. 631 632 Major Issue 4: Age of the Thyolo Fault and modelling of strain in Lower Shire graben (Fig. A3c and 633 pg 37 lines 895-896, pg 38 lines914-915). The authors suggest that the Thyolo Fault is Karoo age. 634 There is no evidence suggesting that there exists karoo-agesedimentary or volcaniclastic deposits on 635 the hanging wall of the Thyolo Fault (Habgood, 1963; Habgood et al., 1973). Mesozoic activity 636 along the Thyolo Fault would require subsidence of its hanging wall and creation of accommodation 637 space for the deposition of volcanic and sedimentary sequences. Both Habgood (1963) and Castaing (1991) suggested that the Mwanza-Namalmbo Fault system is the eastern border fault of the 638 639 Mesozoic Shire Rift. Castaing (1991) suggested that the Thyolo Fault is Cenozoic, bounding the 640 currently active eastern domain of the Shire Rift. Therefore, I think this idea of Thyolo Fault being a 641 Karoo fault needs to be revised except the authors provide data showing the presence of Mesozoic deposits on the hanging wall of the Thyolo Fault. 642 643 644 As discussed for Major Issue #3, we have removed the hanging-wall flexure modelling in the revised 645 manuscript. Therefore, there are no parts in this manuscript that explicitly discuss the age of the 646 Thyolo Fault, nor is this study an appropriate place to do so. The key point is that it is active. 647 Major Issue 5: Definition of "border fault" in southern Malawi (Fig. 2a). The Lisungwe Fault, 648 649 Malombe Fault, and Mwanza Faults are excluded from the 'border fault' definition and I am not particularly sure why. This is an issue for me, particularly because the structure of the basins point 650 651 directly to the essence of these faults. For example, the Malombe Fault is the principal border fault 652 of the Malombe Graben, not the Mwanjage Fault which you've assigned as the main border fault. 653 Even the distribution of the amplitudes and wavelengths of the magnetic fabrics beneath the Malombe Graben in Fig.2c (Laõ-Dávila et al., 2015) clearly shows that the hanging wall of the 654

Malombe Fault has significantly larger subsidence than that of the Mwanjage Fault. Also, the

Mwanza Fault is a major border fault of the NW half of the Shire Rift (as shown in the maps in 657 Figure 2). There is also evidence that the exposed segment of the Mwanza fault has been reactivated 658 in the Cenozoic given by accumulation of Quaternary sediments on its hanging-wall (Habgood in the Cenozoic given by accumulation of Quaternary sediments on its hanging-wall (Habgood
1963).

We thank the reviewer for bringing these points to our attention, and for demonstrating that the classification of border faults and intrabasinal faults is not always as clear-cut as the manuscript 662 663 suggests in its current form. In the resubmitted manuscript, we have included a section where we more 664 explicitly define the difference in border and intrabasinal faults, and how this was applied to southern 665 Malawi (including the use of geological cross sections as outlined for Major Comment #3). We define 666 border faults using the simplest geometric criteria: the fault at the edge of the rift's surface expression (Lines 399-403). In other words, this definition is purely based on the geometry and distribution of 667 668 brittle deformation across the rift. This definition is not inconsistent with differences between border and intrabasinal faults noted in previous studies (e.g. cumulative offset, slip rate, length; Agostini et al.,
2011; Ebinger, 1989; Gawthorpe and Leeder, 2000; Muirhead et al., 2019; Wedmore et al., 2020b), but
equally this definition is not dependent on these factors.

To specifically reply to the reviewer's comments on individual faults: the justification for not including the Malombe Fault as a border fault is discussed in Major Comment #3. With regards to the Mwanza 674 675 Fault, we agree that it has been active during the East African Rifting, hence its inclusion in the SMAFD. 676 Nevertheless, on the basis of the reviewer's comments and more recent mapping by Daly et al., (2020) 677 that suggests this part of the rift may extend further into Mozambique and the Lower Zambezi Rift 678 where it forms a different microplate boundary (Angoni-San), we now consider the Mwanza Fault as 679 the border fault of a different rift section to the Lower Shire Graben (Lines 140, 547, and Fig. 8e). 680 Unfortunately, in this case there are no geodetic constraints on the extension rate across the Zambezi 681 Rift. In this case, we have calculated fault slip rates using values of between 0.2-1 mm/yr (Table 3, 682 Lines 455-458), where the lower estimate represents the minimum strain accrual measurable by 683 geodesy (Calais et al., 2016) and the upper estimate represents that extension rates in the Zambezi Rift 684 are unlikely to be higher than in the Lower Shire Graben given that the Mwanza Fault has only 685 accumulated EAR sediments along its south-eastern most extent (Habgood, 1963). 686 687 The topography at the western edge of the Zomba Graben, where it grades into the Kirk Plateau, is very complex and so it is difficult to fit the Zomba Graben into conventional half-graben/graben models 689 (Wedmore et al., 2020a). In particular, there are a number of N-S trending deeply incised valleys that 690 lie to the west of the Lisungwe Fault and which have been previously interpreted as 'rift valley faults' 691 (Bloomfield and Garson, 1965, see Figure 1 below). In addition, there are a number ENE-WSW trending vallevs that are interpreted as 'cross faults' (i.e. strike-slip) faults (Bloomfield and Garson, 1965). 692 693 Though only one of these faults (the Wamkurumadzi Fault) meet our definition of being active and is 694 included in the SMAFD (section 3.1 of the discussion paper), inclusion of this fault, and the generally 695 complex topography, requires that the Lisungwe Fault does not meet the definition of the border fault 696 as outlined above. In the revised manuscript, we have more carefully outline how we have come to this 697 decision and hope that this study will stimulate further studies into the question of fault activity at the 698 western edge of the Zomba Graben (Lines 528-539, Fig. 8b).

Figure 1: Fault map for Zomba Graben underlain by TanDEM-X 12 m resolution digital elevation model. 702 'Inactive faults' as mapped by Bloomfield and Garson, (1965). TanDEM-X data was obtained via DLR 703 proposal DEM\_GEOL0686. 704 Major Issue 6: Related to "Issue 4" above, the authors classified Namalambo Fault as an active 705 706 Fault (e.g., Figs. 1b & 2). Namalambo Fault cannot be classified as an East African Rift System 707 Fault because there is no evidence supporting its Cenozoic reactivation (see Bloomfield, 1958; 708 Habgood, 1963; Castaing, 1991). The mentioned geological reports specifically stated that there is 709 no Quaternary sediment deposition on top of the karoo sedimentary units at the base of its scarp, 710 suggesting it has not been reactivated in the Cenozoic. Also, this fault does not satisfy the criteria 711 stated by the author in Pg11 lines 260-267. Are the authors including it because they're assuming 712 that it could be reactivated sometime in the future? If yes, I think this should be stated in the relevant 713 figure captions and in the manuscript (particularly because this fault is a prominent fault in the 714 area, and could be confusing to a reader without this additional information). 715 716 Although we noted in the database that the Namlambo Fault formed during Karoo-age rifting, we 717 accept the reviewers comments that there is very little evidence for its reactivation during East African 718 Rifting, and so in the resubmitted manuscript we have removed this fault from the SMAFD and include 719 it in the 'Malawi OtherFault' database instead. 720 721 Major Issue 7 (minor): A 'declaration' that I think is not clearly made in the set-up of the premise of 722 the manuscript is that the parameterization approach focuses on tectonically active continental 723 settings. I do not think that the authors imply that the approach is applicable to relatively more 724 stable intraplate settings where much lower strain rates generally abound and potentially dangerous 725 faults are buried, although some of those areas could be seismically active (e.g., intraplate induced 726 seismicity). Therefore, I'll suggest that the authors state this clearly, at least in the abstract and 727 introduction sections of the paper. I noticed that in different parts of the text, it is subtly implied with 728 phrases relating to plate boundary, interplate setting etc., however, I think it will be beneficial to the 729 reader if this is stated clearer from the onset. 730 731 The reviewer is correct that the approach outlined for characterising seismic hazard is most applicable 732 to low strain rate regions (plate boundary slip rates 0.1-10 mm/yr; Scholz et al 1986), which are 733 distinct from both high strain rate regions (plate boundary slip rates >10 mm/yr) and stable cratons 734 (plate boundary slip rates <0.1 mm/yr). We have stated this in the revised manuscript (Lines 84-85). 735 Question/Comment 1: Pg29 lines 703-707. The authors highlighted the anomalously large 736 737 seismogenic thickness of the southern Malawi area, with continuous 30-60 km long fault sections. 738 Crustal thickness map of southern Malawi (Njinju et al., 2019a) shows that an unusually thick crust 739 dominates the area. In addition, heat flow map of the same area (Njinju et al., 2019b) shows an 740 anomalous thermal gap in the area. Both of these have been associated it with an eastern extension 741 of the Niassa Craton. Do you think that there is a possibility that these have an impact on the 742 observed seismogenic thickness? 743 744 The contribution of low heat flow to the anomalously thick seismogenic crust in southern Malawi is 745 acknowledged (Line 165), however, we thank the reviewer for the suggested reference, which we have 746 included in the revised manuscript. It is also worth noting that Fagereng, (2013) demonstrated that an 747 anomalously low heat flow alone  $(63 \text{ mW/m}^{-2})$  is not sufficient to explain the thick seismogenic crust in 748 this region, with other factors such as strain rate and crustal composition also important. Furthermore, 749 the seismogenic crust is unusually thick throughout Malawi (>30 km), even in places where the heat flow is higher (e.g. 65-70 mW/m-2 in Karonga; Ebinger et al 2019).

| 751                                                  |                                                                                                                                                                                                                                                                                                                                                                                                                                                                                                                                                                                                                                  |  |
|------------------------------------------------------|----------------------------------------------------------------------------------------------------------------------------------------------------------------------------------------------------------------------------------------------------------------------------------------------------------------------------------------------------------------------------------------------------------------------------------------------------------------------------------------------------------------------------------------------------------------------------------------------------------------------------------|--|
| 752
| Question/Comment 2: It is well-known that the patterns of seismogenic fault reactivation are
influenced by the frictional stability of faults. Asides from strain rate and lithology/mineralogical
composition, another important factor that influence the frictional stability is geothermal
gradient/heat flow. Well-constrained heat flow & geothermal gradient maps of southern Malawi
(Njinju et al., 2019b) show interesting thermal anomalies within the areas analyzed in this study. I
am curious as to how the heat flow distribution in the area may affect the results/conclusions of this
study. |  |
| 760
| Although the reviewer raises an interesting point, as noted above there are other factors beyond heat
flow that will control fault reactivation in this region. We consider that to discuss them all will go
beyond the scope of this study, which is focussed on active faulting, not crustal rheology. Furthermore,
classic Mohr-Coulomb theory suggests that brittle fault reactivation is temperature independent
(Sibson 1985), although it will partially influence thickness of the seismogenic crust (as discussed
above).                                                                                |  |
| 768                                                  | Comments on Reviewer Supplement                                                                                                                                                                                                                                                                                                                                                                                                                                                                                                                                                                                                  |  |
| 769
            | Line 31: This sentence, as it is written here, implies circular logic. This is because measurements of fault length is one of the inputs into your model estimates of EQ magnitude. How about "These potentially high magnitudes for continental normal faults are compartible with the observed 11-140 km-long faults and thick (30-35 km) seismogenic crust of southern Malawi."                                                                                                                                                                                                                                               |  |
| 774
                                 | We have removed this sentence from the abstract                                                                                                                                                                                                                                                                                                                                                                                                                                                                                                                                                                                  |  |
| 776
                          | Line 61: The repetition of "estimate" sound awkwardcould reword as "fault slip rates can be estimated using geodetic constraints"?                                                                                                                                                                                                                                                                                                                                                                                                                                                                                        |  |
| 779
                                 | We will correct this in the revised manuscript to geodetic 'data' (Line 86)                                                                                                                                                                                                                                                                                                                                                                                                                                                                                                                                                      |  |
| 781
     | Line 87: I suggest a rewording of this text. Southern Malawi lies NEAR the southern incipient end of the EARS, not "AT" the end.
Rather, it is Central/Southern Mozambique that lies AT the southern incipient end of the EARS and has all those characteristics you've mentionedsee papers on the MOZART project (e.g., Fonseca et al. (2014), Urema Graben, Mazenga Graben, Chissenga-Urema Graben System, and the Changani Graben System (Mueller & Jokat, 2019).                                                                                                                                                          |  |
| 787
                   | We have corrected this in the revised manuscript and outline that southern Malawi lies near and/or towards the southern end of the EARS (e.g. Line 116).                                                                                                                                                                                                                                                                                                                                                                                                                                                           |  |
| 791
            | Line 112: I think the authors should include the Salambidwe Igneous structure (Cooper, 1961) and
the flood basalts associated with the Lupata Volcanic Complex (footwall and hanging wall of the
Panga Fault; Habgood, 1963).
We thank the reviewer for noting these omissions. However, we have removed the section so this
sentence was not included in from the revised manuscript.                                                                                                                                                                                                                               |  |
| 796                                                  |                                                                                                                                                                                                                                                                                                                                                                                                                                                                                                                                                                                                                                  |  |

Line 123: By definition, can a graben can be bounded by 1 border fault? I suggest need rewording 798 We have replaced 'graben' with 'basin' and/or 'half-graben' in the revised manuscript where 799 appropriate (e.g. Line 398) Lines 136-144: This paragraph is written with a mixed context that could create confusion...i.e. 800 written with a context of a political territory (i.e. southern Malawi) and a rift basin (Malawi Rift). I 801 802 will say this is very 'dangerous' as it propogates a very common problem with the way geology has 803 been carried out in Africa for a very long time. Therefore, I will suggest that the authors stick to 804 "Malawi Rift" since this study and this particular paragraph focuses more on tectonic history. 805 806 On the aspect of the evolution of the Malawi Rift, I will refer the author to Scholz et al. (2020) which demonstrates clearly the southward episodic propagation of the Malawi Rift. In addition, studies 807 808 have showed that Karoo sandstones outcrop along the Karonga border fault of the Karonga Basin, 809 and Accardo et al. 2018 observed an anomalous velocity interval that suggests highly lithified 810 Karoo sedimentary rocks directly overlying the basement beneath the Karonga and Usisya Basin 811 fill. 812 As outlined for Major Issue #1 we have more carefully defined the term 'Malawi Rift' in our revised 813 814 manuscript and note that the databases cover the political region of southern Malawi, not the Malawi 815 Rift (Lines 107-113). We have also incorporated the Scholz et al (2020) reference into the revised 816 manuscript, however, note that this study provides age constraints for the southern end of Lake 817 Malawi, not southern Malawi itself (Lines 122-124). 818 819 Lines 146-150: The inferences made in this section, as written, sounds highly speculative, and since it is placed in the "geologic setting" section, I will suggest rewording. First, the text totally ignores 820 821 the Malombe Graben as the primary hydrologic linkage between the Lake Malawi and the Zomba 822 Graben (i.e. hosts the axial stream). Second, the text implies that sedimentation in the grabens 823 referred to are only associated with flooding episodes. This is very strange to me. The area 824 described is defined by the southward bifurcation of the Malawi Rift into two narrower branches: a 825 graben in the east (Malombe Graben), another graben to the west (Makanjira Graben). The Malombe Graben has a well-developed lake from which the axial stream Shire River flows 826 southwards. South of the bifurcation, the two branches merge back into a weakly-extended graben 827 828 (Zomba Graben) south of which the basement is exposed and faulting is diffused. The point here is 829 than the bifurcation troughs are actively subsiding, fault-bounded tectonic elements with 830 structurally-controlled axial (Shire River) and transverse streams (from the rift flanks) channelling 831 sediments into the subsiding basins. Thus, it is more likely that both faulting and climate control sedimentation within these basins, and not 'only climate' (as it is described here). For general 832 833 reference on interactions between faulting and climate in humid rift settings, I refer the authors to 834 Gawthorpe & Leeder (2000). 835 836 In the revised manuscript we have noted that faulting in the rift will have influenced sedimentation (Lines 211-214). Nevertheless, it should be noted that base level changes in Lake Malawi are thought to 837

- be primarily driven by climatic forcing (e.g. Scholz et al 2007; Lyons et al 2015). 838
- 839
- 840 Lines 151-152: What does this sentence mean? ... you mean the steep gradient of the rift floor does
- 841 not correspond to a fault escarpment? As this sentence is written, it precludes every form of 842
- structural control on the gradient...and I am highlighting this because this particular zone is a
- 843 transfer zone between the Malawi Rift and the currently active part of the Shire Rift. Transfer zones
- in areas of incipient rifting are typically characterized by elevated/exposed basement...for reference, 844
- see Heilman et al. (2019) and Gawthorpe & Leeder (2000). 845

This sentence refers to the point that no active faults were identified in the region between the Zomba 848 and Lower Shire Graben from fieldwork and analysis of high resolution digital elevation models (although we of course cannot exclusively prove that there are no active faults in this region). We have 849 850 clarified this in the resubmitted manuscript (Lines 541-543). Note too that this sentence does not 851 preclude that this gradient may reflect pre-existing topography from previous phases of deformation 852 (e.g. Karoo). 853 Lines 157-158: First, I think it should also be added that the Castaing (1991) mapping was infact, 854 855 only limited to the Shire Rift part of southern Malawi...the mapped faults in the paper did not extend 856 into Southern Malawi Rift. 857 858 Second, I have seen a lot of these faults mapped previously in the Malawi Geological Survey reports 859 (e.g., Bloomfield, 1958; Habgood, 1963; Habgood et al., 1973)... some of which were cited in this 860 manuscript. I could see that the authors mentioned some of these faults on pg 12, however, I think 861 the contributions should also be acknowledged here. 862 863 We will make it clearer in the revised manuscript that Castiang (1991) only considered EARS faults in the Lower Shire (Line 143-144). With respect to the second point, we now also discuss the Malawi 864 865 Geological Survey Reports in this section (Lines 146-150), though note that fault traces mapped in 866 these reports were not included in the Global Earthquake Model global active fault database 867 (Christophersen et al., 2015; Styron and Pagani, 2020). 868 Line 161: The mapping in the Geological Survey reports looks pretty fine-scale to me because they 869 870 were done on the field. I think the detailed info on slip rates and recent faulting are the parts that are missing which this current study provide. 871 872 873 We are referring here specifically to the Global Earthquake Model global active fault database here (Fig. 2a; Christophersen et al., 2015; Styron and Pagani, 2020), which do not incorporate the faults mapped 874 875 by the Malawi Geological Survey, but those mapped by Macgregor, (2015). We have clarified this in the 876 revised manuscript (Line 146) 877 Line 178-179: This sentence sounds weird with the "to 1965". Pls check 878 879 We will revise this sentence in the resubmitted manuscript as requested (Line 161). 880 881 Lines 181-188: Wondering if it will also be worth mentioning the crustal thickness in southern 882 Malawi (Njinju et al., 2019a; Tectonics), and heat flow distribution in southern Malawi (Njinju et 883 al., 2019b; J. Volc. & Geoth. Res.) 884 885 Njinju, E.A., Atekwana, E.A., Stamps, D.S., Abdelsalam, M.G., Atekwana, E.A., Mickus, K.L., Fishwick, S., Kolawole, F., Rajaonarison, T.A. and Nyalugwe, V.N. (2019a). Lithospheric structure 886 887 of the Malawi Rift: Implications for magma-poor rifting processes. Tectonics, 38(11), pp.3835-3853. 888 889 Njinju, E.A., Kolawole, F., Atekwana, E.A., Stamps, D.S., Atekwana, E.A., Abdelsalam, M.G. and 890 Mickus, K.L. (2019b). Terrestrial heat flow in the Malawi Rifted Zone, East Africa: Implications for 891 tectono-thermal inheritance in continental rift basins. Journal of Volcanology and Geothermal 892 Research, 387, p.106656. 893 894 As outlined for Question/Comment 1 we have incorporate these references into the revised manuscript 895 (Line 167). 896

Lines 202-203: True. However, another possibility that could be mentioned here is local stress 898 rotations (see Morley, 2010...case study on the Rukwa Rift). 899 900 As discussed in Williams et al., (2019) stress rotations do not explain the discrepancy in extension 901 direction when inferred from geodesy or earthquake focal mechanisms in southern Malawi, as the 902 orientation of recent joint sets in the region is uniform across the rift suggesting that the regional stress 903 field is uniform. Furthermore, the hypothesis of Morley, (2010) is that that stress rotations reflect 904 changes in foliation orientation, in which case it would not account faults locally cross cutting the 905 foliation in southern Malawi. 906 907 Line 237: also depth of medium-large magnitude EQ ruptures? 908 We have included focal depth as a factor of whether an earthquake ruptures to the surface in the 909 revised manuscript (Line 203) 910 Line 250: "little" sounds awkward here, because how little is "little"? I'll suggest "limited" instead 911 912 We have clarified this in the revised manuscript (Lines 218-220), and note that there is some limited 913 dating from <10 ka sediments around Lake Malombe (Van Bocxlaer et al., 2012). 914 915 Lines 458-459: See comments in 'major issues'. 916 See reply to comment on Major Issue 3 917 918 Lines 464-464: Isn't this graben is known as the Urema Graben/Rift (Castaing, 1991; Fonseca etal. 919 2014; Lloyed et al., 2019). Why give it a different name? 920 921 Castaing, C. (1991), Post-Pan-African tectonic evolution of South Malawi in relation to the Karroo 922 and recent East African rift systems. Tectonophysics, 191(1-2), pp.55-73. 923 924 Fonseca, J.F.B.D., Chamussa, J., Domingues, A., Helffrich, G., Antunes, E., van Aswegen, G., Pinto, 925 L.V., Custódio, S. and Manhiça, V.J., 2014. MOZART: A seismological investigation of the East 926 African Rift in central Mozambique. Seismological Research Letters, 85(1), pp.108-116. 927 928 Llovd, R., Biggs, J. and Copley, A., 2019. The decade-long Machaze–Zinave aftershock sequence in 929 the slowly straining Mozambique Rift. Geophysical Journal International, 217(1), pp.504-531. 930 931 As for the Malawi Rift, there is little consensus on the extent of the Urema Graben (see also, Steinbruch, 932 (2010) which suggest it refers to mainly the basins around the river Urema 150 km along strike to the 933 south). Our preference here is to avoid using this term as it may imply that our fault mapping covers 934 the full extent of the Urema Graben, and this will be clarified in the revised manuscript (Line 138). 935 936 Lines 473-474: Does this estimate include subsurface measurement of the total throw at the hanging 937 wall cut-off of the faults? Please, provide reference. Also, based on the wording, does this refer to 938 southern Malawi Rift (excluding the 'Lower Shire Graben') or does it refer to all the faults in 939 southern Malawi geopolitical boundary? 940 Yes, this estimate includes the limited subsurface data that does exist in southern Malawi (i.e. 941 groundwater boreholes, Fig. S1; Bloomfield, 1965; Bloomfield and Garson, 1965; King and Dawson, 942 1976; Walshaw, 1965), and we have clarified this in the revised manuscript (Lines426-428). 943 944 Lines 687-689: For the sake of the reader, I think the hazard part should be stated before 945 implications for continental rift as seismic hazard is the primary focus of this study. Thus, I'll suggest a rewording to: "In the following section, we examine some key results of the SMAFD in

- 947 terms of its implications for seismic hazard in southern Malawi, its contribution to our
- 948 understanding of fault growth in continental rifts, and future strategies to...".
- As discussed with respect to the comments for Lines 690-711 from Reviewer #1, we will be removing
- 950 this section on 'controls on fault growth in southern Malawi' in the revised manuscript. 951
- Line691: This relates to my comment above... While this section is important and relevant, for the
  reader, it seems like a sudden digression from the seismic hazard story to bring it in at this early
  part of the discussion. Pls consider swaping it with '6.2 Implications for seismic hazard in southern
  Malawi'.
- 956 See our reply to the previous comment.
- 958 Line 694: Could also include Scholz et al. (2020)
- Scholz, C.A., Shillington, D.J., Wright, L.J., Accardo, N., Gaherty, J.B. and Chindandali, P., 2020.
  Intrarift fault fabric, segmentation, and basin evolution of the Lake Malawi (Nyasa) Rift, East
  Africa. Geosphere.
- We thank the reviewer for bringing this article to our attention and have incorporated it into the
  revised manuscript e.g. Line 124).
- 967 Line 843: growth,
- 968 Corrected (Line 717) 969
- Line 895-896: This statement, as written, is misleading. Please revise. Castaing (1991) argued that the Thyolo Fault is Cenozoic and along with the reactivated Mwanza Fault, is accommodating strikeslip in present day. He suggested that the Mwanza Fault was the primary eastern Karoo and
- Sup in present day. The suggested that the Mikanza Fadit was the primary easiern Karob and
   Cretaceous border fault of the Shire Rift. See pages 65-66 and figs. 7,9,&10 of Castaing et al.
   (1999).
- 97<del>4</del> 975

As discussed for Major Issue #3, we have removed the hanging-wall flexural modelling from our
revised manuscript, so this sentence was removed anyway.

Lines 897-899: This statement, as written, is speculative. As I explained in my "Major issues 1 & 2", 980 the Shire Rift is a multiphase rift of which the Lower Shire graben is the easternmost sub-basin. The 981 Malawi Rift and Shire Rift have different tectonic histories and structure, and only linked up at the 982 current location of southern Zomba Graben. Although the development of the Lower Shire graben 983 could be syn-tectonic with the southward propagation of the Malawi Rift, the pre-rift and early-984 phase structures in the Shire Rift (which are absent in the Zomba and Makanjira grabens) could 985 greatly impact the strain in the Lower-Shire Graben...e.g., consider the influence of the mechanical 986 load of the thick sequences of volcanic flows in the hanging wall of the Panga Fault and buried 987 Mwanza Fault (all beneath a part of the Lower-Shire graben), which could impact the throw on the 988 Thyolo Fault. Thus, in the absence of actual subsurface data on Thyolo fault throw, I think it is 989 speculative to make a statement like this. I understand that there is a need to assume a maximum 990 throw limit for the modelling, thus, I'll suggest that the authors revise this sentence by stating that 991 the estimate is an assumption.

| 993
   | As also discussed for Major Issue #3, we have removed the hanging-wall flexural modelling from our revised manuscript, so this section was removed anyway. In the revised manuscript, we make no reference to the total throw across the Thyolo Fault.                                                                                                                                             |
|--------------------------------------|----------------------------------------------------------------------------------------------------------------------------------------------------------------------------------------------------------------------------------------------------------------------------------------------------------------------------------------------------------------------------------------------------|
| 997
          | Lines 900-901: See my comments in "Major Issue 3" related to the structure of this rift segment as implemented in the model.                                                                                                                                                                                                                                                                       |
| 1000
       | Again, by removing the hanging-wall flexure modelling section, this sentence is not included in the revised manuscript.                                                                                                                                                                                                                                                                            |
| 1003
       | Line 909: The lower Shire graben is no more than 38-40 km wide (at its widest). I'm curious to know how the extent of this basin is estimated?                                                                                                                                                                                                                                                     |
| 1005
| Again, by removing the hanging-wall flexure modelling section, this section is not included in the revised manuscript.                                                                                                                                                                                                                                                                             |
| 1009
               | Line 914-915: This statement assumes that the Thyolo Fault is a Karoo-age border fault, which I think is problematic considering the observations in Castaing (1991). See "Major Issue 4".                                                                                                                                                                                                         |
| 1011
       | See our reply to Major Issue #4                                                                                                                                                                                                                                                                                                                                                                    |
| 1014
| Line 1620/Figure 1b: For figure 1b: It will be very helpful if you can add symbols or number the colored polygons of the different terranes shown. The colored polygons are faded into the grey scale hillshade map which makes the color slightly different from the ones shown in the legendby adding symbols or numbering, it makes it easier for the reader to identify where what terrane is. |
| 1018
       | As suggested, we have added text labels for the terranes in the revised manuscript                                                                                                                                                                                                                                                                                                                 |
| 1021
| Line 1621/Figure 1a: I'll suggest that you include the Aswa Shear Zone to this map as it is one of the most well-known lithospheric-scale shear zones in East Africa (see Daly et al., 1989; Ruotoistenmäki et al., 2014; Katumwehe et al., 2016; Saalmann et al., 2016). Other ones you could also include are the Lurio Shear Zone and the Sanangoe Shear Zone.                                  |
| 1025
       | For clarity, we no longer include major shear zones in this figure.                                                                                                                                                                                                                                                                                                                                |
| 1028
| Line 1639/Figure 2c: You might want to check the N-striking (80deg-dip) foliation anotated on the Namalambo Fault in Fig2c. The foliation trends in the Nsanje horst are NE-trending. The Namalambo Fault cuts the foliation. For reference, see "Structual Map of the Northern PortHerald Hills" in Bloomfield (1958).                                                                            |
| 1033
       | As suggested by the reviewer, we will remove this strike and dip measurements to reflect the regional NE-striking fabrics in the Nsanje Horst (albeit with local variations).                                                                                                                                                                                                                      |
| 1036
       | Line 1732/Figure A3: The orientation of the Zomba and Lower Shire profiles should also be stated in this caption.                                                                                                                                                                                                                                                                           |
| 1039
       | By removing the hanging-wall flexure modelling from the revised manuscript, this figure is not included anyway.                                                                                                                                                                                                                                                                                    |

Line 1736/Figure A3: You mean WSW-ENE?

[revised manuscript text omitted]